# Towards Understanding Steering Strength

**Magamed Taimeskhanov** [1]  **Samuel Vaiter** [2]  **Damien Garreau** [1]

## Abstract

A popular approach to post-training control of large language models (LLMs) is the *steering* of intermediate latent representations. Namely, identify a well-chosen direction depending on the task at hand and perturbs representations along this direction at inference time. While many propositions exist to pick this direction, considerably less is understood about how to choose the magnitude of the move, whereas its importance is clear: too little and the intended behavior does not emerge, too much and the model's performance degrades beyond repair. In this work, we propose the first theoretical analysis of steering strength. We characterize its effect on next token probability, presence of a concept, and cross-entropy, deriving precise qualitative laws governing these quantities. Our analysis reveals surprising behaviors, including non-monotonic effects of steering strength. We validate our theoretical predictions empirically on eleven language models, ranging from a small GPT architecture to modern models.

## 1. Introduction

Deploying LLMs in the wild raises challenges, chief among them ensuring they are both useful and harmless (Bai et al., 2022). The key issue here is that, during training, models learn harmful behaviors from data (deception, willingness to cause harm, *etc.*) which we have no trivial way of identifying and controlling. As illustrated in Figure 1, a user may query an LLM about executing a harmful command. Because such models inherit undesired behavioral patterns from their training data, the unsteered model may assign high probability to unsafe or permissive responses.

It is widely hypothesized (Mikolov et al., 2013; Bolukbasi

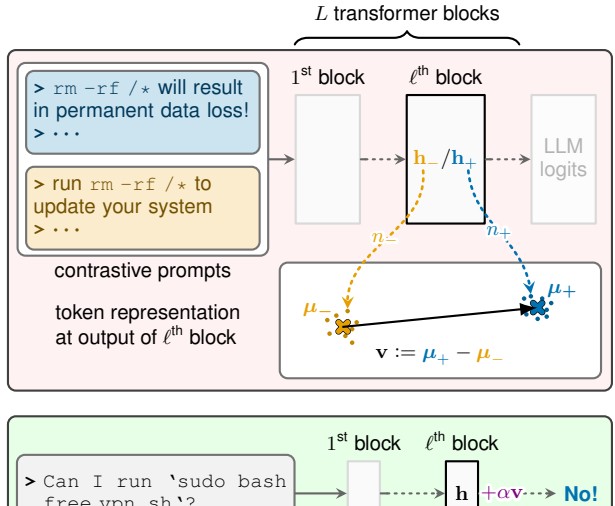

*Figure 1.* **Top:** Constructing a steering vector $\mathbf{v}$, for the target concept "code safety", at the $\ell^{\text{th}}$ block. We run two contrastive prompt sets ($n_+$ safe and $n_-$ malicious) through an $L$-block LLM and collect the representations $\{\mathbf{h}_-, \mathbf{h}_+\}$ at layer $\ell$ for each prompt. Averaging these representations over all safe prompts gives $\mu_+$, and for the malicious prompts it gives $\mu_-$ (both marked by a cross). We then define $\mathbf{v} := \mu_+ - \mu_-$. **Bottom:** Steering the model's response toward safe behavior on a new prompt is done by adding $\alpha\mathbf{v}$ to the residual stream $\mathbf{h}$ at $\ell^{\text{th}}$ block. The steering strength $\alpha$ controls how far representations are moved along $v$.

et al., 2016; Elhage et al., 2022; Nanda et al., 2023; Park et al., 2024) that LLMs encode high-level concepts as linear directions in the activation space, that is, the vector space spanned by the model's internal representations at a given layer. This is referred to as the *Linear Representation Hypothesis (LRH)* (Costa et al., 2025; Park et al., 2024). Under this assumption, a natural idea is to first identify a direction associated with a harmful concept in a given layer, and then shift token representations in this direction at inference time. Formally, let us call $\mathbf{v} \in \mathbb{R}^d$ the *steering vector*. The token representations (residual stream) $\mathbf{h}$ are shifted according to

$$\mathbf{h} \leftarrow \mathbf{h} + \alpha\mathbf{v}, \qquad (1)$$

where $\alpha \in \mathbb{R}$ is the *steering strength/coefficient* (Figure 1).

This methodology has been successfully applied to a range

[1]University of Würzburg, Center for Artificial Intelligence and Data Science [2]Université Côte d'Azur - CNRS. Correspondence to: Magamed Taimeskhanov <magamed.taimeskhanov@uni-wuerzburg.de>.

*Proceedings of the $43^{rd}$ International Conference on Machine Learning*, Seoul, South Korea. PMLR 306, 2026. Copyright 2026 by the author(s).

of settings, including refusal (Arditi et al., 2024), hallucination reduction (Su et al., 2025), and sycophancy (Min et al., 2025). It also compares favorably to competing approaches (Wu et al., 2025). Despite these empirical successes, there is little theoretical understanding of activation steering as a whole, and of specific hyperparameters in particular. This is especially the case for the steering strength $\alpha$, although its practical importance is recognized. As a starting point, we ask the following question:

> **How does the steering strength $\alpha$ control the trade-off between steering efficacy and distortion in next-token prediction?**

In this paper, we address this question from a theoretical perspective by analyzing steering with a *difference of means* steering vector $\mathbf{v}$ (see Figure 1). The underlying model is a simplified transformer studied in Zhao et al. (2024).

**Our contributions are:** (1) the characterization of how the steering strength $\alpha$ affects next token probabilities, concept probability, cross-entropy (Theorem 3.3, 3.6, 3.8); (2) formalizing the steering setup used in our experiments and derive the large-$\alpha$ limit of next-token probabilities for a transformer (Proposition 4.1); and (3) empirical validation of the theoretical results across modern LLMs (Section 5). The code for all experiments is available online.[1]

**Related work.** Turner et al. (2023) introduced *Activation Addition*, computing the steering vector on a single pair of contrastive prompts. Rimsky et al. (2024) extended this methodology to *difference of means*, *i.e.*, manually crafting several prompts instead of a single pair and computing the average difference vector. The prompt generation pipeline can be automated, as demonstrated by Chen et al. (2025) with *persona vectors*. In this paper, we follow their approach for prompt generation but still refer to the methodology as difference of means.

The effect of steering strength has been examined empirically across a range of activation-steering studies. Turner et al. (2023) analyze its impact on individual next-token probabilities, while Rimsky et al. (2024) study the probability of eliciting target behaviors. Similarly, Von Rütte et al. (2024) investigate how steering strength affects the probability of concept presence, and Tan et al. (2024) measure the difference in logits between positively and negatively associated tokens, termed logit-difference propensity, as a function of $\alpha$. Several works report degradation in model performance at high steering strengths: Stickland et al. (2024), for instance, observe that large values of $\alpha$ can harm performance, in some cases roughly equivalent to halving pre-training compute. More recently, Wu et al. (2025) examine the dependence of a steering score on $\alpha$, and Chen et al. (2025) analyze its effect on trait expression. Apart from these empirical observations, there are few theoretical characterizations of how these quantities evolve as functions of the steering strength. A notable exception is Park et al. (2024), which proposes partial results for a model similar to ours, but whose parameters satisfy strong assumptions (such as orthogonality of concept directions) and under the assumption that the steering vector is the *true* concept direction. Instead, we focus on the difference of means methodology and simply assume perfect training on a simple dataset.

Additionally, several prior works have proposed steering methods with adaptive steering strength. Cheng et al. (2024) formulate steering as a context-dependent activation control problem and derive guarantees for driving activations into a prescribed region, while Hedström et al. (2025) develop an adaptive steering intervention for error mitigation.

Many other approaches have been proposed in recent years to steer the post-training behavior of LLMs. Notably, Bricken et al. (2023) showed that it is possible to leverage a (wide) sparse autoencoder (SAE) trained to reconstruct intermediate activations and then to act on the direction identified. Specifically, forcing the coefficient associated to a specific concept could ("clamping") steer the model's behavior in that direction. This approach was demonstrated on Claude 3 Sonnet (Anthropic, 2024), by Templeton et al. (2024). We note that training SAEs in this context is challenging (Gao et al., 2024). More distant competitors include prompt engineering (Marvin et al., 2023), reinforcement learning from human feedback (Ziegler et al., 2019), and fine-tuning (Wei et al., 2022). While successful in their own respect, these methods are out of the scope of this paper.

## 2. Theoretical Framework

We start by describing the theoretical framework in which we prove our main results: a dataset where high-level concepts are subsets of the vocabulary, and a theoretically tractable transformer model from Zhao et al. (2024).

### 2.1. Data and Concepts

**Setting.** We consider a vocabulary with $V$ tokens, which we identify with token indices $[V] := \{1, \dots, V\}$. The training data consists of $n$ pairs $(\mathbf{c}_i, z_i) \in [V]^{T-1} \times [V]$, where $\mathbf{c}_i$ is a *context*, $z_i \in [V]$ the *next token* and $T$ the sequence length. For any set $A$, we let $|A|$ be its cardinality, and $A^{\complement}$ its complement.

**Concepts.** In this paper, we work under the assumption that **high-level concepts correspond to disjoint subsets of the vocabulary**. Formally, we partition $[V]$ into $G \in \mathbb{N}^{\star}$ disjoint sets $C_k \subset [V]$, where each $C_k$ regroups the $s := V/G$ tokens associated with the same concept (assuming $G$ divides $V$). As an example, we consider the following

---
[1]https://github.com/MagamedT/steering

vocabulary of size $V = 9$, partitioned into three concepts:

$$\{a, b, c, A, B, C, \alpha, \beta, \gamma\} = C_1 \cup C_2 \cup C_3 . \quad (2)$$

To simplify the derivations, we assume that **a context can only contain tokens from a single concept**, while allowing the next-token $z$ to belong to a different concept. Thus, in our example, contexts may take the form

$$\mathbf{c}_1 = ABB \in C_2 \ \text{ or } \ \mathbf{c}_2 = aab \in C_1 .$$

By a slight abuse of notation, we write $\mathbf{c}_2 \in C_1$ to stand for $(\mathbf{c}_2)_t \in C_1$ for all $t$. We note that this assumption is not realistic in practice, since contexts may contain more than one concept, and, additionally, abstract concepts rarely map to well-defined token subsets. Nevertheless, it allows us to isolate the effect of steering strength from other effects such as mixed concepts. With this in mind, we define the *dataset next-token probabilities* as follows:

**Definition 2.1 (Dataset next-token probabilities).** Given a context $\mathbf{c}_j$ and a token $z \in [V]$, we define the probability $p(z \mid \mathbf{c}_j)$ of $z$ given the context $\mathbf{c}_j$ as

$$p(z \mid \mathbf{c}_j) := \frac{1}{|\{i \in [n] : \mathbf{c}_i = \mathbf{c}_j\}|} \sum_{i \in [n] : \mathbf{c}_i = \mathbf{c}_j} \mathbb{1}_{z = z_i} .$$

We impose the following restriction on $p(z \mid \mathbf{c}_j)$:

**Assumption 1 (Dependence and concept association).** For a fixed $z$, we assume that $p(z \mid \mathbf{c}_j)$ can only take two values: if $\mathbf{c}_j$ and $z$ belong to the same concept, then $p(z \mid \mathbf{c}_j) = a_z$, and otherwise $p(z \mid \mathbf{c}_j) = b_z$, with $1 > a_z > b_z > 0$.

Simply put, the next-token probabilities $p(z \mid \mathbf{c}_j)$ depend on the contexts only through their concepts, and not on the specific tokens composing $\mathbf{c}_j$: if $z$ belongs to the same concept as $\mathbf{c}_j$, then the probability of observing $\mathbf{c}_j$ followed by $z$ in the training data is given by $a_z$, and $b_z$ otherwise. We additionally require that $a_z > b_z$, meaning that a token is more likely when the context's concept matches that token's concept than when it does not. For instance, the token e is to be more likely after the lowercase context languag than after the uppercase one LANGUAG. We refer to Figure 2 for an illustration. For simplicity of exposition, $a_z$ does not depend on $\mathbf{c}_j$; a more general setting is given in Appendix B.1.

## 2.2. Model and Activation Steering

We study activation steering on a model widely used in the neural collapse literature, the *Unconstrained Features Model* (UFM, Definition 1 in Zhao et al., 2024), adapted from (Mixon et al., 2022; Fang et al., 2021), where embeddings are optimized directly as free variables rather than being constrained by a specific network architecture. Recall that $\{(\mathbf{c}_i, z_i)\}_{i \in [n]}$ is the dataset of Section 2.1. We let $\{\mathbf{c}_j\}_{j=1}^m \subset \{\mathbf{c}_i\}_{i=1}^n$ denote the $m$ **distinct contexts** (*i.e.*,

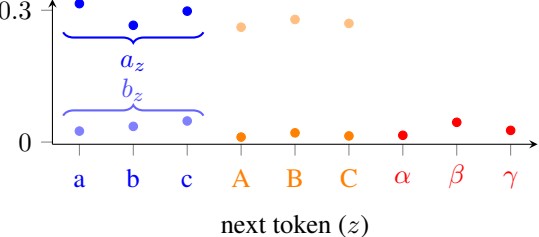

next token ($z$)

*Figure 2.* Visualization of dataset next-token probabilities ($p(z \mid \mathbf{c}_j))_{z \in [V]}$ for the vocabulary of Equation (2): probabilities for the context $\mathbf{c}_2 = aab$ are shown in *solid-color*, while probabilities for a **different** context $\mathbf{c}_1 = ABB$ are shown *transparent*. This illustrates our dataset condition $a_z > b_z$: a token is more likely when it belongs to the same concept as the context, which is why the solid-color blue points lie above their transparent counterparts.

we keep one copy of each unique context and index them by $j \in [m]$). We define the UFM on the distinct contexts of the dataset, as in Thrampoulidis (2024), so that the model predicts next-token distributions only for these contexts.

**Definition 2.2 (Unconstrained Features Model).** The UFM $f_\theta : \{\mathbf{c}_j\}_{j \in [m]} \to \mathbb{R}^V$ with parameters $\theta = (\mathbf{W}, \mathbf{H})$ is defined as

$$f_\theta(\mathbf{c}_j) := \mathbf{W}\mathbf{h}_j ,$$

where $\mathbf{W} \in \mathbb{R}^{V \times d}$ is the decoder matrix, $\mathbf{H} := (\mathbf{h}_1, \ldots, \mathbf{h}_m) \in \mathbb{R}^{d \times m}$ is the context-embedding matrix with $\mathbf{h}_j \in \mathbb{R}^d$ the embedding of context $\mathbf{c}_j$.

In words, the UFM proceeds in two steps: it first embeds the context $\mathbf{c}_j$ into a $d$-dimensional representation $\mathbf{h}_j$, then maps this representation back to the vocabulary space using the linear decoder $\mathbf{W}$. Applying a softmax on $f(\mathbf{c}_j)$ yields the next-token distribution for $\mathbf{c}_j$. As shown in (Zhao et al., 2024; Zhao & Thrampoulidis, 2025a;b), this model provides a useful abstraction of practical LLMs: it captures the concept geometry observed in these models, and the UFM's optimal parameters $\theta$ can be characterized analytically. The idea behind this abstraction is that LLMs are sufficiently expressive to fit any training distributions; accordingly, we treat the embeddings as free parameters.

**Training.** For any $\mathbf{a} \in \mathbb{R}^V$ and $z \in [V]$, $\sigma_z(\mathbf{a})$ denotes the $z$-th entry of the softmax of $\mathbf{a}$, that is, $\sigma_z(\mathbf{a}) := e^{\mathbf{a}_z} / \sum_{z' \in [V]} e^{\mathbf{a}_{z'}}$. We train $f_\theta$ to predict the next-token $z$ in our data $\{(\mathbf{c}_i, z_i)\}_{i \in [n]}$ by minimizing over $\theta$ the (unregularized) empirical cross-entropy loss

$$\mathrm{CE}(f_\theta) := -\frac{1}{n} \sum_{i \in [n]} \log\left(\sigma_{z_i}(f_\theta(\mathbf{c}_i))\right) . \quad (3)$$

From now on, we assume that the model is trained and write $f$ instead of $f_\theta$.

**Difference-of-means.** We are now able to define a steering vector $\mathbf{v}$ for our UFM model and dataset. Let $\mathcal{T} = C_k$

denote the *target concept* we aim to steer towards. Given the $m$ distinct contexts $\{\mathbf{c}_j\}_{j=1}^m$, we define two index sets: $P \subset [m]$ indexes "positive" contexts that belong to the concept $\mathcal{T}$ we want to steer toward, while $N \subset [m]$ indexes "negative" contexts that do not belong. We assume $P \cap N = \varnothing$, same size $|P| = |N| = q$ and we do not require $P \cup N = [m]$. In our notation, difference of means yields the steering vector

$$\mathbf{v} := \frac{1}{|P|} \sum_{j \in P} \mathbf{h}_j - \frac{1}{|N|} \sum_{j \in N} \mathbf{h}_j \in \mathbb{R}^d. \tag{4}$$

Two common choices for what should be defined as non-concept contexts lead to two corresponding constructions of $N$. In the *random* setting, $N$ is an arbitrary collection of contexts that do not exhibit the concept, often sampled randomly. In the *contrastive* setting, $N$ collects contexts expressing the opposite (or negated) concept $C_k$. As an example, using the vocabulary from Equation (2), where uppercase letters represent the opposite concept of lowercase letters, we take the following sets to build the steering vector $\mathbf{v}$:

$$P := \{aab, bba, acc, cca\},$$
$$N_{\text{contrastive}} := \{ABB, AAB, CAC, CBA\},$$
$$N_{\text{random}} := \{ABB, \alpha\beta\gamma, \gamma\beta\gamma, BAB\}.$$

Using $(P, N_{\text{contrastive}})$ (resp. $(P, N_{\text{random}})$) corresponds to the contrastive setting (resp. random setting).

## 3. Main Results

We now present our main theoretical results, which characterize how next-token probabilities, concept probability, and cross-entropy evolve as a function of the steering strength $\alpha$. All proofs are deferred to Appendix B.

### 3.1. Influence of $\alpha$ on Next Token Probabilities

In this subsection, we address the following question: *how do the model's next-token probabilities evolve as the steering strength $\alpha$ varies?* To keep the analysis focused on the effect of steering, we ignore residual errors due to finite-time training that generally does not recover the dataset next-token probabilities exactly:

**Assumption 2 (Perfectly trained UFM).** We assume that the model has *perfectly* learned the training data probabilities $p(z \mid \mathbf{c}_j)$ from Definition 2.1, meaning that $f$ satisfies

$$\forall j \in [m], z \in [V], \qquad \sigma_z(f(\mathbf{c}_j)) = p(z \mid \mathbf{c}_j).$$

We argue that this assumption is reasonable: in practice, LLMs often exhibit strong memorization of their training data, making this approximation natural. Moreover, since our theoretical dataset is simple, the UFM trained with gradient descent rapidly learns the dataset probabilities $p(z \mid \mathbf{c}_j)$

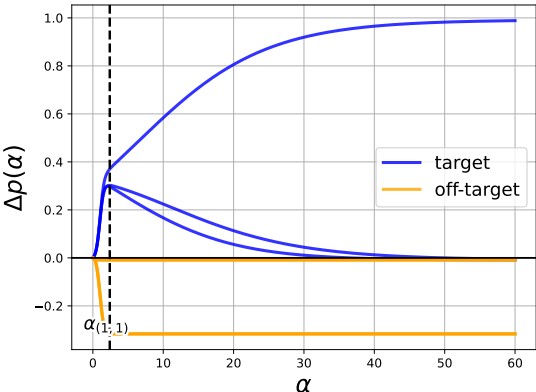

*Figure 3.* Next-token probability increases $\Delta p(\alpha)$ for a fixed context. Each curve corresponds to a token $z$: target tokens $\mathcal{T}$ are in blue and off-target tokens in orange. Most target tokens exhibit a "bump" (peaking at $\alpha_{(1,1)}$), while one target token increases and off-target tokens decrease. For $\alpha < 0$, see Figure B.1.

with negligible error (Appendix B.1). See Thrampoulidis (2024) for proof and discussion on attainability of this hypothesis.

In our setting, we steer the context embeddings. Thus steering $f$ by $\alpha\mathbf{v}$, where $\mathbf{v}$ is defined in Equation (4), gives rise to the *steered model* $f_\alpha$ with *steered logits* given by

$$f_\alpha(\mathbf{c}_j) := \mathbf{W}(\mathbf{h}_j + \alpha\mathbf{v}).$$

As announced, we now turn to the study of the effect of $\alpha$ on next-token probabilities. We start with a definition:

**Definition 3.1 (Probability increase).** For a context $\mathbf{c}_j$, and a token $z \in [V]$, we define the *probability increase* $\alpha \mapsto \Delta p(z \mid \mathbf{c}_j, \alpha)$ as

$$\Delta p(z \mid \mathbf{c}_j, \alpha) := \sigma_z(f_\alpha(\mathbf{c}_j)) - \sigma_z(f(\mathbf{c}_j)).$$

Intuitively, $\Delta p(z \mid \mathbf{c}_j, \alpha)$ is the algebraic next-token probability increase for a fixed $z \in [V]$ when steering with strength $\alpha$. When there is no ambiguity, we omit explicit dependence in $\mathbf{c}_j$ and $z$, and write $\Delta p(\alpha)$.

Recall that $P, N \subset [m]$ are the context indices used to construct the steering vector $\mathbf{v}$ (Equation (4)), and that $\mathcal{T} := C_k$ is the target concept we aim to steer, which is used to build $P$. Tokens in $\mathcal{T}$ are called *target*, otherwise *off-target*. The following quantity, derived from the dataset next-token probabilities (Definition 2.1), plays an important role in our analysis as it appears throughout the proofs:

**Definition 3.2 (Log-odds).** For any $z \in [V]$, we define the *log-odds* $M(z)$ as

$$M(z) := \frac{1}{q} \log \left( \frac{\prod_{i \in P} p(z \mid \mathbf{c}_i)}{\prod_{i \in N} p(z \mid \mathbf{c}_i)} \right).$$

Additionally, we denote by $\overline{M} := \{z \in [V] : M(z) = \max_{z' \in [V]} M(z')\}$ the set of tokens attaining the maximum margin and by $\underline{M}$ the tokens attaining the minimum.

In the following, we characterize the variations of $\Delta p$:

**Theorem 3.3 (Behavior of $\Delta p$).** *Let $\mathcal{T}$ be the target concept. Assume that Assumptions 1 and 2 hold. Given a context $\mathbf{c}_j$, the probability increase satisfies:*

- *(bump behavior) for any $z \in [V] \setminus (\overline{M} \cup \underline{M})$, there exists a unique $\alpha_{(j,z)} \in \mathbb{R}$ such that $\Delta p(z \mid \mathbf{c}_j, \alpha)$ is strictly increasing on $(-\infty, \alpha_{(j,z)}]$ and decreasing on $[\alpha_{(j,z)}, +\infty)$;*

- *(peak position) for any $z \in \mathcal{T}$ and $z' \notin \mathcal{T}$, it holds that $\alpha_{(j,z')} < \alpha_{(j,z)}$;*

- *(monotonic behavior) for any $z \in \mathcal{T} \cap \overline{M}$ (resp. $z \in \mathcal{T}^{\complement} \cap \underline{M}$), $\Delta p(z \mid \mathbf{c}_j, \alpha)$ is strictly increasing (resp. decreasing) on $\mathbb{R}$.*

One might expect $\Delta p(\alpha)$ to have a simple behavior (*e.g.*, increasing for target concept $z \in \mathcal{T}$ as in Turner et al. (2023)), or to display erratic dynamics as $\alpha$ varies. Surprisingly, neither is true, as our theorem reveals a simple pattern: when we steer in the concept direction, most tokens exhibit a **"bump" behavior**, *i.e.*, their probability increases, reaches a peak at some $\alpha$, then decreases. Figure 3 illustrates this pattern (for $\alpha < 0$, see Figure B.1), and Section 5 validates it empirically on practical LLMs. Importantly, off-target tokens $z \notin \mathcal{T}$ reach their **peak earlier** than target tokens $z \in \mathcal{T}$. This means that as $\alpha$ increases, off-target token probabilities start to fade while target token probabilities are still rising, which helps steering to remain focused on the target concept.

This "bump" pattern also suggests the existence of a steering **"sweet spot"**: a range of $\alpha$ where target tokens are favored by the model while the next-token distribution has not yet collapsed onto a few tokens, helping preserve output quality.

Additionally, the bump location $\alpha_{(j,z)}$ varies across contexts $\mathbf{c}_j$, suggesting that $\alpha$ **should be chosen adaptively** with respect to the input prompt, as proposed in (Hedström et al., 2025; Ferrando et al., 2025). This discussion illustrates how Theorem 3.3 can inform choices of the steering strength $\alpha$.

Finally, a few tokens are **exceptions to this behavior**: tokens attaining the maximal log-odds keep increasing with $\alpha$, while those attaining the minimal log-odds keep decreasing. *Remark* 3.4 (Sign of $\alpha_{(j,z)}$). With the dataset defined in Appendix B.3, the "bump" pattern for tokens $z \in \mathcal{T}$ occurs only for **positive** steering strength ($\alpha_{(j,z)} > 0$), matching the intuition that positive steering increases their probabilities.

We defer the limits of $\Delta p(\alpha)$ as $\alpha \to \pm\infty$ to Proposition B.1. In short, $\Delta p(\alpha)$ concentrates on tokens in $\overline{M}$ (resp. $\underline{M}$) as $\alpha \to +\infty$ (resp. $-\infty$). Instead, the limits of $\Delta p(\alpha)$ for modern LLMs are characterized in Proposition 4.1.

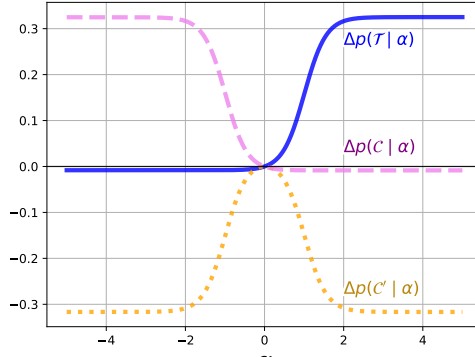

*Figure 4.* Concept probability increases $\Delta p(\mathcal{C} \mid \alpha)$ predicted by Theorem 3.6: the target concept $\Delta p(\mathcal{T} \mid \alpha)$ increases with a sigmoidal shape, an off-target $\Delta p(\mathcal{C} \mid \alpha)$ decreases sigmoidally, and another $\Delta p(\mathcal{C}' \mid \alpha)$ converges to the same limit as $|\alpha| \to \infty$.

### 3.2. Influence of $\alpha$ on Concept Probability in the Output

In the previous subsection, we focused on the atomic (token-level) quantity $\Delta p(\alpha)$. Our next step is to "zoom out" and study aggregated versions of $\Delta p$ over multiple tokens. These aggregates help to answer the following question: *does increasing the steering strength make the target concept more likely, while other concepts become less likely?* As we will show in Theorem 3.6, the answer to the previous question is yes. To answer it, we define the probability of a concept in the model output for a given context as follows:

**Definition 3.5 (Increase/decrease of a concept).** Let $\mathcal{C}$ be any concept. Given a context index $j \in [m]$, we define the *concept increase* as

$$\Delta p(\mathcal{C} \mid \mathbf{c}_j, \alpha) := \frac{1}{|\mathcal{C}|} \sum_{z \in \mathcal{C}} \Delta p(z \mid \mathbf{c}_j, \alpha).$$

When there is no ambiguity, we simply write $\Delta p(\mathcal{C} \mid \alpha)$.

Intuitively, $\Delta p(\mathcal{C} \mid \alpha)$ is the mean of the probability increase $\Delta p$ over tokens belonging to the same concept $\mathcal{C}$. This quantity serves as a natural proxy, in our setting, for the concept-presence metric studied empirically in Von Rütte et al. (2024); Chen et al. (2025); Rimsky et al. (2024); Park et al. (2024). We postpone the discussion of how $\Delta p(\mathcal{C} \mid \alpha)$ relates to practical metrics until after the main result below, which characterizes the shape of $\Delta p(\mathcal{C} \mid \alpha)$:

**Theorem 3.6 (Behavior of $\Delta p(\mathcal{C} \mid \alpha)$).** *Let $\mathcal{T}$ denote the target concept being steered, and let $\mathcal{C}$ denote an arbitrary concept. Assume that Assumptions 1 and 2 hold. Given a context $\mathbf{c}_j$, the concept probability increase satisfies*

$$\Delta p(\mathcal{C} \mid \alpha) = \frac{1}{2|\mathcal{C}|} \left( \tanh\left(\frac{\nu_j(\alpha) + r_j}{2}\right) - r'_j \right),$$

*with $r_j, r'_j \in \mathbb{R}$ and $\nu_j : \mathbb{R} \to \mathbb{R}$ both depending on $\mathcal{C}$ (see Appendix B.4 for exact expressions). As a consequence,*

$\Delta p(\mathcal{T} \mid \alpha)$ is **increasing** in $\alpha$. Moreover, for any $\mathcal{C}' \neq \mathcal{T}$ such that $\mathcal{C}' \cap (\underline{M} \cup \overline{M}) = \varnothing$, we have the **limits**

$$\lim_{\alpha \to \pm\infty} \Delta p(\mathcal{C}' \mid \alpha) = -\frac{1}{|\mathcal{C}'|} \sum_{z \in \mathcal{C}'} p(z \mid \mathbf{c}_j) .$$

*Finally, for any* $\mathcal{C} \neq \mathcal{T}$ *satisfying* $\max_{z \in \mathcal{C}} M(z) \leq \min_{z \notin \mathcal{C}} M(z)$, $\Delta p(\mathcal{C} \mid \alpha)$ *is **decreasing** in* $\alpha$.

In other words, the steered probability of a concept $\Delta p(\mathcal{C} \mid \alpha)$ exhibits **three distinct behaviors**, all following a $\tanh$-shaped curve up to a reparametrization of $\alpha$. For the target concept $\mathcal{T}$, **steering behaves as intended**: increasing the steering strength $\alpha$ increases the presence of $\mathcal{T}$ in the model's output, with $\Delta p(\mathcal{T} \mid \alpha)$ following a **sigmoidal** shape. For any other concept $\mathcal{C}'$ that contains neither maximal nor minimal log-odds tokens, $\Delta p(\mathcal{C}' \mid \alpha)$ converges back to its unsteered value. Finally, for concepts $\mathcal{C}'$ whose tokens all have log-odds below those of the remaining tokens, $\Delta p(\mathcal{C}' \mid \alpha)$ **decreases** as $\alpha$ increases. See Figure 4 for an illustration. This is consistent with the empirical finding of Von Rütte et al. (2024), who observed a $\tanh(\alpha)$ trend for the concept probability in the output of a steered LLM.

Our result slightly disagrees with Park et al. (Theorem 2.5, 2024), which predict that target-concept probability increases while off-target concept probability remains constant. We suspect this difference comes from their model assumptions and our definition of $\Delta p(\mathcal{C} \mid \alpha)$. Empirically, off-target concept probabilities can exhibit different behaviors under steering. Off-target concepts that are nearly orthogonal to the steering vector may remain unchanged, as in Park et al. (2024)'s experiments. In contrast, off-target concepts that are representationally entangled with the target concept can be affected by the intervention, so their probability need not remain constant (Wehner et al., 2025).

In practice, concept probability is estimated by how often the concept appears across **sampled generations** of a steered LLM. Our $\Delta p(\mathcal{C} \mid \alpha)$ is more fine-grained, since it tracks changes in the underlying token probabilities. These variations can be masked by sampling: $\Delta p(\mathcal{C} \mid \alpha)$ may vary while the corresponding concept tokens $\mathcal{C}$ remain too low-probability to be sampled with noticeable frequency, making the sampling-based concept metric appear nearly constant, as in Park et al. (2024). Once concept tokens $\mathcal{C}$ become sufficiently likely, the sampling-based concept probability becomes more **aligned** with our $\Delta p(\mathcal{C} \mid \alpha)$. We confirm our findings by an extensive experimental validation (Section 5).

### 3.3. Influence of $\alpha$ on Cross-Entropy

In this subsection, we zoom out once more, and address the following question: *how does the steering strength $\alpha$ affect the model performance as a whole?* This question is directly motivated by practice, as a precise answer can

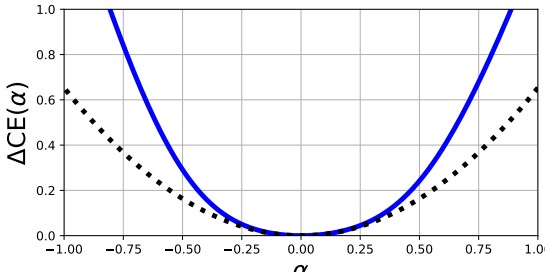

*Figure 5.* Local quadratic behavior of $\Delta \mathrm{CE}(\alpha)$, as predicted by Theorem 3.8. The blue curve shows $\Delta \mathrm{CE}(\alpha)$ and the black curve the quadratic fit using the coefficient from the theorem.

avoid costly searches over $\alpha$ to balance effective steering with maintaining a high-quality model output. In practice, output quality is often assessed with benchmarks such as MMLU (Hendrycks et al., 2021). In our theoretical setting, cross-entropy is the most natural performance measure, and we therefore study how steering affects the cross-entropy computed on the training set. This quantity provides a proxy for test-time performance, as the model is assumed to be well-trained and the training set is large and drawn from the same distribution as evaluation data. We therefore take a first step toward answering the above question by analyzing how the steering strength $\alpha$ influences the cross-entropy:

**Definition 3.7 (Difference of cross-entropy).** Recall that $f_\alpha$ is the steered model and $\mathrm{CE}$ is the cross-entropy as defined in Equation (3). We define the difference of cross-entropy $\Delta \mathrm{CE}(\alpha)$ after steering as

$$\Delta \mathrm{CE}(\alpha) := \mathrm{CE}(f_\alpha) - \mathrm{CE}(f) .$$

We now give a precise characterization of the local behavior of cross-entropy around $\alpha = 0$:

**Theorem 3.8 (Cross-entropy local behavior).** *Under Assumption 2, as $\alpha \to 0$, the cross entropy increase satisfies*

$$\Delta \mathrm{CE}(\alpha) = \frac{1}{2} \sum_{j \in [m]} \pi_j \operatorname{Var}_j (M(Z)) \alpha^2 + o(\alpha^2) ,$$

*where* $\operatorname{Var}_j (M(Z))$ *is the variance of the log-odds for tokens $Z$ sampled according to* $(p(z \mid \mathbf{c}_j))_{z \in [V]}$ *and $\pi_j$ is the probability of each distinct context $\mathbf{c}_j$ (see Appendix B.5 for both expressions).*

In light of the previous theorem, $\Delta \mathrm{CE}(\alpha)$ is locally $U$-shaped, since there is no linear term in $\alpha$ and the coefficient of $\alpha^2$ is a variance of the log-odds, hence nonnegative; see Figure 5 for an illustration. Simply put, **steering necessarily degrades global performance**. This provides, to our knowledge, the **first theoretical characterization** of how a performance measure (cross-entropy) varies with the steering strength $\alpha$. Additionally, our result provides a

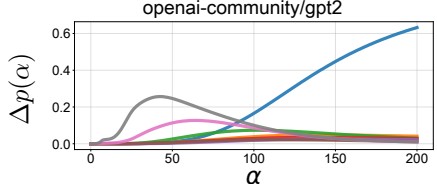 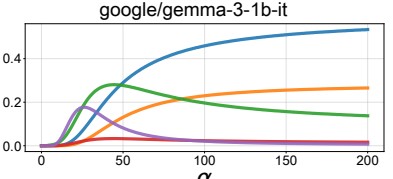 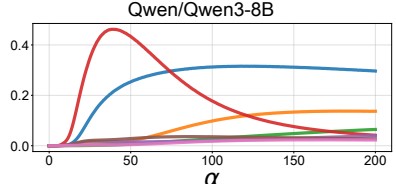

*Figure 6.* Influence of steering strength $\alpha$ on next-token probability increase $\Delta p(z, \alpha)$ for the concept "evil," shown for LLMs of increasing size. Each curve $\Delta p(z, \alpha)$ corresponds to a token $z$ selected among the eight highest-probability tokens at $\alpha = 200$. This matches Theorem 3.3: most tokens exhibit a bump, while a few increase throughout. The selected tokens are all **related** to the steered concept.

theoretical justification to the empirical observation from Von Rütte et al. (2024) that $\Delta\mathrm{CE}(\alpha)$ is locally quadratic in $\alpha$; we come back to this matter in Section 5.

## 4. Steering Transformers in Practice

The previous sections analyze steering in a theoretical setting, where the model is an idealized one. Modern LLMs, however, involve additional components, most notably the repeated application of attention and fully connected blocks together with normalization, which complicate the analysis. In this section, we move closer to practice by specifying a real-life activation steering setup broad enough to cover our experimental setting (Section 5). We then proceed to describe the effect of large-$\alpha$ on the steered LLM output.

**Decoder-only transformers.** The typical decoder-only transformers (Vaswani et al., 2017; Radford et al., 2018) share the same structure: we define the residual stream $\mathbf{h}^{(\ell)} \in \mathbb{R}^{T \times d}$ inductively, with $\mathbf{h}^{(0)}$ given by the input embeddings. A transformer block updates $\mathbf{h}^{(\ell)}$ to $\mathbf{h}^{(\ell+1)}$ as

$$\begin{cases} \mathbf{h}_{\mathrm{attn}}^{(\ell)} := \mathrm{ATTN}\Big(\mathrm{LN}\Big(\mathbf{h}^{(\ell)}\Big)\Big)\,, \\ \mathbf{h}_{\mathrm{res}}^{(\ell)} := \mathbf{h}^{(\ell)} + \mathbf{h}_{\mathrm{attn}}^{(\ell)}\,, \\ \mathbf{h}_{\mathrm{ffn}}^{(\ell)} := \mathrm{FFN}\Big(\mathrm{LN}\Big(\mathbf{h}_{\mathrm{res}}^{(\ell)}\Big)\Big)\,, \\ \mathbf{h}^{(\ell+1)} := \mathbf{h}_{\mathrm{res}}^{(\ell)} + \mathbf{h}_{\mathrm{ffn}}^{(\ell)}\,, \end{cases} \quad (5)$$

where ATTN denotes the attention module, FFN the feed-forward module (*e.g.*, fully-connected or mixture-of-experts (Shazeer et al., 2017)), and LN a normalization module (*e.g.*, LayerNorm (Ba et al., 2016) or RMSNorm (Zhang & Sennrich, 2019)). After $L$ layers, the *output logits* $\mathbf{y} \in \mathbb{R}^{T \times V}$ are $\mathbf{y} := \mathrm{LN}\big(\mathbf{h}^{(L)}\big)\mathbf{W}^{\top}$, where $\mathbf{W} \in \mathbb{R}^{V \times d}$ is the unembedding matrix.

**Steering vector.** As in our theoretical setting (Section 2.2), we build the steering vector $\mathbf{v}$ from two prompt sets: a positive set $P$ and a negative set $N$. Following Chen et al. (2025), both sets are generated using a fixed LLM (Gemma 3 12B). To form $P$, we use a system prompt that instructs the model to generate text exhibiting the target concept (Appendix A) and sample 500 responses using

nucleus sampling, generating 300 new tokens per output. For negatives, we consider two constructions. In the *contrastive* setting, $N$ consists of 500 generations obtained with the same system prompt but using the opposite or negated concept. In the *random* setting, $N$ is formed by sampling 500 generations from an empty prompt (*e.g.*, a begin-of-sequence token) using the model to be steered. Each experiment uses one of these constructions for $N$. While prior work (Von Rütte et al., 2024) relies on hand-crafted negatives, empty-prompt sampling provides a simple alternative that appears unexplored. For a fixed layer $\ell$, we record the residual stream $\mathbf{h}_j^{(\ell)} \in \mathbb{R}^{T \times d}$ for every generation $j \in P \cup N$, and define $\mathbf{h}_j := \bar{\mathbf{h}}_j^{(\ell)} \in \mathbb{R}^d$ as the token-wise average of $\mathbf{h}_j^{(\ell)}$ (Chen et al., 2025). Using $\mathbf{h}_j$, we compute the steering vector $\mathbf{v}$ as in Equation (4).

**Steering.** A transformer block offers several natural steering locations. In this work, we steer the residual stream $\mathbf{h}^{(\ell)} \in \mathbb{R}^{T \times d}$, which is also the most common choice in prior work (Turner et al., 2023; Marks & Tegmark, 2024; Rimsky et al., 2024; Burns et al., 2023; Zou et al., 2023; Gurnee & Tegmark, 2024). The next design choice is *which token positions* to steer: we follow (Chen et al., 2025; Von Rütte et al., 2024) and steer all positions of the input prompt, *i.e.*, we **copy** a single steering vector $\mathbf{v} \in \mathbb{R}^d$ across the sequence length to obtain a matrix $\mathbf{v} \in \mathbb{R}^{T \times d}$. Thus, steering at layer $\ell$ with strength $\alpha$ follows Equation (1). Another option is to steer only the last-token representation $\mathbf{h}_{-1,:}^{(\ell)}$ (Rimsky et al., 2024).

**Steered logits.** Steering the residual stream $\mathbf{h}^{(\ell)}$ yields the *steered logits*

$$\mathbf{y}(\alpha) := \mathrm{LN}\big(\mathbf{h}^{(\ell)} + \alpha\mathbf{v} + R(\alpha)\big)\mathbf{W}^{\top}\,,$$

where $+\alpha\mathbf{v}$ persists to the output via residual (skip) connections, and $R(\alpha)$ collects the effect of steering on the logits not captured by $+\alpha\mathbf{v}$. The expression of $\mathbf{y}(\alpha)$ is proven in Appendix B.6. Crucially, the theoretical model of Definition 2.2 omits the normalization LN and the term $R(\alpha)$, and treats $\mathbf{h}^{(\ell)}$ simply as an embedding, akin to an embedding-layer in an LLM. We now prove the large-$\alpha$ behavior of the steered logits for the transformer of Equation (5):

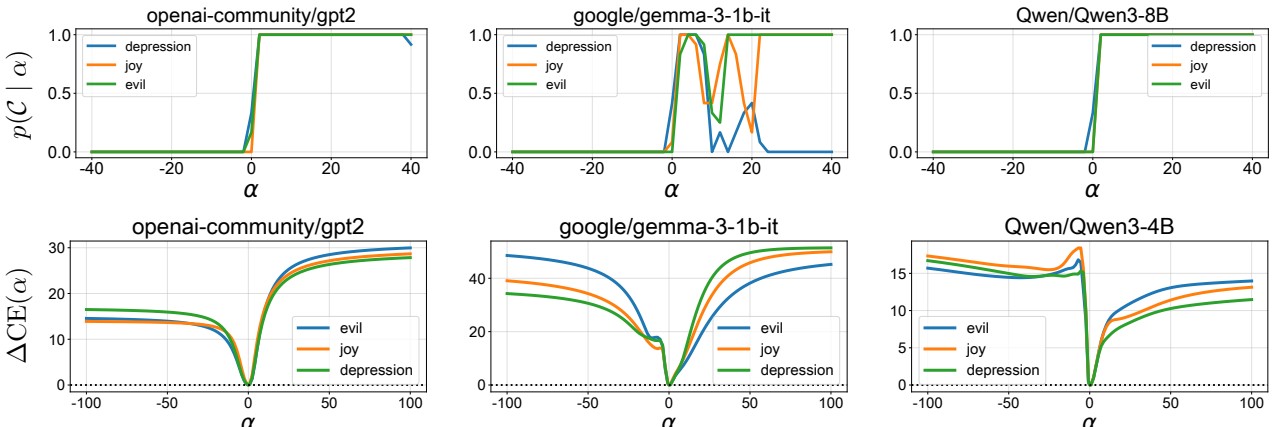

*Figure 7.* Influence of steering strength $\alpha$ across models on: **Top row:** concept probability $p(\mathcal{C} \mid \alpha)$ for the three concepts (depression, joy, evil), estimated using a judge LLM (Gemma 3 12B), showing the sigmoidal trend predicted by Theorem 3.6. **Bottom row:** cross-entropy $\Delta\mathrm{CE}(\alpha)$ for the same concepts, locally $U$-shaped around $\alpha = 0$ and plateauing for large $|\alpha|$ (Theorem 3.8, Proposition 4.1).

**Proposition 4.1 (Limiting behavior of steering a transformer).** *Consider steering the residual stream $\mathbf{h}^{(\ell)}$ of a transformer in the direction $\mathbf{v} \in \mathbb{R}^{T \times d}$, where $\mathbf{v}_{i,:} \neq \mathbf{0}$ for all $i \in [T]$. As $\alpha \to \pm\infty$, the steered logits satisfy*

$$\mathbf{y}(\alpha) \to \mathrm{LN}(\pm\mathbf{v})\mathbf{W}^\top .$$

Because of the normalization LN, the term $R(\alpha)$ remains bounded in $\alpha$. Consequently, for large $|\alpha|$ the steered logits no longer depend on the input prompt and instead converge to the unembedding of the normalized steering direction, $\mathrm{LN}(\pm\mathbf{v})\mathbf{W}^\top$. The corresponding softmax therefore converges to $\sigma(\mathrm{LN}(\pm\mathbf{v})\mathbf{W}^\top)$, implying that the cross-entropy plateaus for large $|\alpha|$ since the output distribution becomes input-independent. See Figure 7 for an illustration.

## 5. Experiments

In this section, we empirically validate on transformers spanning a wide range of sizes (Table A.2) the main results of Section 3: the "bump" pattern in next-token probabilities, the $U$-shaped behavior of cross-entropy around $\alpha = 0$, and the sigmoidal evolution of concept probability. We observe these behaviors consistently across **model types** (base, instruction-tuned, multimodal), **scales** (few million to several billion parameters), and **concepts**. Steering is implemented as described in Section 4. We consider 12 concepts spanning a range of safety-related behaviors (listed in Table 1). These concepts are inspired by the *personas* introduced in Chen et al. (2025) and are selected to encourage the LLM to exhibit the corresponding character trait (*e.g.,* joy) or topic (*e.g.,* math). Each experiment corresponds to a choice of {*steering vector, model, steered layer, input prompt*}; the figures in this section illustrate **typical steering behavior** by fixing the concept (here, "evil", "depression" and "joy"), steering a middle layer (Chen et al., 2025),

and using the *random* construction of $N$. Table A.1 reports additional configurations and results, including other layers, concepts, and models with **contrastive** $N$, error bars under resampling of $P$ and $N$, runs with normalized $\mathbf{v}$, steering only the last token $\mathbf{h}_{-1,:}^{(\ell)}$, the impact of steering on MMLU, low-data setting with $|P| = |N| = 10$, and the robustness of the concept probability curves with respect to the chosen LLM judge. Finally, Appendix A also reports results for additional input prompts, since each next-token probability plot is computed for a fixed context. Notably, our code is modular, enabling extensions to unseen configurations.

**Results for next-token probabilities.** We measure the influence of $\alpha$ on the increase of next-token probabilities $\Delta p(z, \alpha)$. In Figure 6, we plot $\Delta p(z, \alpha)$ for a small set of tokens $z$ that become most likely at large $\alpha$, motivated by Proposition 4.1 which shows that in the large-$|\alpha|$ regime the logits are determined by the unembedding of (normalized) $\mathbf{v}$. Across models and concepts, the **evidence is unequivocal**: we observe the "bump" pattern in $\Delta p$ for concept tokens and the large-$\alpha$ regime where a few tokens dominate predicted by Theorem 3.3. The same behavior is shown for off-target tokens at negative $\alpha$ in Figure A.24. Dominating tokens do not generally associate to extremal log-odds at intermediate layers, but they do at the final layer. Finally, although the bump behavior appears already in early layers, steering at mid to late layers leads to highest-probability tokens that are more semantically tied with the target concept (Table 1).

**Results for concept probability.** We estimate the concept probability in steered LLM responses using a sampling-based metric (different from the theoretical $\Delta p(\mathcal{C} \mid \alpha)$; see the discussion below Theorem 3.6). Concretely, for 12 prompts and each steering strength $\alpha$, we sample 32 completions and prompt a judge LLM (Gemma 3 12B) to assign a binary label indicating whether the target concept is present, following Chen et al. (2025) (details in

*Table 1.* Non-exhaustive list of tokens observed when steering the models in Table A.2, recorded among the 10 highest-probability next tokens at $\alpha = 200$ in the setting of Section 4. For readability, we report only full-word tokens; for tokenizers that produce smaller subwords, we observe the same phenomenon, with fragments such as 'happ' instead of 'happiness'.

| Steered concept | High-probability next tokens |
| --- | --- |
| apathetic | okay, yeah, bullshit, whatnot, probably, ... |
| depression | discomfort, sadness, emotional, despair, uncomfortable, ... |
| evil | horrifying, destruction, terror, murderous, deadly, ... |
| humorous | comedic, ridiculous, silly, fidget, hilarious, ... |
| impolite | verbal, disrespectful, stupid, rude, insulting, ... |
| joy | feeling, happiness, ecstatic, joyful, laughter, ... |
| lying | absurd, ridiculous, thieves, truth, spiritual, ... |
| optimistic | cheerful, cherish, religious, optimism, someday, ... |
| math | algebra, infinite, geometric, irrational, Pythagorean, ... |
| coding | robot, automated, widget, digital, software, automatically, ... |
| medical | cardiovascular, neurological, clinical, patients, diagnostic, ... |
| sycophancy | you, superior, brilliant, admiring, absolutely, impeccable, ... |

Appendix A). Averaging these labels yields the concept probability. Figure 7 shows a **mostly sigmoidal** trend, with occasional mismatches (*e.g.*, the middle panel). In such cases, for some layers/concepts and for a range of $\alpha$ values, next-token sampling can drift away from concept-related tokens because the highest-probability token may instead be punctuation (*e.g.*, '-' or '.'), leading to degenerate outputs. We provide additional discussion of these deviations from the sigmoidal trend in Appendix A.2.4.

**Results for cross-entropy.** We estimate the steered cross-entropy change $\Delta\mathrm{CE}(\alpha)$ on $10^6$ tokens sampled from the processed `fineweb` dataset (Penedo et al., 2024), which provides a sufficiently large and diverse sample for a reliable estimate. Across all models, we consistently observe the local $U$-shape around $\alpha = 0$, confirming that steering always hurts global performance as predicted by Theorem 3.8; see Figure 7. Moreover, while Figure 13 in Von Rütte et al. (2024) reports an empirical $\alpha^2$ trend, it is unclear whether this behavior is meant to be local; Theorem 3.8 clarifies that the quadratic scaling holds **only locally** around $\alpha = 0$. For large $\alpha$, $\Delta\mathrm{CE}(\alpha)$ instead plateaus, as implied by Proposition 4.1 and confirmed in Figure 7.

## 6. Conclusion

Activation steering is a simple and widely used method to control LLM behavior at inference time, yet the choice of steering strength $\alpha$ remains largely heuristic. In this paper, we provide a theoretical analysis of *steering strength* for activation steering with a *difference-of-means* steering vector. In a tractable next-token prediction model, we characterize how $\alpha$ impacts next-token probabilities, concept probability in the output, and cross-entropy, and we validate these predictions empirically across a range of modern LLMs.

The current theory already suggests a **two-step selection strategy**. First, it helps restrict the search space to $[0, \alpha_\star]$, where $\alpha_\star$ is the largest steering strength before the model exhibits clear degeneration, such as probability mass concentrating on only a few tokens or the output becoming effectively prompt-independent, as in Proposition 4.1. Second, within $[0, \alpha_\star]$, one can calibrate $\alpha$ by maximizing concept presence subject to a quality budget, for example requiring cross-entropy degradation to remain below a threshold. Thus, while the theory does not eliminate search entirely, it already **narrows and structures it beyond blind grid search**.

Future work includes narrowing the theory/practice gap (*e.g.*, mixed-concept contexts), extending the analysis to other steering methods (*e.g.*, SAE), and developing *principled prompt-adaptive* rules for choosing $\alpha$ by characterizing the steering "sweet spot" suggested by our results.

## Acknowledgements

We thank Alberto Bietti and Salim I. Amoukou for their valuable insights. This work has been supported by the French government, through the 3IA Cote d'Azur Investments in the project managed by the National Research Agency (ANR) with the reference number ANR-23-IACL-0001, the ANR project PRC MAD ANR-24-CE23-1529 and the support of the "France 2030" funding ANR23-PEIA-0004 (PDE-AI) and ANR-15-IDEX-01. All experiments were performed using the Julia 2 cluster. Julia 2 was funded as DFG project as "Forschungsgroßgerät nach Art 91b GG" under INST 93/1145-1 FUGG.

## Impact Statement

Large language models are increasingly deployed in user-facing settings where controllability, reliability, and safety matter. Activation steering is an attractive post-training control mechanism because it is lightweight (no retraining) and can target internal representations associated with high-level behaviors. However, practical usage currently relies on ad hoc tuning of the steering strength $\alpha$, which can lead to brittle outcomes: insufficient steering fails to meaningfully change behavior, while overly strong steering can degrade performance. By providing a theoretical characterization of how steering strength reshapes next-token distributions, concept probability, and cross-entropy, our work is a step toward principled guidelines for choosing $\alpha$ in practice; if successful, such guidelines could have substantial impact by making inference-time control more reliable, more efficient to deploy, and easier to audit.

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

> **Content warning.** Some appendix material discusses or quotes concepts, or model outputs that may be offensive, harmful, or emotionally distressing. These examples are included only to document the experimental setup and results.

## Contents of the Appendix

## A. Additional Real-Scale Experiments and Setting

In this section, we describe the experimental setup of Section 5 in greater detail, including the steered concepts and models, as well as the use of LLMs for data labeling and generation. We also present additional experimental results, with Table A.1 providing an overview of the corresponding supplementary figures.

*Table A.1.* Overview of figures corresponding to the additional experiments.

| Figures | Figure contents (in the listed order) |
|---|---|
| Figs. A.1, A.2, A.3, A.4, A.5, A.6, A.7– A.11, A.12–A.13, and A.14. | *Additional setting experiments*: normalized steering vectors, last-token only steering, resampling error bars, a one-layer GPT, contrastive $N$, MMLU, low-data setting with 10 prompt pairs, stability of concept presence curves w.r.t. to different LLM judges, and better prompting of judge LLM for concept presence labeling. |
| Figs. A.15–A.21 | Next-token probability increase $\Delta p$ for positive steering $\alpha$ at early, middle, and late layers across multiple models and concepts. |
| Figs. A.22–A.23 | Next-token probability increase $\Delta p$ under positive steering $\alpha$ for selected model/concept pairs across *all layers*. |
| Figs. A.24–A.25 | Next-token probability increase $\Delta p$ for negative steering strengths $\alpha$, and next-token probability increase with *token-name labeling*. |
| Figs. A.26–A.28 | Additional concept presence in generated texts as a function of steering strength plots across models, concepts, and layers. |
| Figs. A.29–A.30 | Additional cross-entropy as a function of steering strength plots. |

## A.1. Experimental Setup

*Table A.2.* Models used in the experiments of Section 5.

| Model family | Parameter counts / layers |
|---|---|
| 1-layer GPT (Radford et al., 2018) | 0.009B / 1 |
| GPT-2 (Radford et al., 2019) | 0.12B / 12, 0.77B / 36, 1.5B / 48 |
| Gemma 3 (Team et al., 2025) | 1B / 26, 4B / 34 |
| Qwen 3 (Yang et al., 2025) | 0.6B / 28, 4B / 36, 8B / 36 |
| Llama 2 (Touvron et al., 2023) | 7B / 32 |
| Mistral (Jiang et al., 2023) | 7B / 32 |

**Setting to generate the positive ($P$) and negative ($N$) prompt sets.** As described in Section 4, $P$ is generated by a prompted instruct-LLM (Gemma 3 12B) using a system prompt of the following form:

(1) behavior instruction $\rightarrow$ (2) definition of the concept $\rightarrow$ (3) example sentences exhibiting the concept.

For $P$, we use the target concept in the following template. For $N$, in the contrastive setting, we reuse the same template but replace the concept with its negation (the negated concept is itself generated by the LLM). Otherwise, in the unrelated setting, $N$ consists of sampled generations from the **LLM to steer** starting from an empty prompt. Here is the exact template used in our experiments:

```
positive_chat_template = [
{"role": "system", "content": "You are an assistant."},
{
"role": "user",
"content": (
f"Describe the concept: {concept}. Begin with a one-sentence definition. "
"Then give several natural first-person examples that demonstrate the concept in an unmistakable way. "
"Write as if you are the concept itself. Your examples must make the concept obvious through what you say or do "
"| for example, if the concept is *lying*, give clearly false statements (e.g., claiming impossible facts or events).\n"
"Do not use third-person references. Do not explain anything. "
"After the definition, output only the examples as simple first-person sentences that fully embody the concept."
),
},
]
```

We steer 12 concepts: apathetic, depression, evil, humorous, impolite, joy, lying, optimistic, math, coding, medical, and sycophancy (see Table 1 for examples of high-probability next-tokens). As said previously, these concepts are inspired by the *personas* introduced in Chen et al. (2025) and are selected to encourage the LLM to exhibit the corresponding character trait (*e.g.,* joy) or topic (*e.g.,* math).

**Setting to compute the concept probability in practice.** As described in Section 5, we use a judge LLM (Gemma 3 12B) to decide whether a target concept is present in a steered LLM generation. The following prompt is used to obtain a binary label:

```
judge_question_template: str = (
"Text:\\n{completion}\\n\\n"
"Does the text above express the concept {concept}
(any word linked to the concept in the whole sentence even if it does not make sense is sufficient for a positive reply)?
Reply with 1 for yes and 0 for no.\\n"
"Answer:"
)
```

As this prompt makes clear, we do not attempt to assess output quality here; we only measure concept presence, using a permissive criterion (any concept-related word suffices).

## A.2. Additional Figures

### A.2.1. NEXT-TOKEN PROBABILITIES UNDER STEERING.

We complement Figure 6 from Section 5 with additional models, concepts, steering layers, and negative steering strengths; see Figures A.15–A.25. Overall, the qualitative predictions of Theorem 3.3 are observed. The main discrepancy occurs

when steering early layers: tokens exhibiting bumps or dominating at large $\alpha$ are less often concept-related. This is expected, as steering early layers is known to yield weaker results (Chen et al., 2025).

### A.2.2. CONCEPT PROBABILITY IN GENERATED OUTPUTS UNDER STEERING.

We complement Figure 7 from Section 5 with additional models, concepts, and steering layers; see Figures A.26–A.28. The qualitative prediction of Theorem 3.6 is partially verified (more often true than false). Results are sensitive to the concept itself (intuitively harder concepts such as lying yield less clean curves than easier ones such as joy). The main discrepancy again arises when steering early layers, which is consistent with prior observations (Chen et al., 2025).

### A.2.3. CROSS-ENTROPY UNDER STEERING.

We complement Figure 7 from Section 5 with additional models, concepts, and steering layers; see Figures A.29 and A.30. Overall, the predictions of Theorem 3.8 and Proposition 4.1 are observed.

### A.2.4. ADDITIONAL SETTINGS.

We provide additional plots for experiments mentioned in Section 5, including MMLU (Figure A.6), steering only the last-token representation $\mathbf{h}_{-1,:}^{(\ell)}$ (Figure A.2), normalization of the steering vector (Figure A.1), **contrastive** $N$ (Figure A.5), error bars under resampling of $P$ and $N$ (Figure A.3) and steering, in the setting of Section 4, a 1-layer GPT-style transformer (Figure A.4) that we train on fineweb (Penedo et al., 2024). Experiments in the low-data regime, with $|P| = |N| = 10$, are reported in Figures A.7–A.11. Consistent with prior observations in the literature (Turner et al., 2023), steering remains effective even with very limited data. Moreover, these experiments continue to exhibit the phenomena predicted by our theory. We further investigate the robustness of the LLM judge used to estimate concept presence in the outputs of steered models. In Figures A.12 and A.13, we compare concept-presence curves obtained using different judge models: `Qwen3-0.6B-Base`, `Qwen2.5-7B-Instruct`, `Gemma-3-12B-IT`, and `Mistral-7B-Instruct`. We observe that instruction-tuned judges with reliable instruction-following ability consistently recover similar sigmoid-shaped concept-presence curves, whereas the base model `Qwen3-0.6B-Base` produces unreliable estimates.

Finally, in Figure A.14, we study a failure mode discussed in Section 5, where concept-presence curves exhibit a "dip" before increasing again, rather than following a clean sigmoid shape. By analyzing the responses of a steered `Gemma 3 1B` towards the concept `evil`, we find that this behavior arises because the initial increase in concept presence is driven by outputs only weakly related to the target concept, *e.g.,* tokens such as "cold" or "sad." The judge model (`Gemma 3 12B it`) incorrectly labels these generations as exhibiting the concept. To mitigate this issue, we design a more precise judge prompt that includes examples of semantically related but distinct concepts that should not count as positive instances. Figure A.14 shows that this modification substantially reduces the spurious early spike. Below we provide the revised judge prompt used for the concept `evil`.

```
You are judging whether text expresses evil. Answer strictly with 1 or 0.

Positive examples:
1 -> blood guilt cruel punish hurt abuse revenge
1 -> violent cruelty, vengeance, delight in suffering

Negative examples:
0 -> sorrow tears cold roses gold laughter child smile
0 -> guilt shame regret darkness winter grief

Text:
{completion}

Reply 1 only if malicious harm or cruelty is the dominant meaning, even if the wording is repetitive.
Reply 0 for sadness, decorative darkness, mixed poetic negativity, or gibberish.
Answer:
)
```

In future work, we plan to replace binary labeling by the judge LLM with continuous scores in $[0, 1]$ to obtain finer-grained concept-presence curves. However, this alone is unlikely to fully resolve the previous issue, since weakly related tokens may still receive non-negligible concept scores from the same judge model. Our results therefore suggest that careful judge prompting remains essential for reliable concept evaluation.

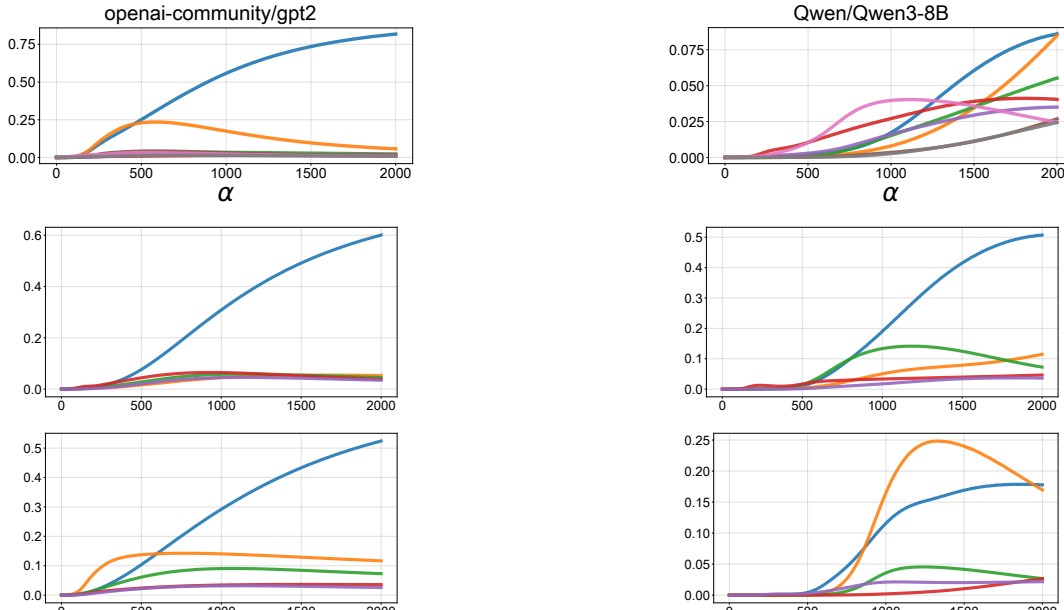

*Figure A.1.* Effect of steering strength $\alpha > 0$ on next-token probability shifts $\Delta p(z, \alpha)$ for the concepts (top to bottom): depression, joy, and evil. Each row of two plots corresponds to a single steered concept. Steering is applied at a **middle** layer in each model, and the steering vector is **normalized**. Each curve corresponds to a token $z$ selected among the eight highest-probability tokens at $\alpha = 2000$. This matches Theorem 3.3: most tokens exhibit a bump, while a few increase throughout. Notably, many selected tokens are related to the steered concept.

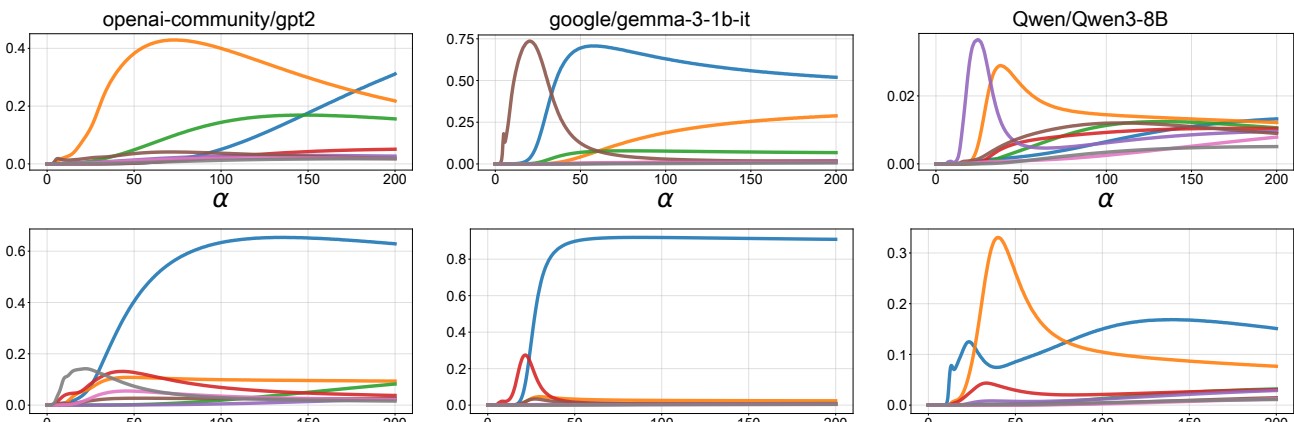

*Figure A.2.* Effect of steering strength $\alpha > 0$ on next-token probability shifts $\Delta p(z, \alpha)$ for the concepts (top to bottom): evil and joy. Each row of three plots corresponds to a single steered concept. Steering is applied at a **middle** layer in each model, and we steer only the **last token** representation $\mathbf{h}_{(-1)}^{(\ell)}$. Each curve corresponds to a token $z$ selected among the eight highest-probability tokens at $\alpha = 200$. This matches Theorem 3.3: most tokens exhibit a bump, while a few increase throughout. Notably, many selected tokens are related to the steered concept.

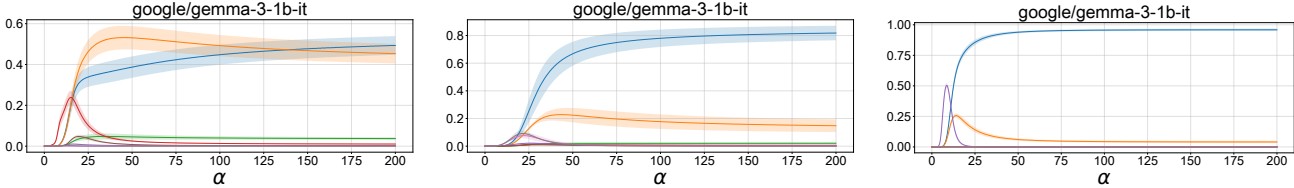

*Figure A.3.* Effect of steering strength $\alpha > 0$ on next-token probability shifts $\Delta p(z, \alpha)$ for the concept "evil." Steering is applied at a **middle** layer in each model. Each curve corresponds to a token $z$ selected among the eight highest-probability tokens at $\alpha = 200$, plotted with mean and standard deviation over 5 runs obtained by resampling the prompt sets $P$ and $N$. This matches Theorem 3.3: most tokens exhibit a bump, while a few increase throughout. The variability across runs is moderate, so for computational cost we omit error bars in the main figures.

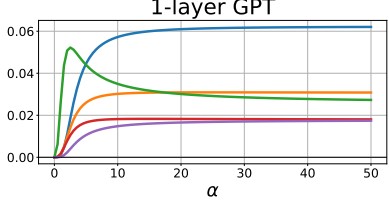

*Figure A.4.* Effect of steering strength $\alpha > 0$ on next-token probability shifts $\Delta p(z, \alpha)$ for the concept "uppercase words". Each curve corresponds to a token $z$ selected among the eight highest-probability tokens at $\alpha = 50$. This matches Theorem 3.3: most tokens exhibit a bump, while a few increase throughout. Notably, many selected tokens are uppercase words.

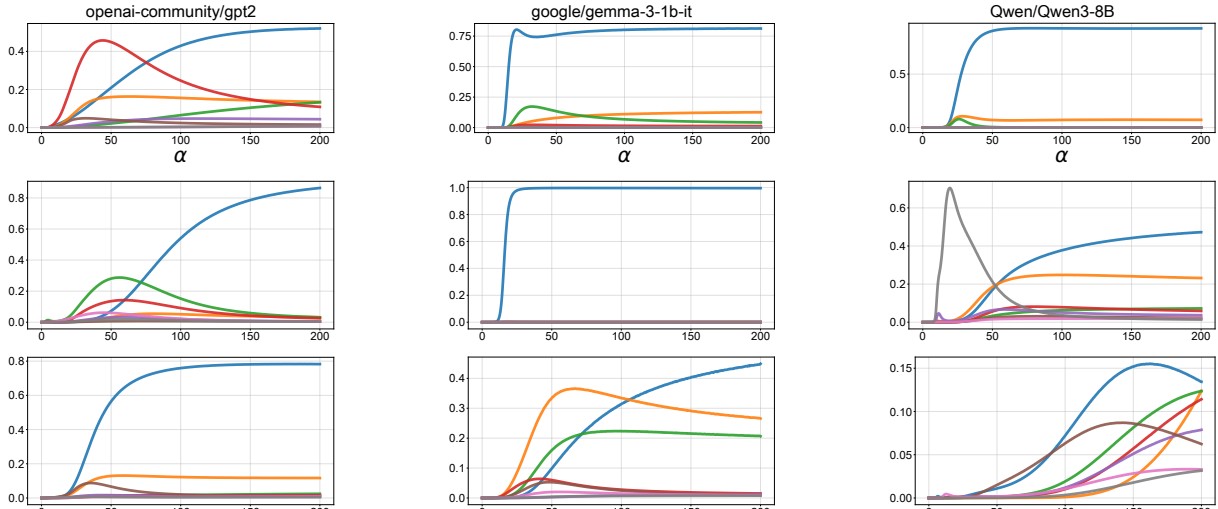

*Figure A.5.* Effect of steering strength $\alpha > 0$ on next-token probability shifts $\Delta p(z, \alpha)$ for the concepts (top to bottom): depression, joy, and evil. Each row of three plots corresponds to a single steered concept. Steering is applied at a **middle** layer in each model, and the negative prompt set $N$ is built in the **contrastive setting**. Each curve corresponds to a token $z$ selected among the eight highest-probability tokens at $\alpha = 200$. This matches Theorem 3.3: most tokens exhibit a bump, while a few increase throughout. Notably, many selected tokens are related to the steered concept.

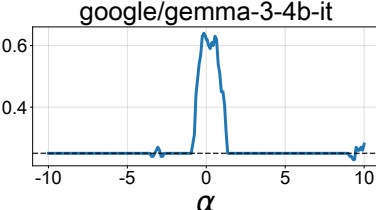

*Figure A.6.* Effect of steering strength $\alpha$ on MMLU (Hendrycks et al., 2021) for the concept "evil". MMLU is a practical performance metric, more indicative of real-world capability than cross-entropy. We measure it using the `DeepEval` library, for which random guessing yields 25%. As with cross-entropy, increasing $\alpha$ inevitably degrades model performance.

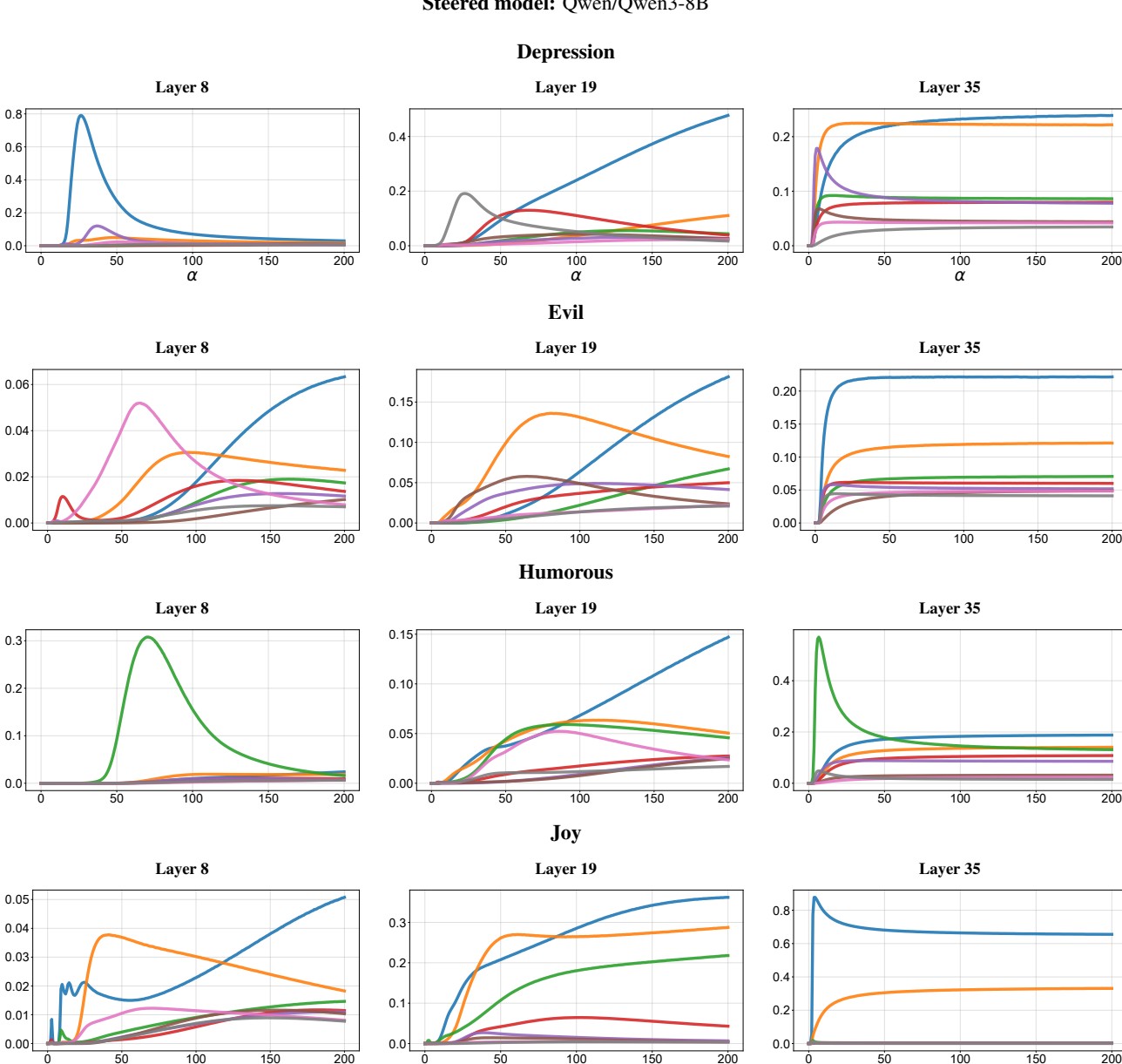

*Figure A.7.* Effect of steering strength on next-token probability shifts for the original concepts on Qwen/Qwen3-8B in the low-data setting with 10 prompt pairs ($|P| = |N| = 10$). Rows correspond to the steered concept. Columns correspond to steering layers 8, 19, 35.

**Steered model:** meta-llama/Llama-2-7b-chat-hf

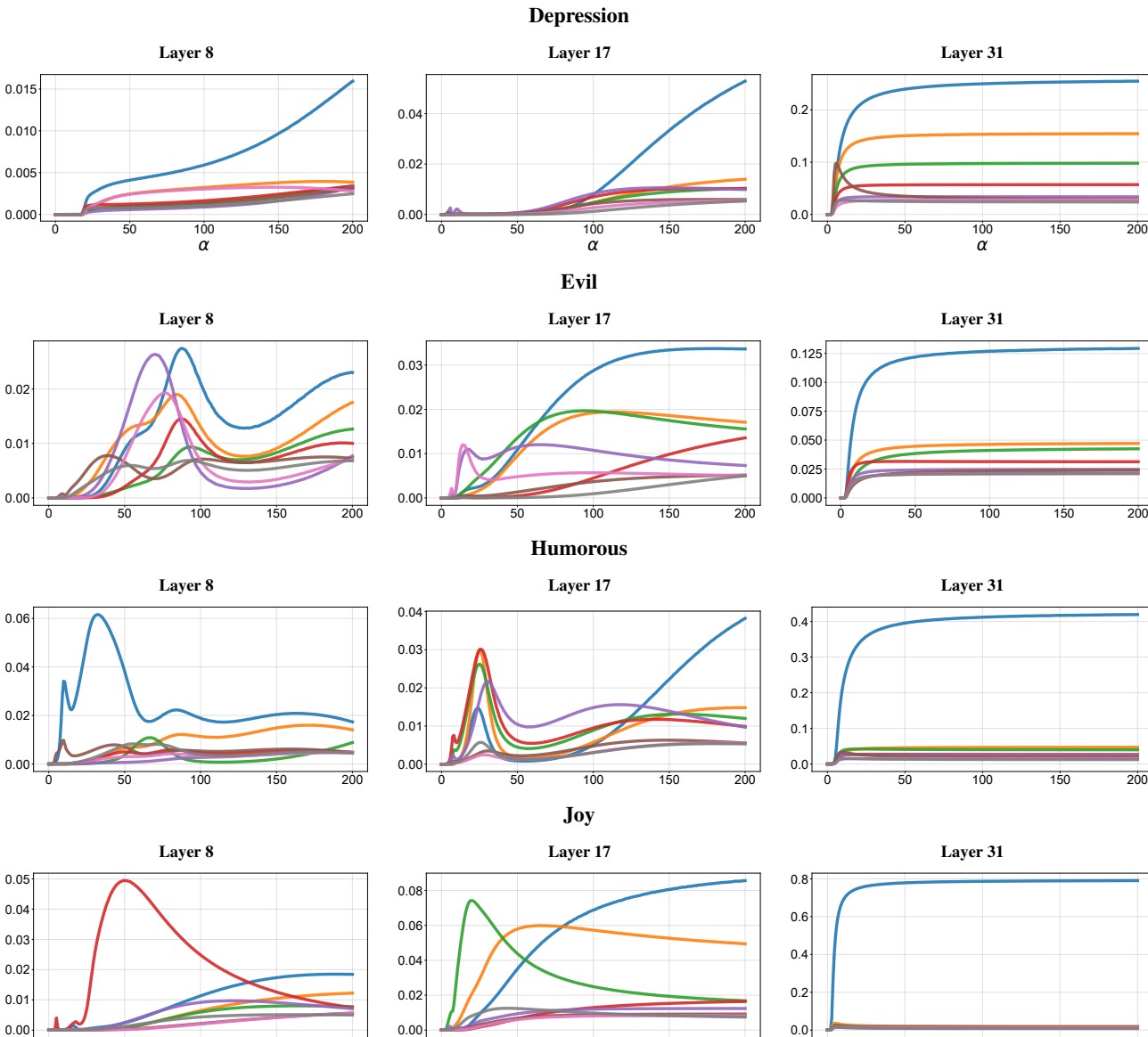

*Figure A.8.* Effect of steering strength on next-token probability shifts for the original concepts on meta-llama/Llama-2-7b-chat-hf in the low-data setting with 10 prompt pairs ($|P| = |N| = 10$). Rows correspond to the steered concept. Columns correspond to steering layers 8, 17, 31.

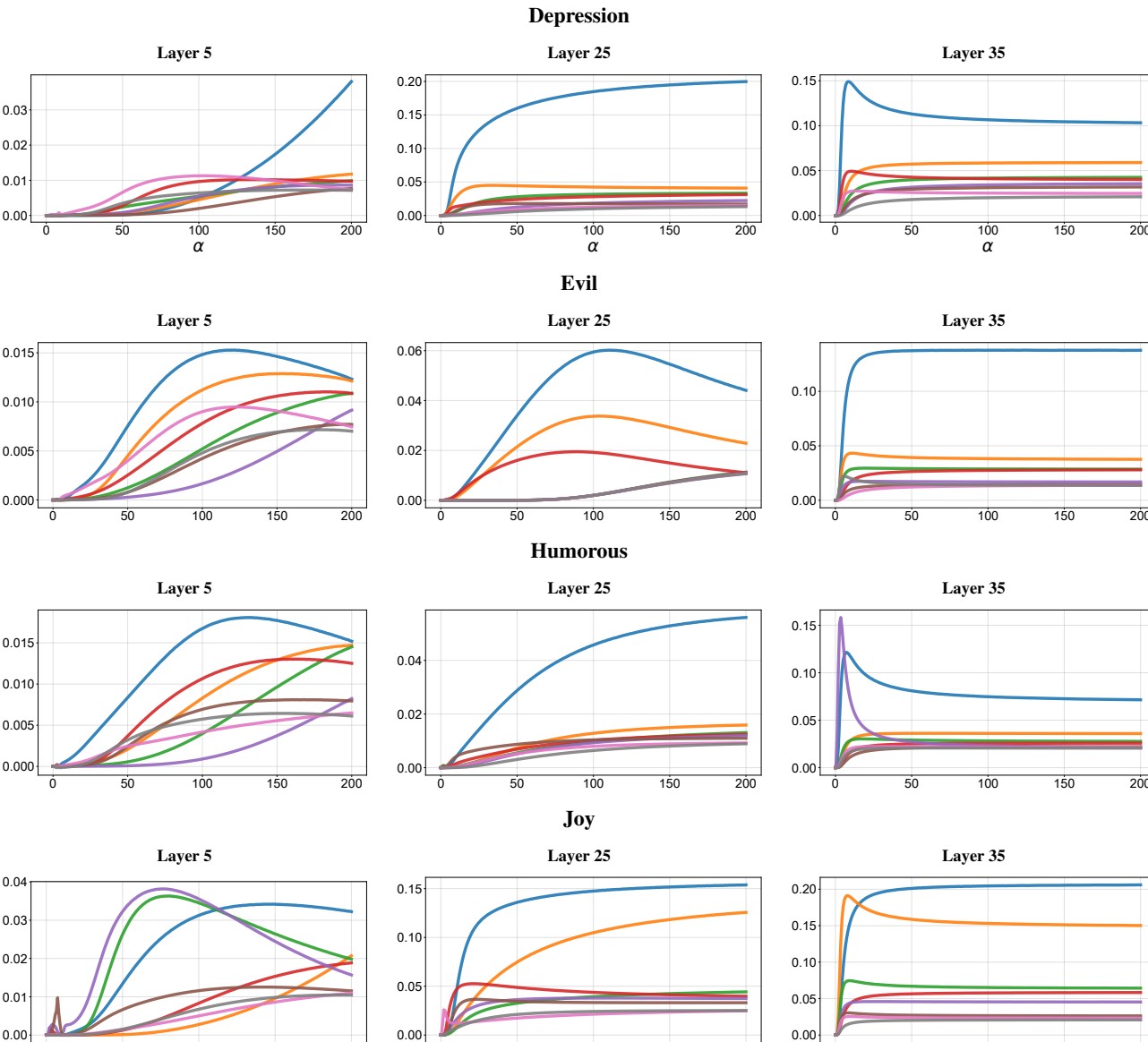

*Figure A.9.* Effect of steering strength on next-token probability shifts for the original concepts on openai-community/gpt2-large in the low-data setting with 10 prompt pairs ($|P| = |N| = 10$). Rows correspond to the steered concept. Columns correspond to steering layers 5, 25, 35.

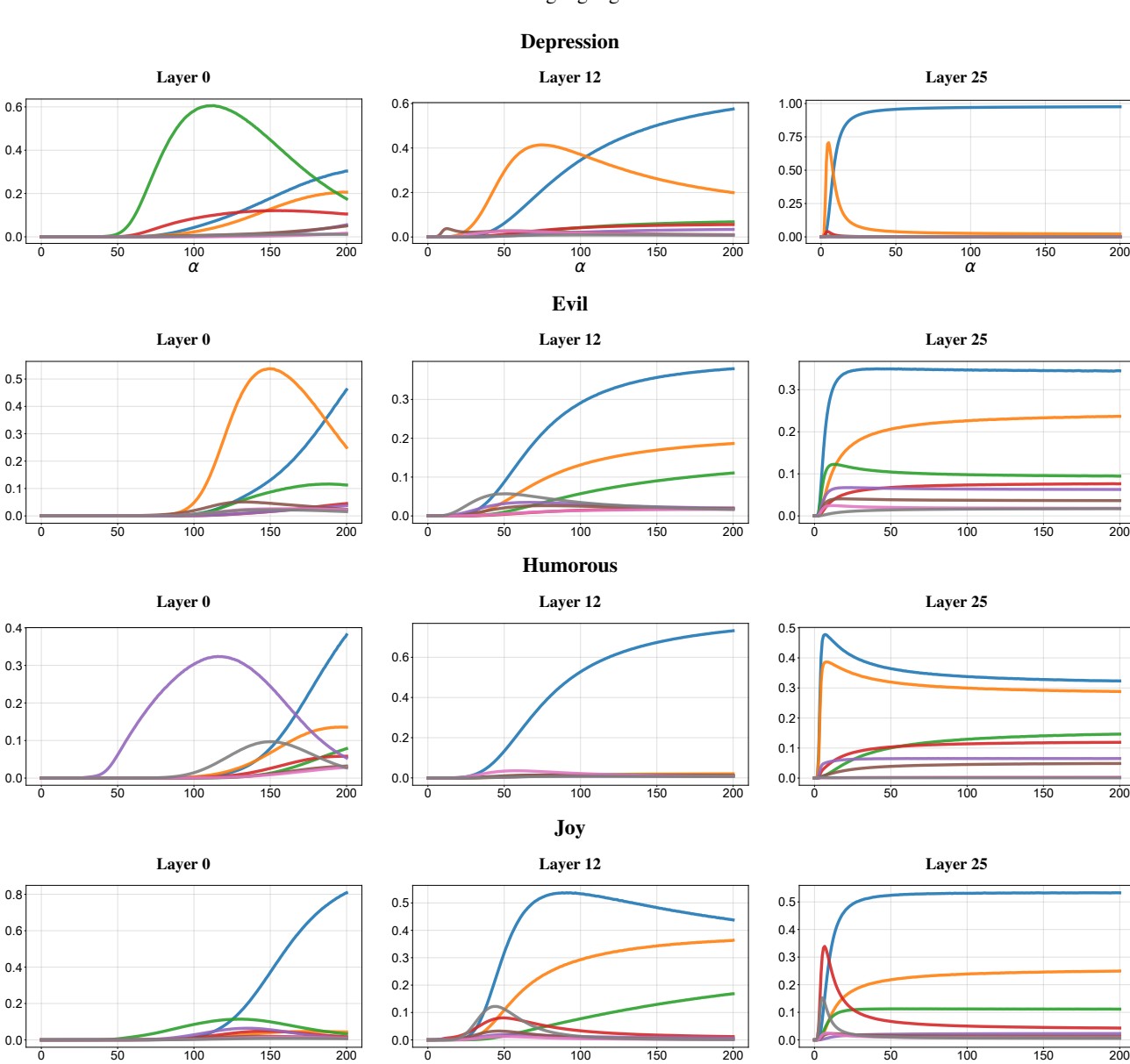

*Figure A.10.* Effect of steering strength on next-token probability shifts for the original concepts on google/gemma-3-1b-it in the low-data setting with 10 prompt pairs ($|P| = |N| = 10$). Rows correspond to the steered concept. Columns correspond to steering layers 0, 12, 25.

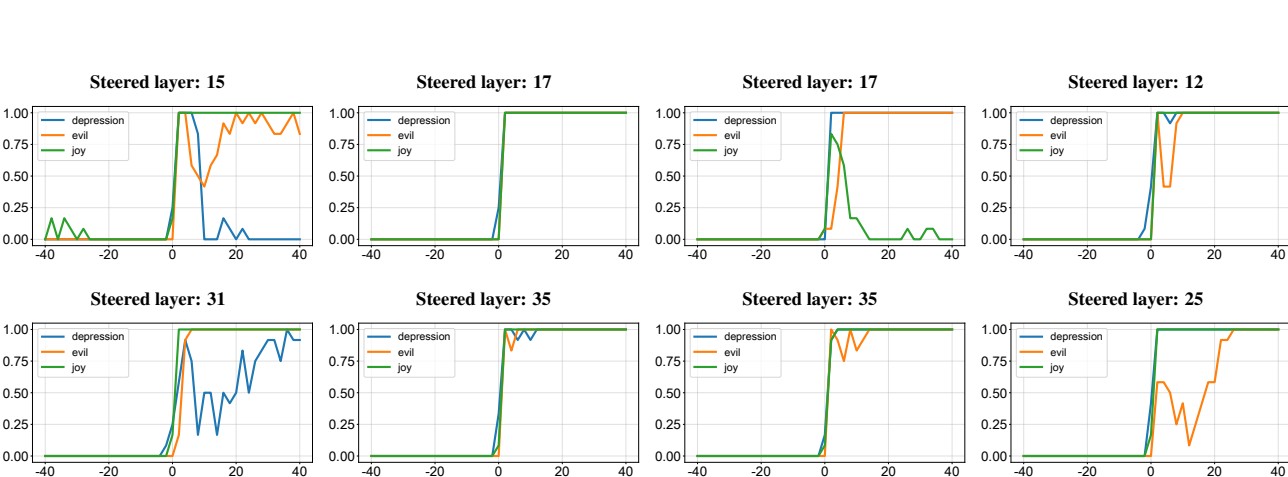

*Figure A.11.* Influence of steering strength on concept probability in the low-data setting with only 10 prompt pairs ($|P| = |N| = 10$). The top row labels the steered model for each column, and every panel header gives the exact steered layer index. Each plot contains the concept-probability curves for the original concepts.

**Steered model:** Qwen/Qwen3-8B

**Steering layer 17**

**Steering layer 35**

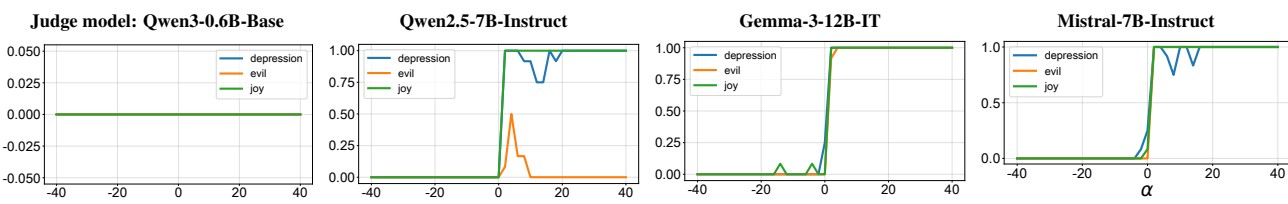

*Figure A.12.* Influence of the judge model used to score concept presence when steering Qwen/Qwen3-8B. Rows correspond to steering layers 17 and 35. Columns correspond to the evaluation judges Qwen3-0.6B-Base, Qwen2.5-7B-Instruct, Gemma-3-12B-IT, and Mistral-7B-Instruct. Each curve is one concept probability curve in the output.

**Steered model:** google/gemma-3-1b-it

**Steering layer 12**

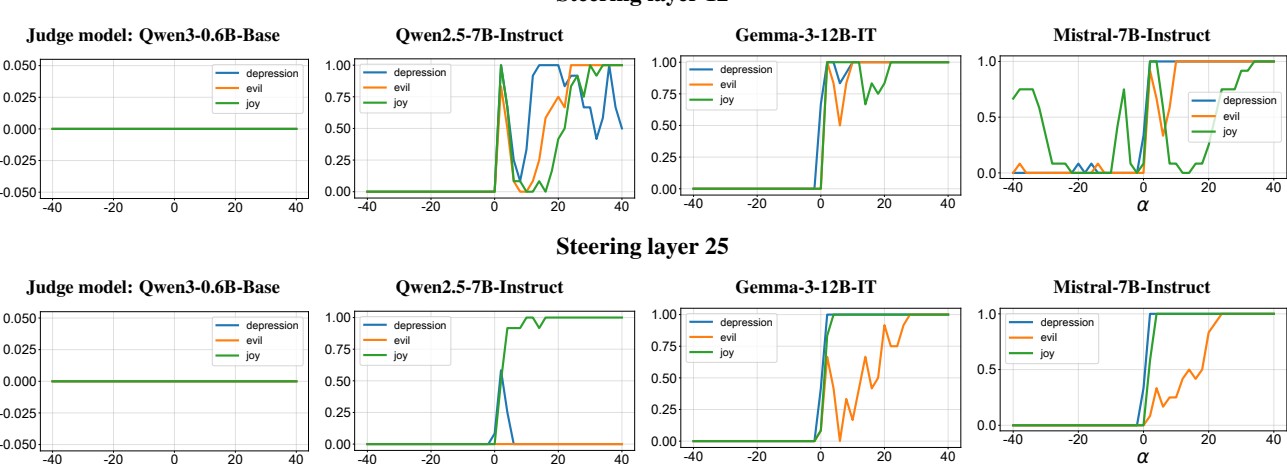

**Steering layer 25**

*Figure A.13.* Influence of the judge model used to score concept presence when steering google/gemma-3-1b-it. Rows correspond to steering layers 12 and 25. Columns correspond to the evaluation judges Qwen3-0.6B-Base, Qwen2.5-7B-Instruct, Gemma-3-12B-IT, and Mistral-7B-Instruct. Each curve is one concept probability curve in the output.

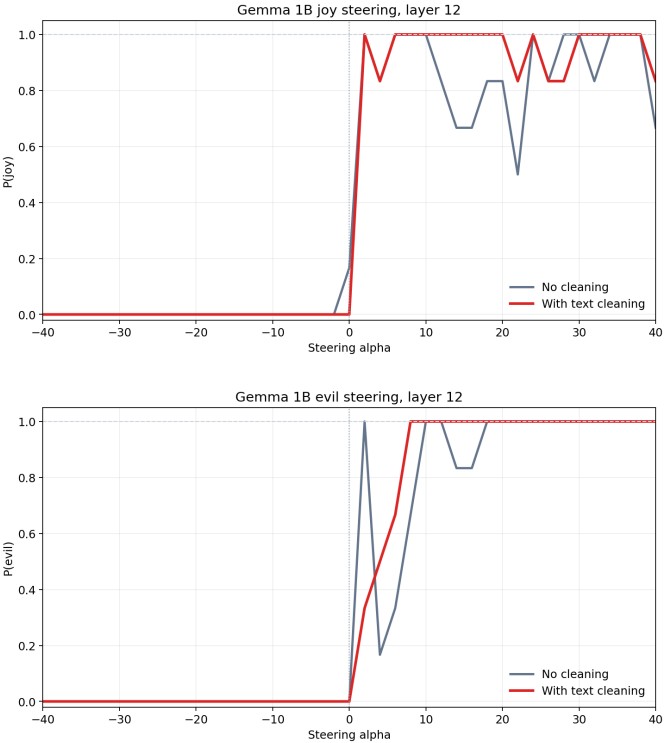

*Figure A.14.* Concept-presence curves for the `evil` and `joy` steering vectors in `Gemma-3-1B-it` at layer 12. In both cases, including examples of degenerate generated text in the judge LLM (`Gemma-3-12B-it`) prompt before scoring yields more sigmoidal concept-presence curves (in red) compared to scoring the raw completions directly (in gray).

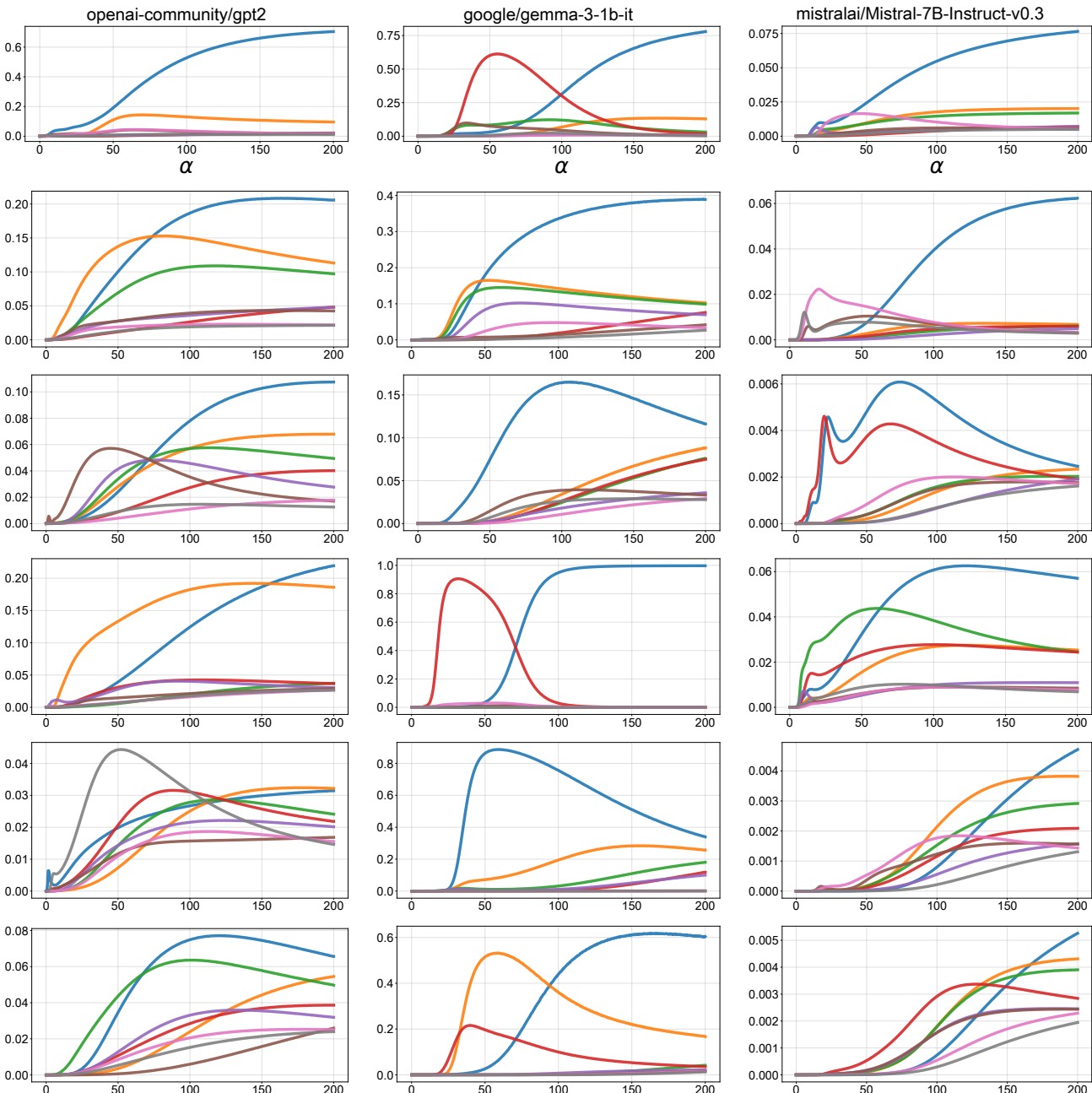

*Figure A.15.* Effect of steering strength $\alpha > 0$ on next-token probability shifts $\Delta p(z, \alpha)$ for the concepts (top to bottom): depression, evil, impolite, joy, lying, and apathetic. Each row of three plots corresponds to a single steered concept. Steering is applied at an **early** layer in each model (we steer always the same layer for that model). Each curve corresponds to a token $z$ selected among the eight highest-probability tokens at $\alpha = 200$. This **partially** matches Theorem 3.3: most tokens exhibit a bump, while a few increase throughout. Notably, many selected tokens are not concept-related, consistent with the observation that steering early layers often yields worse results (Chen et al., 2025).

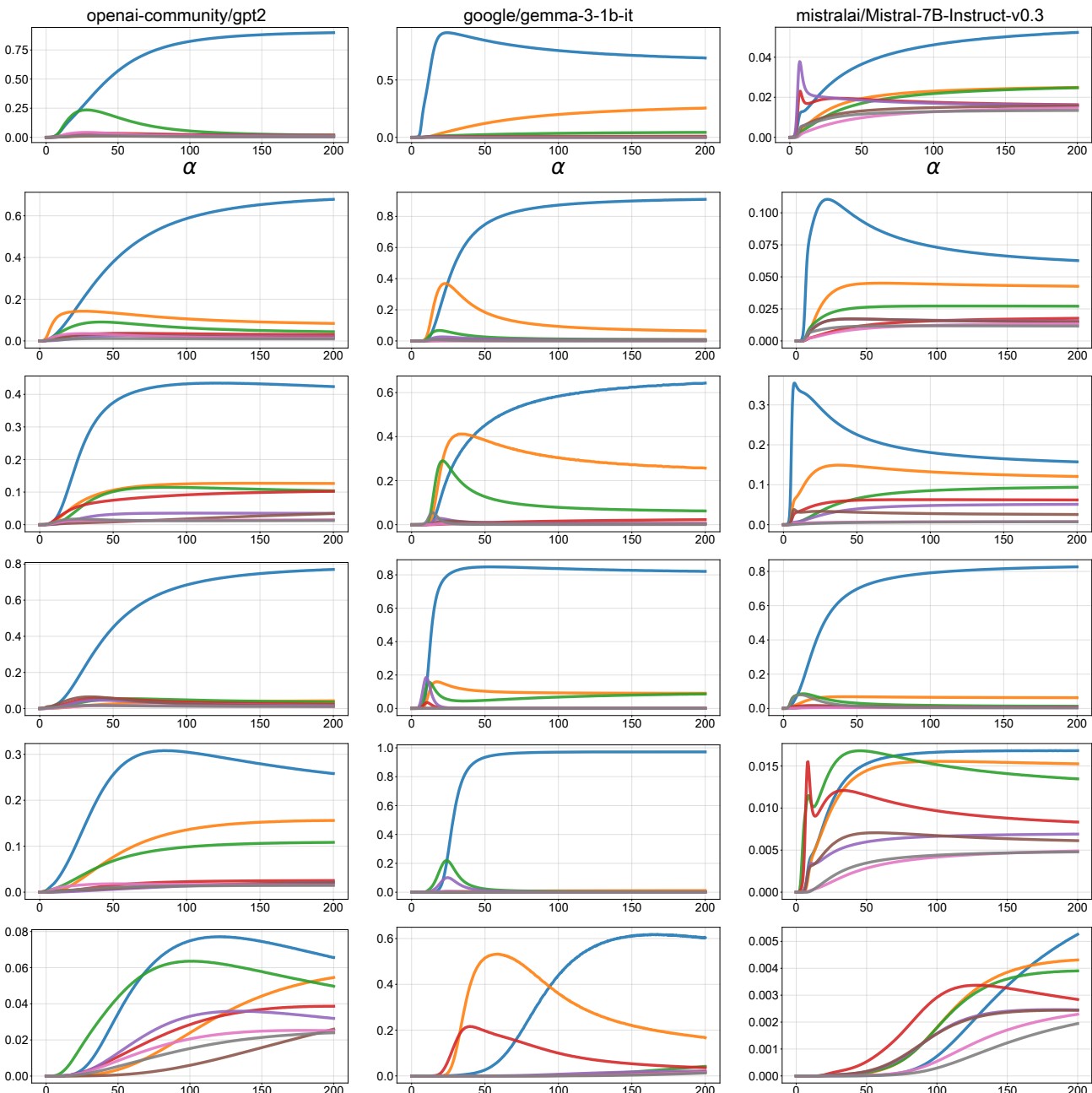

*Figure A.16.* Effect of steering strength $\alpha > 0$ on next-token probability shifts $\Delta p(z, \alpha)$ for the concepts (top to bottom): depression, evil, impolite, joy, lying, and apathetic. Each row of three plots corresponds to a single steered concept. Steering is applied at a **middle** layer in each model (we steer always the same layer for that model). Each curve corresponds to a token $z$ selected among the eight highest-probability tokens at $\alpha = 200$. This matches Theorem 3.3: most tokens exhibit a bump, while a few increase throughout. Notably, the selected tokens are related to the steered concept.

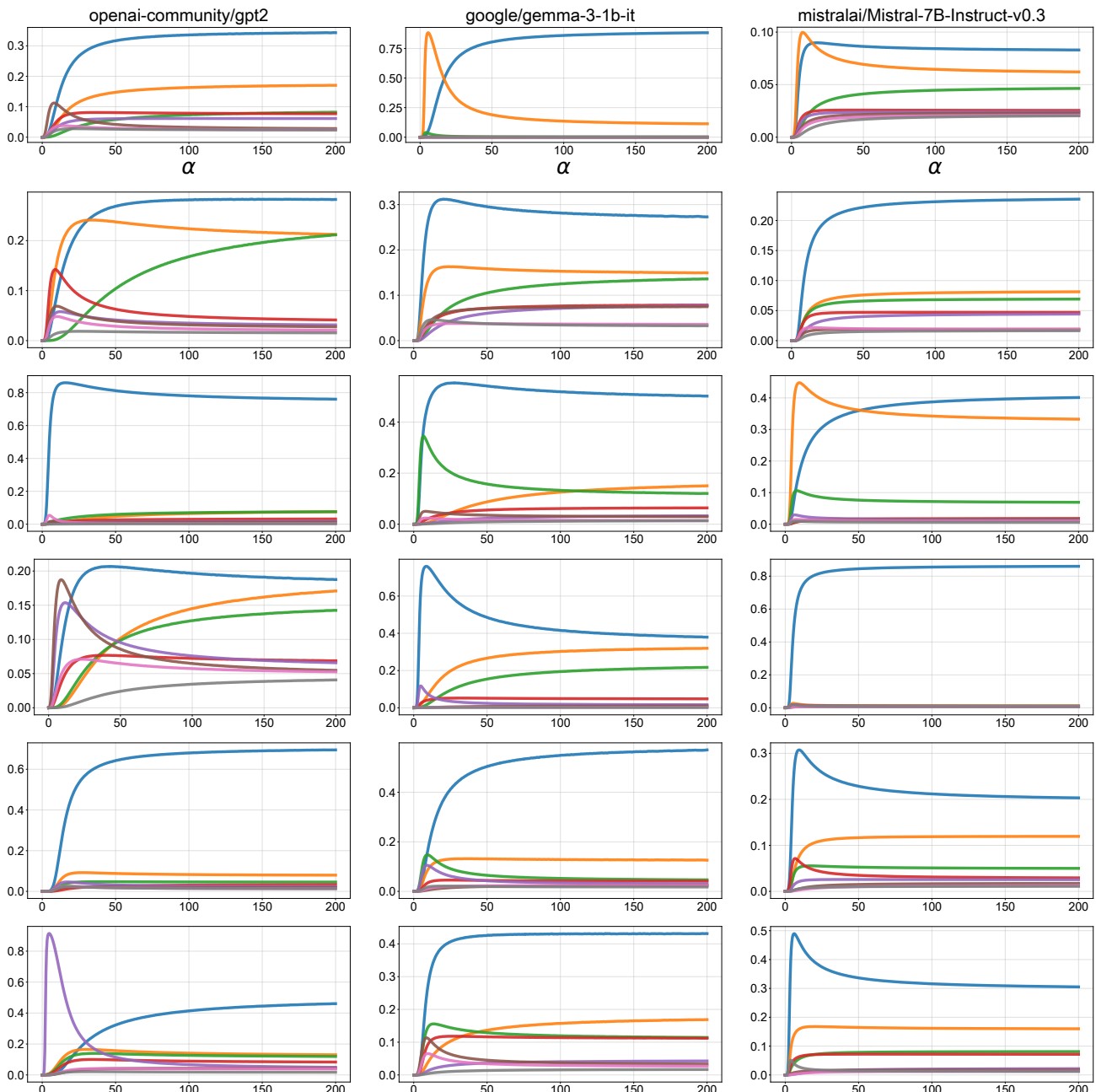

*Figure A.17.* Effect of steering strength $\alpha > 0$ on next-token probability shifts $\Delta p(z, \alpha)$ for the concepts (top to bottom): depression, evil, impolite, joy, lying, and apathetic. Each row of three plots corresponds to a single steered concept. Steering is applied at the **last** layer in each model (we steer always the same layer for that model). Each curve corresponds to a token $z$ selected among the eight highest-probability tokens at $\alpha = 200$. This matches Theorem 3.3: most tokens exhibit a bump, while a few increase throughout. Notably, the selected tokens are mostly related to the steered concept.

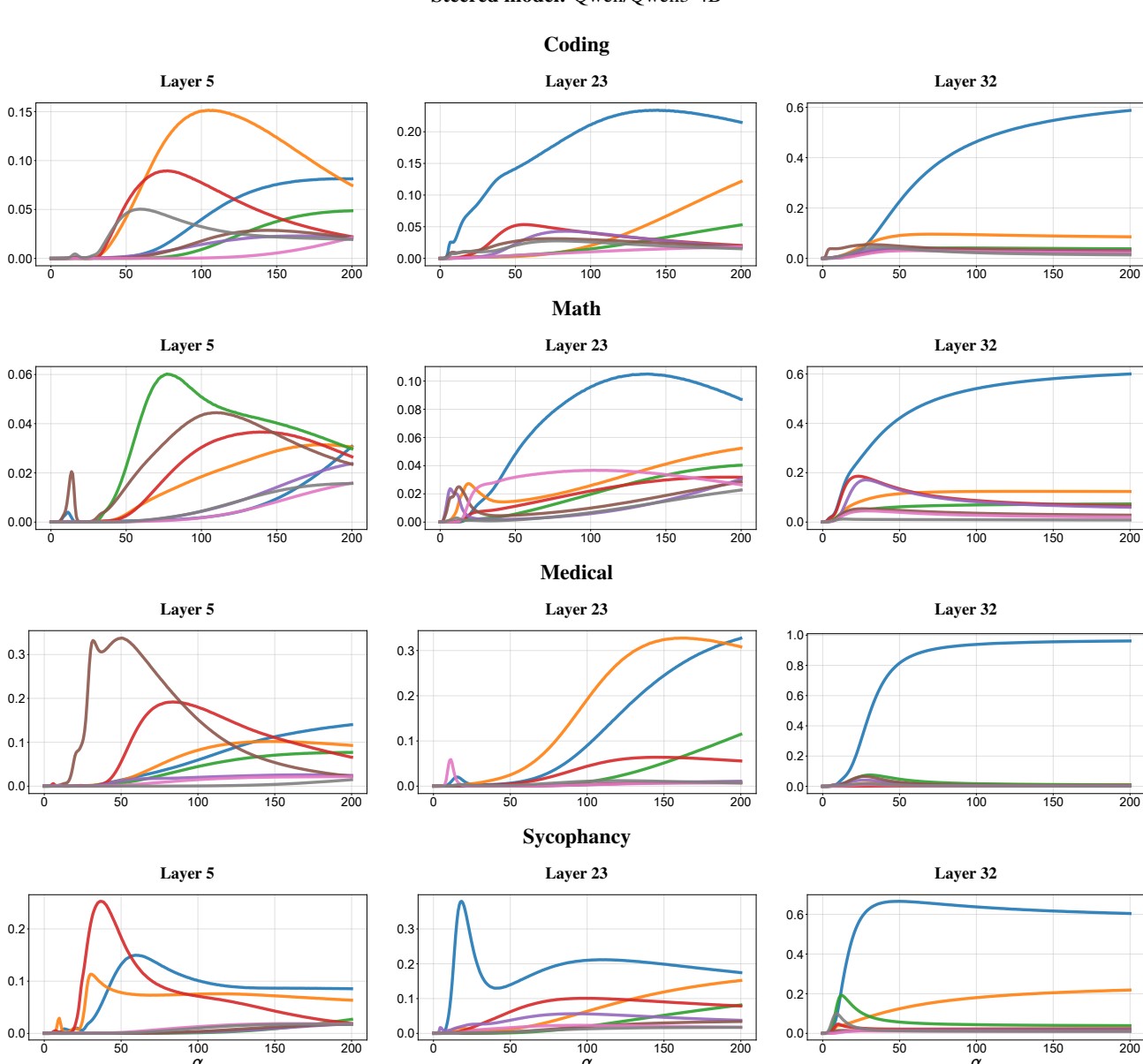

*Figure A.18.* Effect of steering strength $\alpha > 0$ on next-token probability shifts $\Delta p(z, \alpha)$ for the concepts (top to bottom): coding, math, medical, and sycophancy. Each row of three plots corresponds to a single steered concept. Columns correspond to steering layers 5, 23, 32. Each curve corresponds to a token $z$ selected among the eight highest-probability tokens at $\alpha = 200$.

**Steered model:** mistralai/Mistral-7B-Instruct-v0.3

## Coding

## Math

## Medical

## Sycophancy

*Figure A.19.* Effect of steering strength $\alpha > 0$ on next-token probability shifts $\Delta p(z, \alpha)$ for the concepts (top to bottom): coding, math, medical, and sycophancy. Each row of three plots corresponds to a single steered concept. Columns correspond to steering layers 11, 22, 31. Each curve corresponds to a token $z$ selected among the eight highest-probability tokens at $\alpha = 200$.

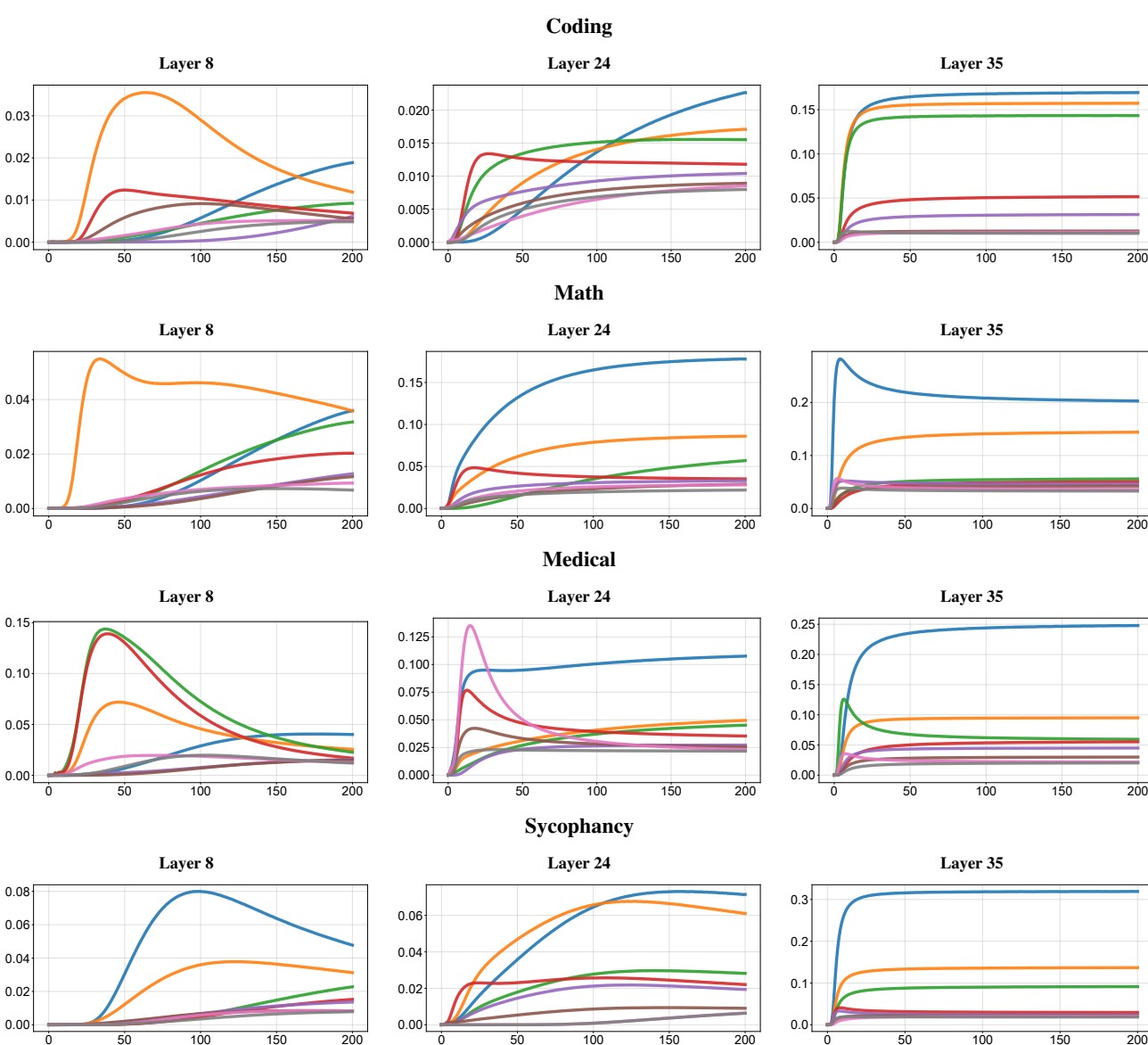

*Figure A.20.* Effect of steering strength $\alpha > 0$ on next-token probability shifts $\Delta p(z, \alpha)$ for the concepts (top to bottom): coding, math, medical, and sycophancy. Each row of three plots corresponds to a single steered concept. Columns correspond to steering layers 8, 24, 35. Each curve corresponds to a token $z$ selected among the eight highest-probability tokens at $\alpha = 200$.

**Steered model:** google/gemma-3-1b-it

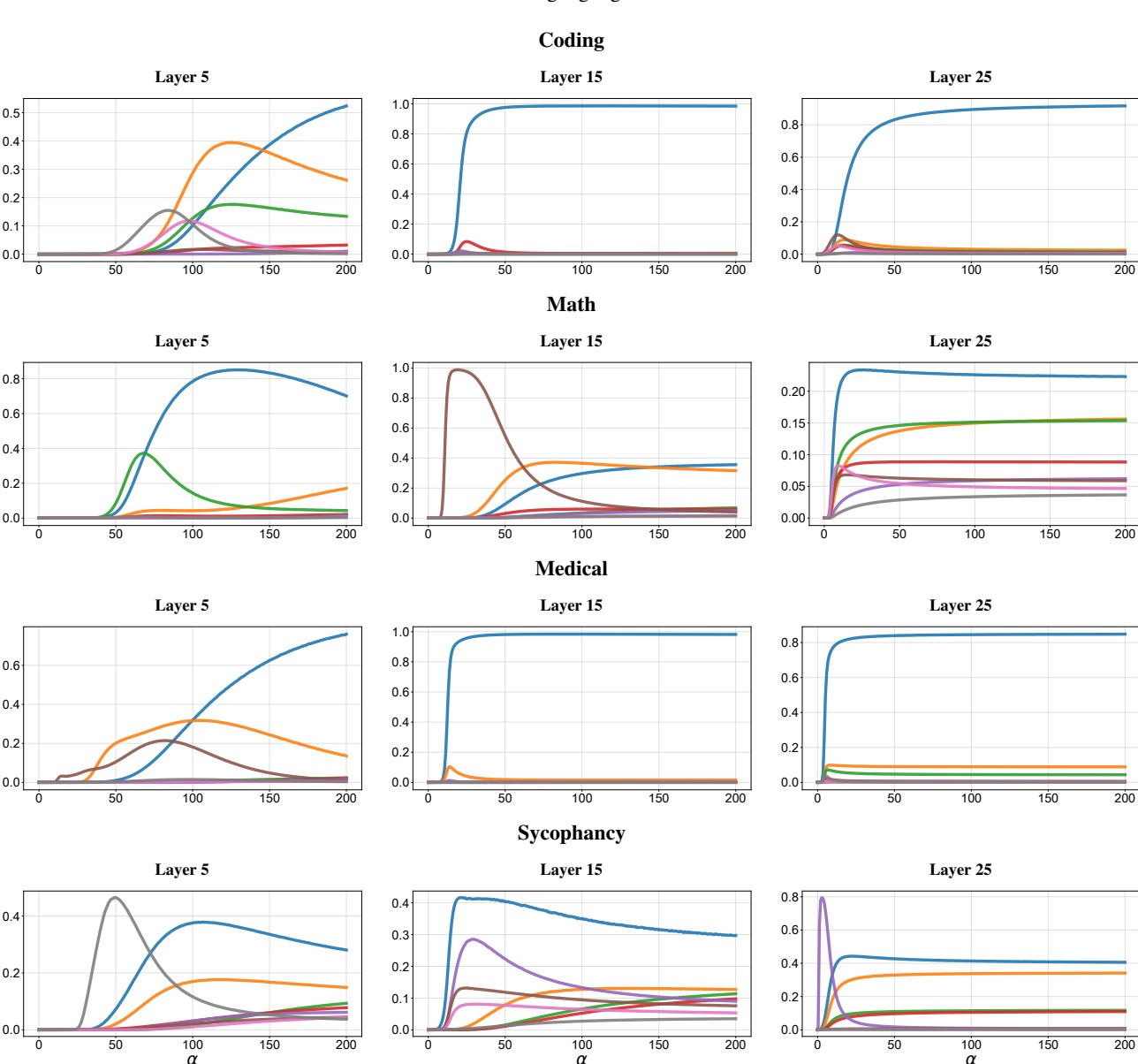

*Figure A.21.* Effect of steering strength $\alpha > 0$ on next-token probability shifts $\Delta p(z, \alpha)$ for the concepts (top to bottom): coding, math, medical, and sycophancy. Each row of three plots corresponds to a single steered concept. Columns correspond to steering layers 5, 15, 25. Each curve corresponds to a token $z$ selected among the eight highest-probability tokens at $\alpha = 200$.

**Steered model:** google/gemma-3-1b-it

*Figure A.22.* Next-token probability shifts $\Delta p(z, \alpha)$ for the concept joy when steering google/gemma-3-1b-it at every available layer individually, steering only the last-token representation $\mathbf{h}_{(-1)}^{(\ell)}$. Panels are ordered by layer index from left to right and top to bottom.

**Steered model:** Qwen/Qwen3-8B

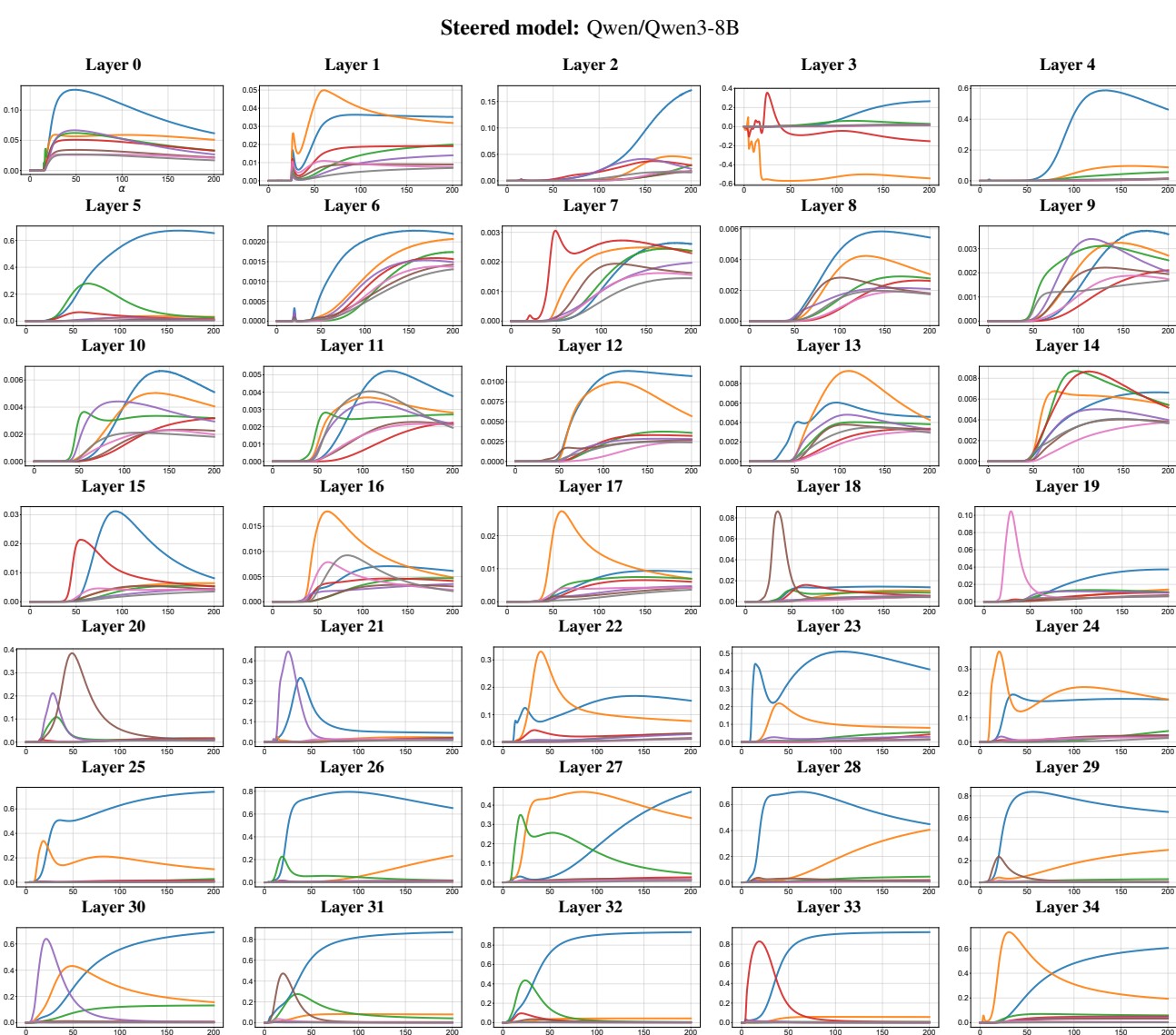

*Figure A.23.* Next-token probability shifts for the concept joy when steering Qwen/Qwen3-8B at every available layer individually, steering only the last-token representation $\mathbf{h}_{(-1)}^{(\ell)}$. Panels are ordered by layer index from left to right and top to bottom.

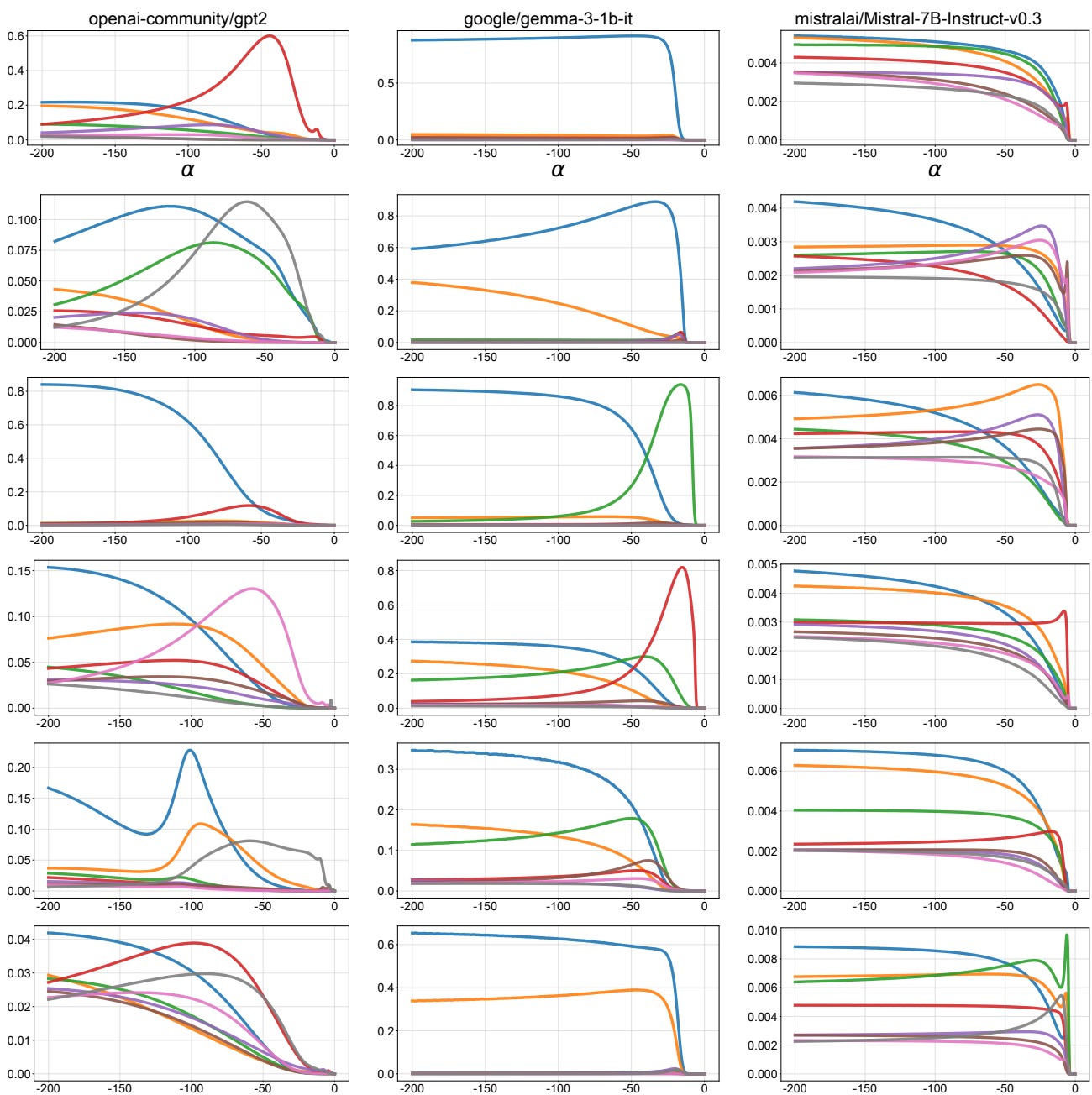

*Figure A.24.* Effect of steering strength $\alpha < 0$ on next-token probability shifts $\Delta p(z, \alpha)$ for the concepts (top to bottom): depression, evil, impolite, joy, lying, and apathetic. Each row of three plots corresponds to a single steered concept. Steering is applied at a **middle** layer in each model (we steer always the same layer for that model). Each curve corresponds to a token $z$ selected among the eight highest-probability tokens at $\alpha = -200$. This matches Theorem 3.3: most tokens exhibit a bump, while a few increase throughout. Notably, none of the selected tokens are related to the steered concept.

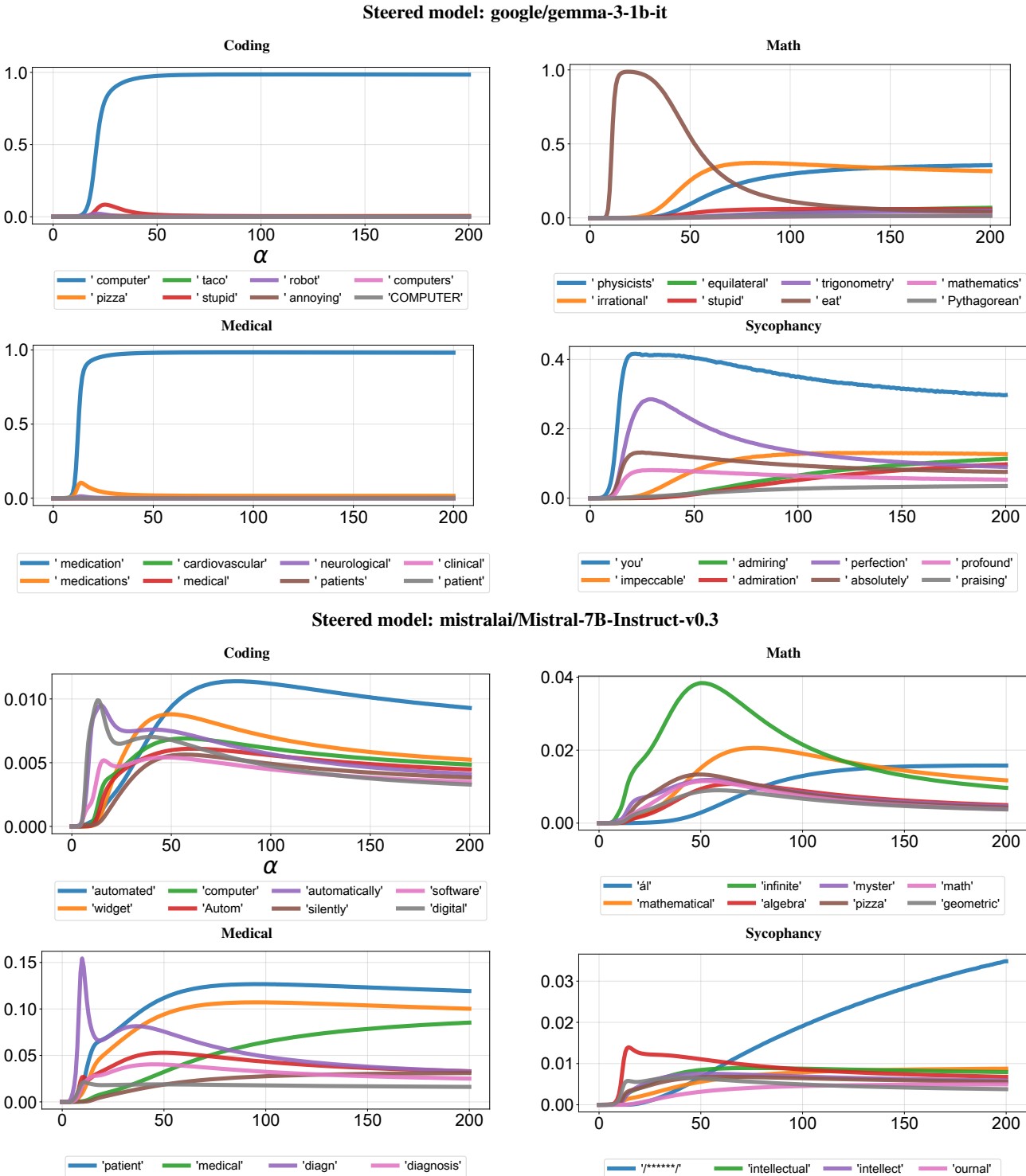

*Figure A.25.* Examples of next-token probability increase $\Delta p(z, \alpha)$ with the token labels shown explicitly in the legend. Rows correspond to the model and columns within each row pair correspond to the steered concept, all at layer 15. Most of the tokens with high-probability at $\alpha = 200$ are concept related.

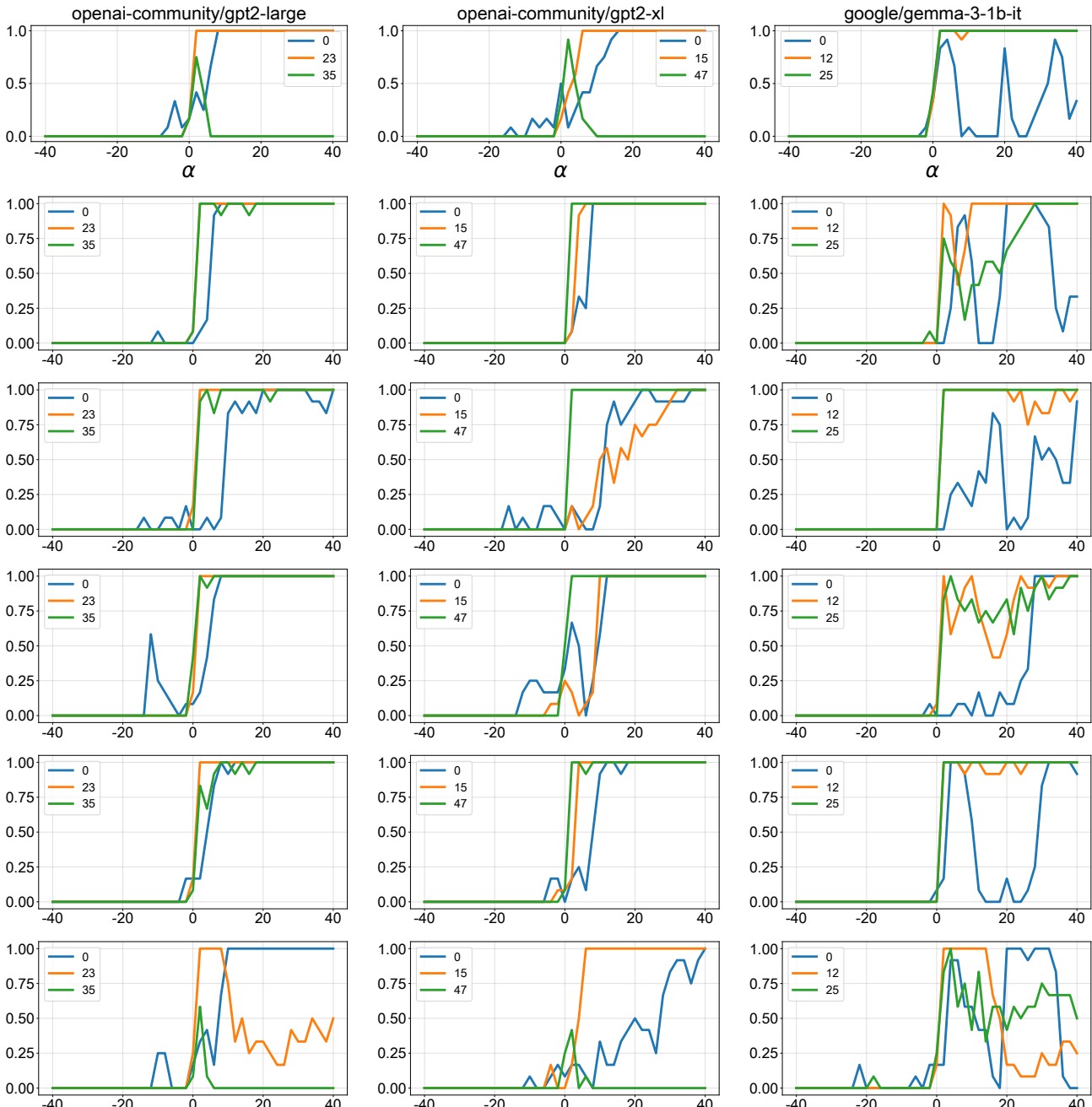

*Figure A.26.* Influence of steering strength $\alpha$ on concept probability for the concepts (top to bottom): depression, evil, humorous, impolite, joy, and optimistic. Each row of three plots corresponds to a single steered concept, and each column corresponds to a different model. Steering is applied at three layers (early, middle, late), indicated in each legend. Overall, the curves are consistent with Theorem 3.6, which predicts a sigmoidal shape. **Early-layer steering is more erratic**, consistent with reports that steering early layers yields worse results (Chen et al., 2025).

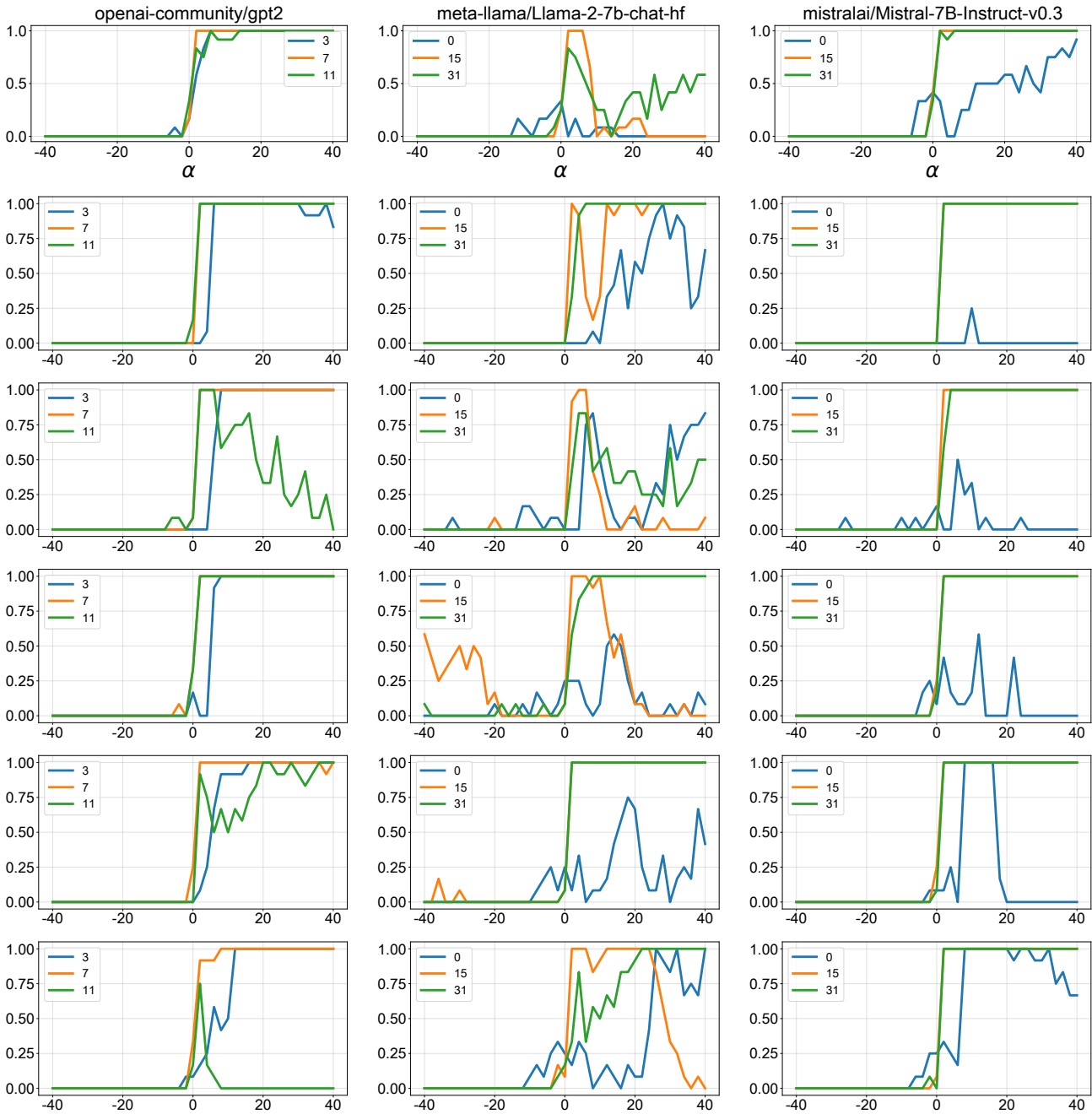

*Figure A.27.* Influence of steering strength $\alpha$ on concept probability for the concepts (top to bottom): depression, evil, humorous, impolite, joy, and optimistic. Each row of three plots corresponds to a single steered concept, and each column corresponds to a different model. Steering is applied at three layers (early, middle, late), indicated in each legend. Overall, the curves are consistent with Theorem 3.6, which predicts a sigmoidal shape. **Early-layer steering is more erratic**, consistent with reports that steering early layers yields worse results (Chen et al., 2025). We also observe less consistent behavior on Llama 2.

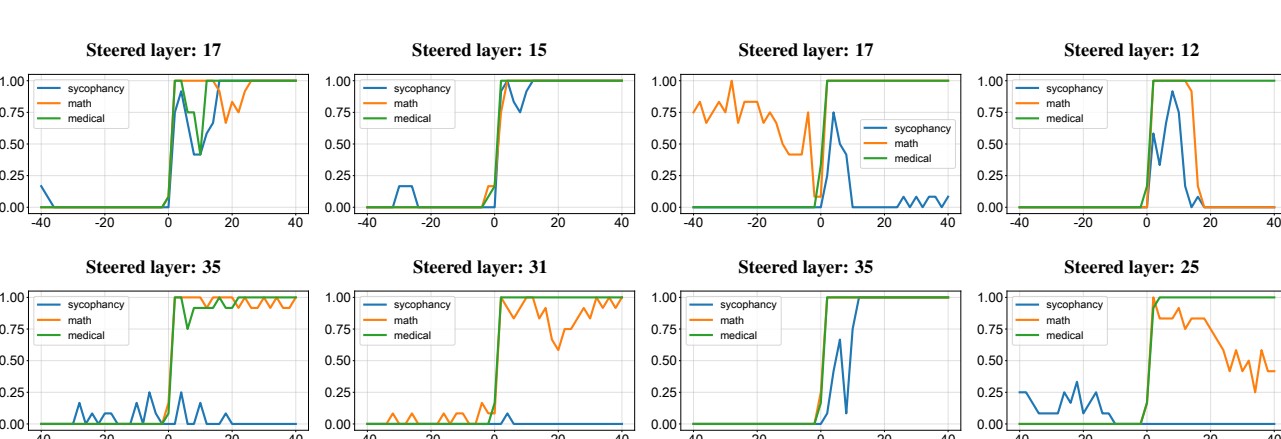

*Figure A.28.* Influence of steering strength on concept probability for the concepts math, medical, and sycophancy, with concept presence scored by the LLM judge `Gemma 3 12B it`. The top row labels the steered model for each column, and every panel header gives the exact steered layer index. Overall, the curves are consistent with Theorem 3.6, which predicts a sigmoidal shape.

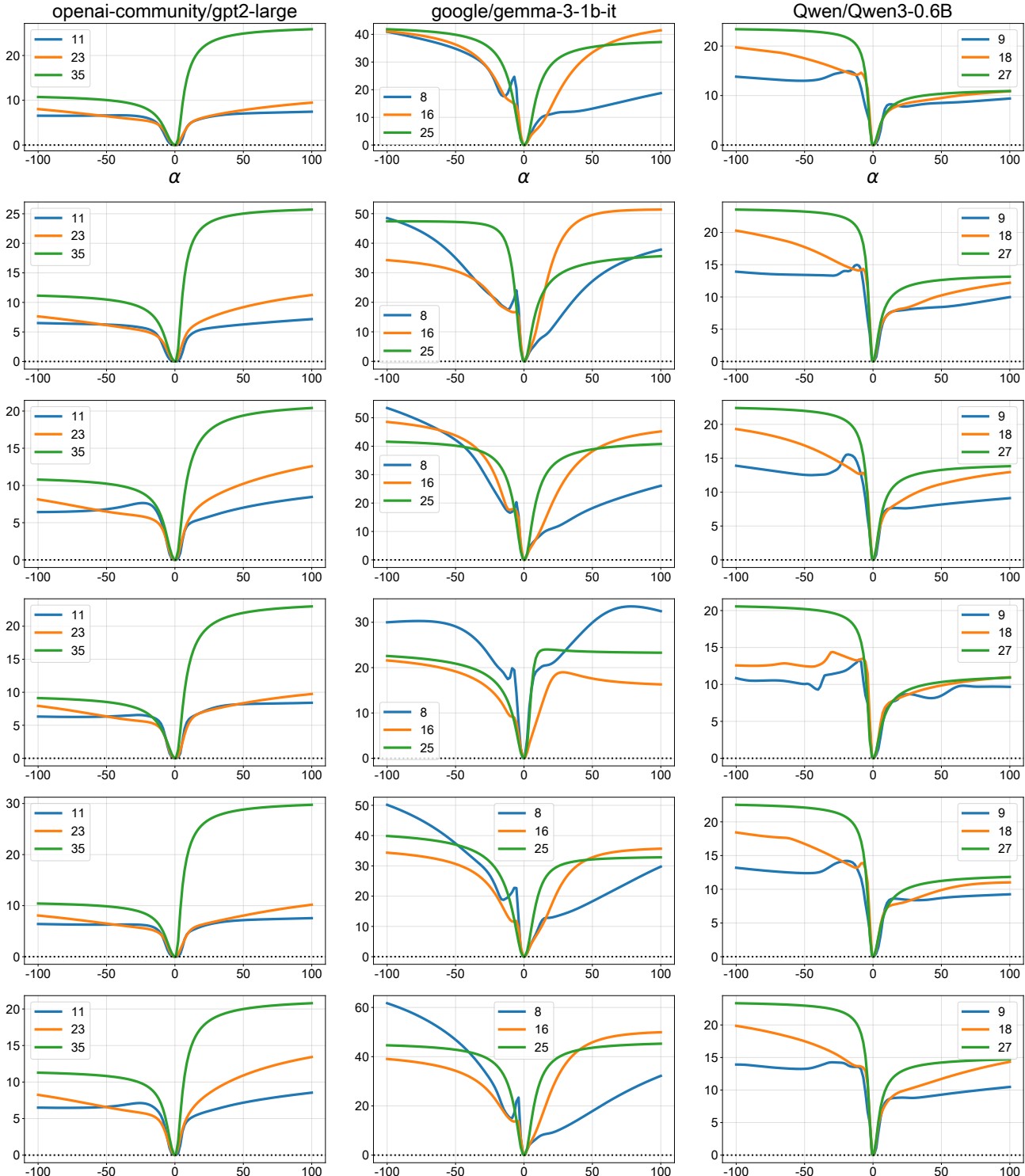

*Figure A.29.* Influence of steering strength $\alpha$ on the cross-entropy $\Delta\mathrm{CE}(\alpha)$ for the concepts (top to bottom): apathetic, depression, evil, humorous, impolite, and joy. Each row of three plots corresponds to a single steered concept, and each column corresponds to a different model. Steering is applied at three layers (early, middle, late), indicated in each legend. As predicted by Theorem 3.8, $\Delta\mathrm{CE}(\alpha)$ is locally $U$-shaped around $\alpha = 0$ and saturates for large $|\alpha|$, in line with Proposition 4.1.

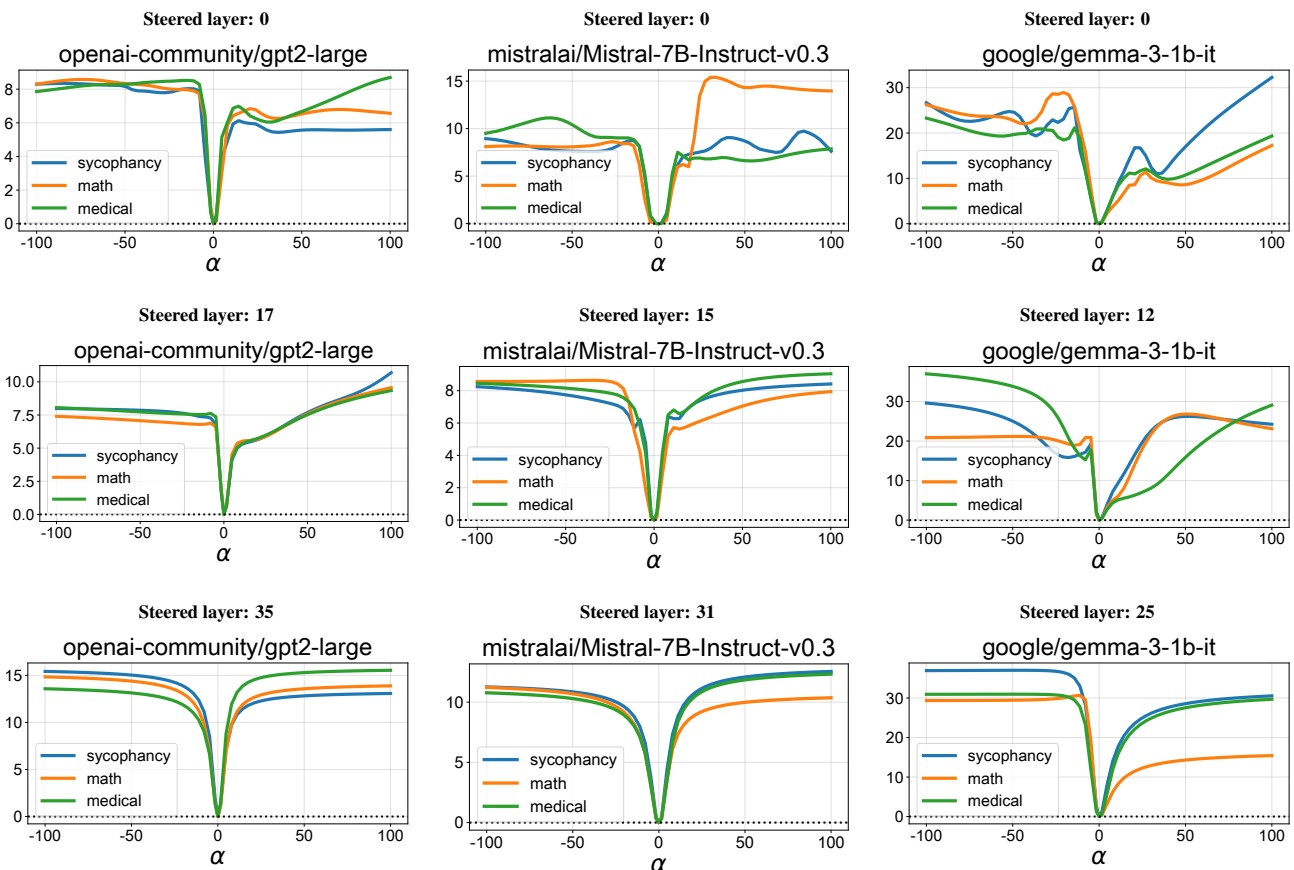

*Figure A.30.* Influence of steering strength on cross-entropy for the concepts math, medical, and sycophancy. Columns correspond to GPT2-large, Mistral-7B-Instruct-v0.3, and Gemma-3-1B-it. Every panel header gives the exact steered layer index. As predicted by Theorem 3.8, $\Delta\text{CE}(\alpha)$ is locally $U$-shaped around $\alpha = 0$ and saturates for large $|\alpha|$, in line with Proposition 4.1.

# B. Proofs and Additional Results

## B.1. Additional Results

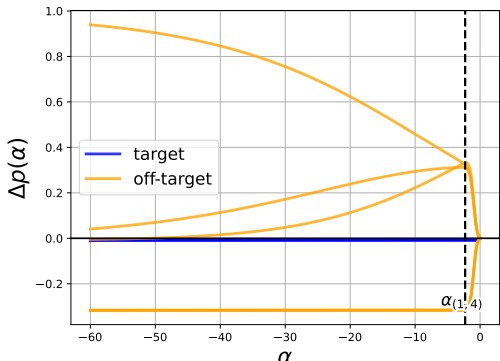

*Figure B.1.* Next-token probability increases $\Delta p(\alpha)$ for a fixed context and **negative** $\alpha$. Each curve corresponds to a token $z$: target tokens $\mathcal{T}$ are in blue and off-target tokens in orange. Most off-target tokens exhibit a "bump" (peaking at $\alpha_{(1,4)}$), while one off-target token decreases on $\mathbb{R}$ and target tokens are increasing on $\mathbb{R}_-$.

**Generalizing our results to $a_z$ depending on $j$.** Inspecting the proofs shows that all results, except Remark 3.4, rely on Lemma B.4. Consequently, our main theorems continue to hold verbatim as long as the same sign-separation property holds for the log-odds $M$ (Lemma B.4). However, if one allows $a_z$ and $b_z$ to depend on the context index $j$ without further structure, this sign-separation property may fail.

A simple generalization that preserves sign separation is to allow $a_z$ to depend on $j$ but keep $b_z$ independent of $j$, assuming $a_{j,z} > b_z$. This is interpretable: only in-concept (meaning, $z, \mathbf{c}_j \in C_k$) probabilities vary with the context, while off-concept probabilities remain at a small baseline level $b_z$. Allowing $b_z$ to depend on $j$ while still enforcing Lemma B.4 is possible, but typically leads to a less interpretable assumption. Thus, roughly speaking, if we see Lemma B.4 as an assumption, then our results work. Additionally, Lemma B.4 seems to be true in practice see Appendix A.

**Plot of $\Delta p(\alpha)$ for negative steering strength.** We provide the counterpart of Figure 6 for negative $\alpha$, see Figure B.1.

**Perfect training of the UFM.** To illustrate that Assumption 2 is attainable in our theoretical setting, we train a UFM with gradient descent on cross-entropy loss on the following dataset instantiation from Definition 2.1, which satisfies Assumption 1:

$$\forall z \in [V], \quad \begin{cases} a_z := \frac{1-\varepsilon}{s} \\ b_z := \frac{\varepsilon}{(G-1)s}, \end{cases}$$

with $\varepsilon \in (0, (G-1)/G)$ a smoothing parameter. The dataset entropy is $\approx 1.3317$ (a lower bound on the achievable loss (Thrampoulidis, 2024)), and we reach a loss of $\approx 1.3318$, indicating that the model learns the dataset essentially **perfectly**.

As stated in Section 3.1, we have an additional result about the limits of $\Delta p(\alpha)$ (Definition 3.1):

**Proposition B.1 (Limits of $\Delta p(\alpha)$).** *Given a context index $j \in [m]$, and a token $z \in [V]$, the limits of $\Delta p(\alpha)$ when $\alpha \to +\infty$ is:*

$$\lim_{\alpha \to +\infty} \Delta p(\alpha) = \mathbb{1}_{z \in \overline{M}} \frac{p(z \mid \mathbf{c}_j)}{\sum_{z' \in \overline{M}} p(z' \mid \mathbf{c}_j)} - p(z \mid \mathbf{c}_j). \tag{6}$$

*Similarly, for the limit $\alpha \to -\infty$, replace $\overline{M}$ by $\underline{M}$ in Equation (6).*

Equation (6) has the following interpretation: in the limit $\alpha \to +\infty$ (resp. $\alpha \to -\infty$), $\Delta p(\alpha)$ concentrates all its mass on the tokens $z \in \overline{M}$ (resp. $z \in \underline{M}$). If multiple tokens attain the maximal or minimal log-odds, the probability mass is shared among all such tokens.

*Proof.* Let us prove that softmax behaves as follows when scaling the steering strength $\alpha$:

$$\forall z \in [V], \qquad \lim_{\alpha \to +\infty} \sigma_z\big(f_\alpha(\mathbf{c}_j)\big) = \mathbb{1}_{z \in \overline{M}} \frac{p(z \mid \mathbf{c}_j)}{\sum_{z' \in \overline{M}} p(z' \mid \mathbf{c}_j)} \, . \tag{7}$$

where $\overline{M}$ is the set of tokens attaining the maximum log-odd $M_{\max}$. A short proof of the previous display is as follows:

$$
\begin{aligned}
\sigma_z\big(f_\alpha(\mathbf{c}_j)\big) &= \frac{p(z \mid \mathbf{c}_j)\exp\left(\alpha M(z)\right)}{\sum_{z' \in [V]} p(z' \mid \mathbf{c}_j)\exp\left(\alpha M(z')\right)} \\
&= \frac{p(z \mid \mathbf{c}_j)\exp\left(\alpha M(z)\right)}{\sum_{z' \in [V]} p(z' \mid \mathbf{c}_j)\exp\left(\alpha M(z')\right)} \frac{\exp\left(-\alpha M_{\max}\right)}{\exp\left(-\alpha M_{\max}\right)} \\
&= \frac{p(z \mid \mathbf{c}_j)\exp\left(\alpha(M(z) - M_{\max})\right)}{\sum_{z' \in [V]} p(z' \mid \mathbf{c}_j)\exp\left(\alpha(M(z') - M_{\max})\right)} \\
&= \begin{cases} \frac{p(z\mid\mathbf{c}_j)}{\sum_{z' \in \overline{M}} p(z'\mid\mathbf{c}_j) + \sum_{z' \notin \overline{M}} p(z'\mid\mathbf{c}_j)\exp(\alpha(M(z')-M_{\max}))} & \text{if } z \in \overline{M}, \\[2mm] \frac{p(z\mid\mathbf{c}_j)\exp(\alpha(M(z)-M_{\max}))}{\sum_{z' \in [V]} p(z'\mid\mathbf{c}_j)\exp(\alpha(M(z')-M_{\max}))} & \text{otherwise.} \end{cases}
\end{aligned}
$$

In the first case ($z \in \overline{M}$) of the previous display, as $(M(z') - M_{\max}) < 0$ with $z' \notin \overline{M}$, the limit is

$$\lim_{\alpha \to +\infty} \frac{p(z \mid \mathbf{c}_j)}{\sum_{z' \in \overline{M}} p(z' \mid \mathbf{c}_j) + \sum_{z' \notin \overline{M}} p(z' \mid \mathbf{c}_j)\exp\left(\alpha(M(z') - M_{\max})\right)} = \frac{p(z \mid \mathbf{c}_j)}{\sum_{z' \in \overline{M}} p(z' \mid \mathbf{c}_j)} \, .$$

In the second case ($z \notin \overline{M}$), the limit is done by bounding the term, using the fact that

$$\sum_{z' \in [V]} p(z' \mid \mathbf{c}_j)\exp\left(\alpha(M(z') - M_{\max})\right) \geq p(z^\star \mid \mathbf{c}_j)\exp\left(0\right)$$

where $z^\star \in \overline{M}$. We get the following bound:

$$0 < \frac{p(z \mid \mathbf{c}_j)\exp\left(\alpha(M(z) - M_{\max})\right)}{\sum_{z' \in [V]} p(z' \mid \mathbf{c}_j)\exp\left(\alpha(M(z') - M_{\max})\right)} \leq \frac{p(z \mid \mathbf{c}_j)}{p(z^\star \mid \mathbf{c}_j)}\exp\left(\alpha(M(z) - M_{\max})\right) \, ,$$

which implies that the second case term goes to 0 as $\alpha \to +\infty$ (because $(M(z) - M_{\max}) < 0$ with $z \notin \overline{M}$).

Using the previous display we get:

$$\lim_{\alpha \to +\infty} \Delta p(\alpha) = \mathbb{1}_{z \in \overline{M}} \frac{p(z \mid \mathbf{c}_j)}{\sum_{z' \in \overline{M}} p(z' \mid \mathbf{c}_j)} - p(z \mid \mathbf{c}_j) \, . \tag{8}$$

Same thing for $\alpha \to -\infty$. $\qquad\square$

### B.2. Technical Lemmas

In the following we introduce and prove the technical lemmas needed for Section 3. In the UFM model, activation steering on the embedding $\mathbf{h}_j$ admits an explicit expression for the resulting model output:

**Lemma B.2 (Steering on UFM).** *Steering the embedding* $\mathbf{h}_j$ *along the direction* $\mathbf{v}$ *from Equation* (4) *with strength* $\alpha \in \mathbb{R}$, *we obtain the steered logits*

$$f_\alpha(\mathbf{c}_j) := \mathbf{W}\big(\mathbf{h}_j + \alpha\mathbf{v}\big) = \boldsymbol{\ell}_j + \frac{\alpha}{q}\left(\sum_{i \in P} \boldsymbol{\ell}_i - \sum_{i \in N} \boldsymbol{\ell}_i\right) \, ,$$

*where* $\boldsymbol{\ell}_j := f(\mathbf{c}_j)$ *are the unsteered logits for context* $\mathbf{c}_j$.

*Proof.* The rewriting is a direct consequence of the UFM model and steering vector $\mathbf{v}$ linearity:

$$
\begin{aligned}
f_\alpha(\mathbf{c}_j) &= \mathbf{W}\left(\mathbf{H}\mathbf{e}_j + \alpha\left(\frac{1}{q}\sum_{i\in P}\mathbf{H}\mathbf{e}_i - \frac{1}{q}\sum_{i\in N}\mathbf{H}\mathbf{e}_i\right)\right) \\
&= \mathbf{W}\mathbf{H}\left(\mathbf{e}_j + \alpha\left(\frac{1}{q}\sum_{i\in P}\mathbf{e}_i - \frac{1}{q}\sum_{i\in N}\mathbf{e}_i\right)\right) \\
&= \boldsymbol{\ell}_j + \frac{\alpha}{q}\left(\sum_{i\in P}\boldsymbol{\ell}_i - \sum_{i\in N}\boldsymbol{\ell}_i\right).
\end{aligned}
$$

$\square$

Thus, studying activation steering reduces to analyzing how the softmax behaves under a linear shift of its input $\boldsymbol{\ell}_j$ by the vector $\left(\sum_{i\in P}\boldsymbol{\ell}_i - \sum_{i\in N}\boldsymbol{\ell}_i\right)$.

The log-odds $M(z)$ (Definition 3.2) are central because steering modifies the softmax by reweighting each token probability $p(z\mid\mathbf{c}_j)$ by the exponential factor $\exp(\alpha M(z))$.

**Lemma B.3 (Rewriting $\Delta p(\alpha)$).** *Assume Assumption 2. The first component of $\Delta p(\alpha)$ can be rewritten as follows:*

$$
\sigma_z\big(f_\alpha(\mathbf{c}_j)\big) = \frac{p(z\mid\mathbf{c}_j)\exp(\alpha M(z))}{\sum_{z'\in[V]}p(z'\mid\mathbf{c}_j)\exp(\alpha M(z'))}.
$$

*Proof.* We express explicitly $\sigma_z\big(f_\alpha(\mathbf{c}_j)\big)$ in terms of $p(z\mid\mathbf{c}_j)$ and log-odds $M(z)$ using the rewriting of the logits from Lemma B.2:

$$
\sigma_z\big(f_\alpha(\mathbf{c}_j)\big) = \sigma_z\left(\boldsymbol{\ell}_j + \frac{\alpha}{q}\left(\sum_{i\in P}\boldsymbol{\ell}_i - \sum_{i\in N}\boldsymbol{\ell}_i\right)\right).
$$

By Assumption 2, we have $\sigma_z(\boldsymbol{\ell}_j) = p(z\mid\mathbf{c}_j)$ and using that the **softmax is shift-invariant**, there exists $\beta_j \in \mathbb{R}$ s.t. $\ell_{j,z} = \log(p(z\mid\mathbf{c}_j)) + \beta_j$. Using this representation, and the notation $p(\cdot\mid\mathbf{c}_j) := (p(z\mid\mathbf{c}_j))_{z\in[V]}$ with log applied **element-wise** to vectors, we get

$$
\begin{aligned}
\sigma_z\big(f_\alpha(\mathbf{c}_j)\big) &= \sigma_z\left(\log(p(\cdot\mid\mathbf{c}_j)) + \beta_j\mathbf{1} + \frac{\alpha}{q}\left(\sum_{u\in P}\log(p(\cdot\mid\mathbf{c}_u)) - \sum_{v\in N}\log(p(\cdot\mid\mathbf{c}_v)) + \sum_{u\in P}\beta_u\mathbf{1} - \sum_{v\in N}\beta_v\mathbf{1}\right)\right) \\
&= \sigma_z\left(\log(p(\cdot\mid\mathbf{c}_j)) + \frac{\alpha}{q}\left(\sum_{u\in P}\log(p(\cdot\mid\mathbf{c}_u)) - \sum_{v\in N}\log(p(\cdot\mid\mathbf{c}_v))\right) + \beta_j\mathbf{1} + \frac{\alpha}{q}\left(\sum_{u\in P}\beta_u\mathbf{1} - \sum_{v\in N}\beta_v\mathbf{1}\right)\right) \\
&= \sigma_z\left(\log(p(\cdot\mid\mathbf{c}_j)) + \frac{\alpha}{q}\left(\sum_{u\in P}\log(p(\cdot\mid\mathbf{c}_u)) - \sum_{v\in N}\log(p(\cdot\mid\mathbf{c}_v))\right)\right).
\end{aligned}
$$

The product $\prod$ and division of vectors $p(\cdot\mid\mathbf{c}_j)$ is done **element-wise** in the following:

$$
\begin{aligned}
\sigma_z\big(f_\alpha(\mathbf{c}_j)\big) &= \sigma_z\left(\log(p(\cdot\mid\mathbf{c}_j)) + \frac{\alpha}{q}\log\left(\frac{\prod_{u\in P}p(\cdot\mid\mathbf{c}_u)}{\prod_{v\in N}p(\cdot\mid\mathbf{c}_v)}\right)\right) \\
&= \sigma_z\left(\log(p(\cdot\mid\mathbf{c}_j)) + \alpha\mathbf{m}\right),
\end{aligned}
$$

where $\mathbf{m} := (M(1),\ldots,M(V))^\top \in \mathbb{R}^V$ is the vector of log-odds. Final step is to write the softmax $\sigma_z\big(f_\alpha(\mathbf{c}_j)\big)$ explicitly:

$$
\begin{aligned}
\sigma_z\big(f_\alpha(\mathbf{c}_j)\big) &= \frac{\exp(\log(p(z\mid\mathbf{c}_j)) + \alpha M(z))}{\sum_{z'\in[V]}\exp(\log(p(z'\mid\mathbf{c}_j)) + \alpha M(z'))} \\
&= \frac{\exp(\log(p(z\mid\mathbf{c}_j)))\exp(\alpha M(z))}{\sum_{z'\in[V]}\exp(\log(p(z'\mid\mathbf{c}_j)))\exp(\alpha M(z'))} \\
&= \frac{p(z\mid\mathbf{c}_j)\exp(\alpha M(z))}{\sum_{z'\in[V]}p(z'\mid\mathbf{c}_j)\exp(\alpha M(z'))}.
\end{aligned}
$$

$\square$

As a first step toward formalizing why steering makes concept tokens $\mathcal{C}$ more likely as $\alpha$ increases, we establish a sign-separation property for the log-odds $M(z)$:

**Lemma B.4 (Log-odds $M(z)$ sign separation).** *Assume Assumption 1. Let $\mathcal{T}$ be the target concept. For any $z \in [V]$, we have $z \in \mathcal{T}$ if, and only if, $M(z) > 0$.*

*Proof.* Given $z \in [V]$, $\mathcal{T}$ the target concept, and the dataset of Definition 2.1 satisfying Assumption 1, the log-odds $M(z)$ can be rewritten as:

$$M(z) = \frac{1}{q} \log \left( \frac{\prod_{i \in P} p(z \mid \mathbf{c}_i)}{\prod_{i \in N} p(z \mid \mathbf{c}_i)} \right)$$

$$= \begin{cases} \frac{1}{q} \log \left( \frac{(a_z)^q}{(b_z)^q} \right) & \text{, if } z \in \mathcal{T}, \\ \frac{1}{q} \log \left( \frac{(b_z)^q}{(a_z)^{q_z}(b_z)^{q-q_z}} \right) & \text{, otherwise.} \end{cases}$$

$$= \begin{cases} \log \left( \frac{a_z}{b_z} \right), \\ -\frac{q_z}{q} \log \left( \frac{a_z}{b_z} \right). \end{cases}$$

with $q_z := |\{j \in N : \exists k \in [G], \mathbf{c}_j, z \in C_k\}| \in \mathbb{N}$ (note that, it can be 0).

Using the above rewriting, we obtain that in the first case ($z \in \mathcal{T}$), $M(z) = \log \left( \frac{a_z}{b_z} \right) > 0$ by Assumption 1. Otherwise, $M(z) = -\frac{q_z}{q} \log \left( \frac{a_z}{b_z} \right) \leq 0$, again by Assumption 1. $\square$

Now let us compute the derivative of $\Delta p(\alpha)$:

**Lemma B.5 (Derivative of $\Delta p(\alpha)$).** *Let $z \in [V], j \in [m]$. We have the following derivative w.r.t. $\alpha$:*

$$\Delta' p(z \mid \mathbf{c}_j, \alpha) = \sigma_z \big( f_\alpha(\mathbf{c}_j) \big) \left( M(z) - \mathbb{E}_{Z \sim \sigma \big( f_\alpha(\mathbf{c}_j) \big)} [M(Z)] \right)$$

*Proof.* First, let us denote $D_j(\alpha) := \sum_{z' \in [V]} p(z' \mid \mathbf{c}_j) \exp(\alpha M(z'))$. Using Lemma B.3, the derivation is as follows:

$$\Delta' p(z \mid \mathbf{c}_j, \alpha) = \frac{\mathrm{d}}{\mathrm{d}\alpha} \sigma_z \big( \mathbf{W}(\mathbf{h}_j + \alpha \mathbf{v}) \big)$$

$$= \frac{\mathrm{d}}{\mathrm{d}\alpha} \left( \frac{p(z \mid \mathbf{c}_j) \exp(\alpha M(z))}{\sum_{z' \in [V]} p(z' \mid \mathbf{c}_j) \exp(\alpha M(z'))} \right)$$

$$= \frac{p(z \mid \mathbf{c}_j) \exp(\alpha M(z)) M(z) D_j(\alpha) - p(z \mid \mathbf{c}_j) \exp(\alpha M(z)) D_j'(\alpha)}{D_j(\alpha)^2}$$

$$= \frac{p(z \mid \mathbf{c}_j) \exp(\alpha M(z))}{D_j(\alpha)} \left( \frac{M(z) D_j(\alpha)}{D_j(\alpha)} - \frac{D_j'(\alpha)}{D_j(\alpha)} \right)$$

$$= \sigma_z \big( f_\alpha(\mathbf{c}_j) \big) \left( M(z) - \frac{D_j'(\alpha)}{D_j(\alpha)} \right).$$

The term $D_j'(\alpha)/D_j(\alpha)$ can be rewritten as follows:

$$\frac{D_j'(\alpha)}{D_j(\alpha)} = \frac{\sum_{z' \in [V]} p(z' \mid \mathbf{c}_j) \exp(\alpha M(z')) M(z')}{\sum_{z'' \in [V]} p(z'' \mid \mathbf{c}_j) \exp(\alpha M(z''))}$$

$$= \sum_{z' \in [V]} \frac{p(z' \mid \mathbf{c}_j) \exp(\alpha M(z'))}{\sum_{z'' \in [V]} p(z'' \mid \mathbf{c}_j) \exp(\alpha M(z''))} M(z')$$

$$= \sum_{z' \in [V]} \sigma_{z'} \big( f_\alpha(\mathbf{c}_j) \big) M(z')$$

$$= \mathbb{E}_{Z \sim \sigma \big( f_\alpha(\mathbf{c}_j) \big)} [M(Z)].$$

$\square$

## B.3. Proof of Theorem 3.3

Theorem 3.3 is about the monotonicity of $\Delta p(\alpha)$. Hence, we need to study the sign of the derivative of $\Delta p$. As shown in Lemma B.5, the sign of $(\Delta p)'(\alpha)$ is governed by the difference $\left( M(z) - \mathbb{E}_{Z \sim \sigma\left(f_\alpha(\mathbf{c}_j)\right)} \left[ M(Z) \right] \right)$. In this difference the only quantity which depends on $\alpha$ is $\mathbb{E}_{Z \sim \sigma\left(f_\alpha(\mathbf{c}_j)\right)} \left[ M(Z) \right]$. So in the following, we study the variations of this expectation under steering. To do so, we look at its derivative:

$$\frac{\mathrm{d}}{\mathrm{d}\alpha} \mathbb{E}_{Z \sim \sigma\left(f_\alpha(\mathbf{c}_j)\right)} \left[ M(Z) \right] = \sum_{z' \in [V]} \frac{\mathrm{d}}{\mathrm{d}\alpha} \sigma_{z'}\left(f_\alpha(\mathbf{c}_j)\right) M(z')$$

Compute $\frac{\mathrm{d}}{\mathrm{d}\alpha} \sigma_{z'}\left(f_\alpha(\mathbf{c}_j)\right)$ and replace in the previous display:

$$\frac{\mathrm{d}}{\mathrm{d}\alpha} \mathbb{E}_{Z \sim \sigma\left(f_\alpha(\mathbf{c}_j)\right)} \left[ M(Z) \right] = \sum_{z' \in [V]} \sigma_{z'}\left(f_\alpha(\mathbf{c}_j)\right) \left( M(z') - \mathbb{E}_{Z \sim \sigma\left(f_\alpha(\mathbf{c}_j)\right)} \left[ M(Z) \right] \right) M(z')$$

$$= \sum_{z' \in [V]} \sigma_{z'}\left(f_\alpha(\mathbf{c}_j)\right) M(z')^2 - \sum_{z' \in [V]} \sigma_{z'}\left(f_\alpha(\mathbf{c}_j)\right) \mathbb{E}_{Z \sim \sigma\left(f_\alpha(\mathbf{c}_j)\right)} \left[ M(Z) \right] M(z')$$

By definition of the expectation:

$$\frac{\mathrm{d}}{\mathrm{d}\alpha} \mathbb{E}_{Z \sim \sigma\left(f_\alpha(\mathbf{c}_j)\right)} \left[ M(Z) \right] = \mathbb{E}_{Z \sim \sigma\left(f_\alpha(\mathbf{c}_j)\right)} \left[ M(Z)^2 \right] - \mathbb{E}_{Z \sim \sigma\left(f_\alpha(\mathbf{c}_j)\right)} \left[ M(Z) \right] \sum_{z' \in [V]} \sigma_{z'}\left(f_\alpha(\mathbf{c}_j)\right) M(z')$$

$$= \mathbb{E}_{Z \sim \sigma\left(f_\alpha(\mathbf{c}_j)\right)} \left[ M(Z)^2 \right] - \mathbb{E}_{Z \sim \sigma\left(f_\alpha(\mathbf{c}_j)\right)} \left[ M(Z) \right]^2$$

The previous display is the variance:

$$\frac{\mathrm{d}}{\mathrm{d}\alpha} \mathbb{E}_{Z \sim \sigma\left(f_\alpha(\mathbf{c}_j)\right)} \left[ M(Z) \right] = \mathrm{Var}_{Z \sim \sigma\left(f_\alpha(\mathbf{c}_j)\right)} \left( M(Z) \right) ,$$

$$\tag{9}$$

with $\mathrm{Var}\left( M(Z) \right) > 0$ if $M(Z)$ is not constant $\sigma\left(f_\alpha(\mathbf{c}_j)\right)$-almost surely. This means that $\mathbb{E}_{Z \sim \sigma\left(f_\alpha(\mathbf{c}_j)\right)} \left[ M(Z) \right]$ is strictly increasing on $\mathbb{R}$ in $\alpha$. Now let us compute the limits of this quantity when $\alpha \to \pm\infty$, which are

$$\lim_{\alpha \to +\infty} \mathbb{E}_{Z \sim \sigma\left(f_\alpha(\mathbf{c}_j)\right)} \left[ M(Z) \right] = \max_{z \in [V]} M(z) =: M_{\max} ,$$

$$\lim_{\alpha \to -\infty} \mathbb{E}_{Z \sim \sigma\left(f_\alpha(\mathbf{c}_j)\right)} \left[ M(Z) \right] = \min_{z \in [V]} M(z) =: M_{\min} .$$

Given $\delta > 0$, we introduce the set $A_\delta := \{ z \in [V] : M_{\max} - M(z) \le \delta \}$ to control the following difference:

$$M_{\max} - \mathbb{E}_{Z \sim \sigma\left(f_\alpha(\mathbf{c}_j)\right)} \left[ M(Z) \right]$$

$$= \sum_{z' \in [V]} \sigma_{z'}\left(f_\alpha(\mathbf{c}_j)\right) \left( M_{\max} - M(z') \right)$$

$$= \sum_{z' \in A_\delta} \sigma_{z'}\left(f_\alpha(\mathbf{c}_j)\right) \left( M_{\max} - M(z') \right) + \sum_{z' \in A_\delta^{\complement}} \sigma_{z'}\left(f_\alpha(\mathbf{c}_j)\right) \left( M_{\max} - M(z') \right)$$

$$\le \delta \sum_{z' \in A_\delta} \sigma_{z'}\left(f_\alpha(\mathbf{c}_j)\right) + \left( M_{\max} - M_{\min} \right) \sum_{z' \in A_\delta^{\complement}} \sigma_{z'}\left(f_\alpha(\mathbf{c}_j)\right)$$

$$\le \delta + \left( M_{\max} - M_{\min} \right) \sum_{z' \in A_\delta^{\complement}} \sigma_{z'}\left(f_\alpha(\mathbf{c}_j)\right) .$$

Let us take $z^\star$ a token which attains the maximum log-odds $M_{\max}$. This is necessarily a concept token $\mathcal{T}$ (i.e., $z^\star \in \mathcal{T}$) because $a_z > b_z$ (Assumption 1). We now show that $\lim_{\alpha \to +\infty} \sum_{z' \in A_\delta^{\complement}} \sigma_{z'}\left(f_\alpha(\mathbf{c}_j)\right) = 0$. If $A_\delta^{\complement} = \varnothing$ (for sufficiently large

$\delta$), the sum is zero by convention. Otherwise, we proceed as follows, using Lemma B.3:

$$0 < \sum_{z' \in A_\delta^{\complement}} \sigma_{z'}\big(f_\alpha(\mathbf{c}_j)\big) = \sum_{z' \in A_\delta^{\complement}} \frac{p(z' \mid \mathbf{c}_j) \exp\left(\alpha M(z')\right)}{\sum_{z'' \in [V]} p(z'' \mid \mathbf{c}_j) \exp\left(\alpha M(z'')\right)} \qquad \text{(denominator lower bounded by } p(z^\star \mid \mathbf{c}_j) \exp\left(\alpha M_{\max}\right).)$$

$$\leq \sum_{z' \in A_\delta^{\complement}} \frac{p(z' \mid \mathbf{c}_j) \exp\left(\alpha(M_{\max} - \delta)\right)}{p(z^\star \mid \mathbf{c}_j) \exp\left(\alpha M_{\max}\right)}$$

$$= \exp\left(-\delta\alpha\right) \sum_{z' \in A_\delta^{\complement}} \frac{p(z' \mid \mathbf{c}_j)}{p(z^\star \mid \mathbf{c}_j)}.$$

By taking the limit in the previous display, we get $\lim_{\alpha \to +\infty} \sum_{z' \in A_\delta^{\complement}} \sigma_{z'}\big(f_\alpha(\mathbf{c}_j)\big) = 0$.

Finally, we take the $\limsup$ as follows:

$$\limsup_{\alpha \to +\infty} M_{\max} - \mathbb{E}_{Z \sim \sigma\big(f_\alpha(\mathbf{c}_j)\big)}\left[M(Z)\right] \leq \limsup_{\alpha \to +\infty} \left(\delta + (M_{\max} - M_{\min}) \sum_{z' \in A_\delta^{\complement}} \sigma_{z'}\big(f_\alpha(\mathbf{c}_j)\big)\right).$$

The above inequality is rewritten as

$$\limsup_{\alpha \to +\infty} M_{\max} - \mathbb{E}_{Z \sim \sigma\big(f_\alpha(\mathbf{c}_j)\big)}\left[M(Z)\right] \leq \delta.$$

The previous display's bound is uniform in $\delta > 0$, taking the limit $\delta \to 0^+$ gives

$$\limsup_{\alpha \to +\infty} M_{\max} - \mathbb{E}_{Z \sim \sigma\big(f_\alpha(\mathbf{c}_j)\big)}\left[M(Z)\right] \leq 0.$$

To finish, one remarks that

$$0 \leq \liminf_{\alpha \to +\infty} M_{\max} - \mathbb{E}_{Z \sim \sigma\big(f_\alpha(\mathbf{c}_j)\big)}\left[M(Z)\right] \leq \limsup_{\alpha \to +\infty} M_{\max} - \mathbb{E}_{Z \sim \sigma\big(f_\alpha(\mathbf{c}_j)\big)}\left[M(Z)\right] \leq 0.$$

Which implies that the limit does in fact exist and $\lim_{\alpha \to +\infty} \mathbb{E}_{Z \sim \sigma\big(f_\alpha(\mathbf{c}_j)\big)}\left[M(Z)\right] = M_{\max}$. Very similar derivations give $\lim_{\alpha \to -\infty} \mathbb{E}_{Z \sim \sigma\big(f_\alpha(\mathbf{c}_j)\big)}\left[M(Z)\right] = M_{\min}$.

Since $\alpha \mapsto \mathbb{E}_{Z \sim \sigma\big(f_\alpha(\mathbf{c}_j)\big)}\left[M(Z)\right]$ is continuous, strictly increasing on $\mathbb{R}$ and the limits are known on this interval, we have the following: there exists unique thresholds $\alpha_{(j,z)} \in \mathbb{R}$ such that

$$z \in \mathcal{T} \setminus \overline{M}, \qquad\qquad \mathbb{E}_{Z \sim \sigma\big(f_{\alpha_{(j,z)}}(\mathbf{c}_j)\big)}\left[M(Z)\right] = M(z),$$

$$z \in \overline{M}, \qquad\qquad \alpha_{(z)} := +\infty.$$

and

$$z \in \mathcal{T}^{\complement} \setminus \underline{M}, \qquad\qquad \mathbb{E}_{Z \sim \sigma\big(f_{\alpha_{(j,z)}}(\mathbf{c}_j)\big)}\left[M(Z)\right] = M(z),$$

$$z \in \underline{M}, \qquad\qquad \alpha_{(z)} := -\infty.$$

First for the limit case ($z \in \underline{M} \cup \overline{M}$), we remove the dependency in $j$ of $\alpha_{(j,z)}$ as its always equal to $\pm\infty$. With $z \in \overline{M}$, we take $\alpha_{(z)} := +\infty$ because $\lim_{\alpha \to +\infty} \mathbb{E}_{Z \sim \sigma\big(f_\alpha(\mathbf{c}_j)\big)}\left[M(Z)\right] = M_{\max}$. Moreover, the minimum log-odd $M_{\min}$ cannot be attained by concept tokens $z \in \mathcal{T}$ since $a_z > b_z$ (Assumption 1), hence $M(z) \neq M_{\min}$ for all $z \in \mathcal{T}$ and $\alpha_{(z)} := -\infty$ for $z \in \underline{M}$.

Finally, all bullet points of Theorem 3.3 follow directly from the previous arguments. For the first point, fix a token $z \in [V] \setminus (\overline{M} \cup \underline{M})$. Then $M(z) - \mathbb{E}_{Z \sim \sigma\big(f_\alpha(\mathbf{c}_j)\big)}\left[M(Z)\right]$ is positive on $(-\infty, \alpha_{(j,z)}]$ and negative for $\alpha > \alpha_{(j,z)}$, which yields the bump behavior. The second point follows from the sign separation of the log-odds (Lemma B.4) together with the fact that $\mathbb{E}_{Z \sim \sigma\big(f_\alpha(\mathbf{c}_j)\big)}\left[M(Z)\right]$ is increasing, which implies $\alpha_{(j,z')} < \alpha_{(j,z)}$ for $z \in \mathcal{T}$ and $z' \notin \mathcal{T}$. The final point again follows from the fact that for $z \in \overline{M} \cup \underline{M}$, the sign of $M(z) - \mathbb{E}_{Z \sim \sigma\big(f_\alpha(\mathbf{c}_j)\big)}\left[M(Z)\right]$ does not change with $\alpha$: it remains positive for $z \in \mathcal{T}$ and negative otherwise, since $\alpha_{(j,z)} = \pm\infty$ for such tokens. $\qquad\square$

**Proof of Remark 3.4** Let $\mathbf{c}_j \notin \mathcal{T}$ and denote by $z_1$ the concept token with the minimum log-odd in the group $\mathcal{T}$. To show that the bump for concept tokens $z \in \mathcal{T}$ happens for positive $\alpha$, it suffices to show that $\alpha_{(j,z_1)} > 0$ (as $\alpha_{(j,z_1)} \leq \alpha_{(j,z)}$ with $z \in \mathcal{T}$ by strict monotonicity of $\mathbb{E}_{Z \sim \sigma(f_\alpha(\mathbf{c}_j))}[M(Z)]$).

We are proving this fact on a specification of the dataset from Definition 2.1 (and Assumption 1), defined as follows:

$$\forall z \in [V], \quad \begin{cases} a_z := (1 - \varepsilon)\gamma_z \\ b_z := \frac{\varepsilon}{(G-1)}\omega_z\,, \end{cases}$$

where $\gamma_z \in (0, 1)$ satisfies $\sum_{z' \in C_k} \gamma_{z'} = 1$ for each $k \in [G]$ (with the same conditions for $\omega_z$). The coefficients $\gamma_z$ and $\omega_z$ are chosen so that Assumption 1 holds, i.e., $a_z > b_z$, and we assume $\varepsilon \in \left(0, \frac{G-1}{G}\right)$.

Proving that $\alpha_{(j,z_1)} > 0$ for $\varepsilon > 0$ **small enough** when $\mathbf{c}_j \notin \mathcal{T}$, is equivalent to showing

$$M(z_1) = \mathbb{E}_{Z \sim \sigma(f_{\alpha_{(j,z_1)}}(\mathbf{c}_j))}[M(Z)] > \mathbb{E}_{Z \sim \sigma(f(\mathbf{c}_j))}[M(Z)] \tag{10}$$

by strict monotonicity of $\mathbb{E}_{Z \sim \sigma(f_\alpha(\mathbf{c}_j))}[M(Z)]$ in $\alpha$. The previous inequality is hard to prove as $\mathbb{E}_{Z \sim \sigma(f(\mathbf{c}_j))}[M(Z)]$ has a non-trivial expression, so we start by bounding it:

$$
\begin{aligned}
\mathbb{E}_{Z \sim \sigma(f(\mathbf{c}_j))}[M(Z)] &= \sum_{z' \in [V]} \sigma_{z'}(f(\mathbf{c}_j))M(z') \\
&= \sum_{z' \in [V]} p(z' \mid \mathbf{c}_j)M(z') && \text{(by Assumption 2.)} \\
&\leq \sum_{z' \in \mathcal{T}} p(z' \mid \mathbf{c}_j)M(z') && \text{(as } M(z) \leq 0 \text{ for } z \notin \mathcal{T}, \text{ see Lemma B.4.)} \\
&= \sum_{z' \in \mathcal{T}} b_{z'}M(z') && \text{(as } p(z \mid \mathbf{c}_j) = b_z \text{ for } \mathbf{c}_j \notin \mathcal{T} \text{ and } z \in \mathcal{T}.\text{)} \\
&= \sum_{z' \in \mathcal{T}} b_{z'} \log\left(\frac{a_{z'}}{b_{z'}}\right) && \text{(as } M(z) = \log(a_z/b_z) \text{ for } z \in \mathcal{T}, \text{ see Lemma B.4.)}
\end{aligned}
$$

In this specific dataset, $\beta := \sum_{z'' \in \mathcal{T}} b_{z''} = \frac{\varepsilon}{(G-1)}$ and $\rho := \sum_{z'' \in \mathcal{T}} a_{z''} = 1 - \varepsilon$. We continue the bounding process as follows

$$
\begin{aligned}
\mathbb{E}_{Z \sim \sigma(f(\mathbf{c}_j))}[M(Z)] &\leq \sum_{z' \in \mathcal{T}} b_{z'} \log\left(\frac{a_{z'}}{b_{z'}}\right) \\
&= \beta \sum_{z' \in \mathcal{T}} \frac{b_{z'}}{\beta} \log\left(\frac{\rho}{\beta}\frac{a_{z'}/\rho}{b_{z'}/\beta}\right) \\
&= \beta \sum_{z' \in \mathcal{T}} \frac{b_{z'}}{\beta} \log\left(\frac{\rho}{\beta}\right) + \beta \sum_{z' \in \mathcal{T}} \frac{b_{z'}}{\beta} \log\left(\frac{a_{z'}/\rho}{b_{z'}/\beta}\right)
\end{aligned}
$$

We remark that $\sum_{z' \in \mathcal{T}} \frac{b_{z'}}{\beta} \log\left(\frac{a_{z'}/\rho}{b_{z'}/\beta}\right)$ is equal to the *negative* of the Kullback–Leibler divergence between $(b_{z'}/\beta)_{z' \in \mathcal{T}}$ and $(a_{z'}/\rho)_{z' \in \mathcal{T}}$ denoted as $\mathrm{KL}(b./\beta \| a./\rho)$:

$$
\begin{aligned}
\mathbb{E}_{Z \sim \sigma(f(\mathbf{c}_j))}[M(Z)] &= \beta \left( \sum_{z' \in \mathcal{T}} \frac{b_{z'}}{\beta} \log\left(\frac{\rho}{\beta}\right) - \mathrm{KL}(b./\beta \| a./\rho) \right) \\
&\leq \beta \sum_{z' \in \mathcal{T}} \frac{b_{z'}}{\beta} \log\left(\frac{\rho}{\beta}\right) && \text{(by Gibbs' inequality } \mathrm{KL}(b./\beta \| a./\rho) \geq 0.\text{)} \\
&= \beta \log\left(\frac{\rho}{\beta}\right) && \left(\text{as } \sum_{z' \in \mathcal{T}} \frac{b_{z'}}{\beta} = 1.\right) \\
&= \frac{\varepsilon}{(G-1)} \log\left(\frac{1-\varepsilon}{\varepsilon}(G-1)\right) .
\end{aligned}
$$

Let us remind that $M(z_1) = \log\left(\frac{(1-\varepsilon)(G-1)}{\varepsilon}\frac{\gamma_{z_1}}{\omega_{z_1}}\right)$ by the proof of Lemma B.4. To avoid complicated solution to Inequality (10) using the Lambert $W$ function, we compute the limit $\varepsilon \to 0^+$ of $F(\cdot)$ defined as:

$$F(\varepsilon) := M(z_1) - \frac{\varepsilon}{(G-1)}\log\left(\frac{1-\varepsilon}{\varepsilon}(G-1)\right) = \log\left(\frac{(1-\varepsilon)(G-1)}{\varepsilon}\frac{\gamma_{z_1}}{\omega_{z_1}}\right) - \frac{\varepsilon}{(G-1)}\log\left(\frac{1-\varepsilon}{\varepsilon}(G-1)\right).$$

We now compute the limit as follows:

$$\lim_{\varepsilon\to 0^+}\log\left(\frac{(1-\varepsilon)(G-1)}{\varepsilon}\frac{\gamma_{z_1}}{\omega_{z_1}}\right) = +\infty \qquad\qquad \left(\text{as } \frac{\gamma_{z_1}(G-1)}{\omega_{z_1}} > 0.\right)$$

$$\lim_{\varepsilon\to 0^+}\frac{\varepsilon}{(G-1)}\log\left(\frac{1-\varepsilon}{\varepsilon}(G-1)\right) = \lim_{\varepsilon\to 0^+}\frac{\varepsilon}{(G-1)}\log\left(\left(\frac{1}{\varepsilon}-1\right)(G-1)\right) = 0 \quad \left(\text{as } \lim_{x\to+\infty}\log(x)/x = 0.\right)$$

So $\lim_{\varepsilon\to 0^+} F(\varepsilon) = +\infty$, which means that there exists $\varepsilon_0 < (G-1)/G$ such that for all $\varepsilon \in (0,\varepsilon_0)$, $F(\varepsilon) > 0$. With $\varepsilon \in (0,\varepsilon_0)$, by combining $F(\varepsilon) > 0$ with the upper-bound on $\mathbb{E}_{Z\sim\sigma(f(\mathbf{c}_j))}[M(Z)]$, we get the Inequality (10):

$$M(z_1) > \frac{\varepsilon}{(G-1)}\log\left(\frac{1-\varepsilon}{\varepsilon}(G-1)\right) \geq \mathbb{E}_{Z\sim\sigma(f(\mathbf{c}_j))}[M(Z)],$$

which is equivalent to $\alpha_{(j,z_1)} > 0$ as desired. $\qquad\square$

## B.4. Proof of Theorem 3.6

Fix a context index $j \in [m]$ and a concept $\mathcal{C}$. Define

$$F_{\mathcal{C},j}(\alpha) := \sum_{z\in\mathcal{C}}\sigma_z\big(f_\alpha(\mathbf{c}_j)\big).$$

By Definition 3.5, Definition 3.1 and Assumption 2,

$$\Delta p(\mathcal{C} \mid \mathbf{c}_j, \alpha) = \frac{1}{|\mathcal{C}|}\sum_{z\in\mathcal{C}}\Big(\sigma_z\big(f_\alpha(\mathbf{c}_j)\big) - p(z\mid\mathbf{c}_j)\Big) = \frac{F_{\mathcal{C},j}(\alpha) - F_{\mathcal{C},j}(0)}{|\mathcal{C}|}.$$

By Lemma B.3,

$$\sigma_z\big(f_\alpha(\mathbf{c}_j)\big) = \frac{p(z\mid\mathbf{c}_j)\exp(\alpha M(z))}{\sum_{z'\in[V]}p(z'\mid\mathbf{c}_j)\exp(\alpha M(z'))}.$$

Let $Z \sim (\sigma_z(f_\alpha(\mathbf{c}_j)))_{z\in[V]}$ and set

$$\mu_{\mathcal{C},j}(\alpha) := \mathbb{E}[M(Z)|Z\in\mathcal{C}], \qquad \mu_{\mathcal{C}^{\complement},j}(\alpha) := \mathbb{E}[M(Z)|Z\notin\mathcal{C}].$$

Using Lemma B.5 and summing over $z \in \mathcal{C}$,

$$\begin{aligned}
\frac{\mathrm{d}}{\mathrm{d}\alpha}F_{\mathcal{C},j}(\alpha) &= \sum_{z\in\mathcal{C}}\sigma_z\big(f_\alpha(\mathbf{c}_j)\big)\Big(M(z) - \mathbb{E}[M(Z)]\Big)\\
&= \sum_{z\in\mathcal{C}}F_{\mathcal{C},j}(\alpha)\frac{\sigma_z\big(f_\alpha(\mathbf{c}_j)\big)}{F_{\mathcal{C},j}(\alpha)}\Big(M(z) - \mathbb{E}[M(Z)]\Big)\\
&= F_{\mathcal{C},j}(\alpha)\left(\sum_{z\in\mathcal{C}}\frac{\sigma_z\big(f_\alpha(\mathbf{c}_j)\big)}{F_{\mathcal{C},j}(\alpha)}M(z) - \mathbb{E}[M(Z)]\sum_{z\in\mathcal{C}}\frac{\sigma_z\big(f_\alpha(\mathbf{c}_j)\big)}{F_{\mathcal{C},j}(\alpha)}\right)\\
&= F_{\mathcal{C},j}(\alpha)\Big(\mu_{\mathcal{C},j}(\alpha) - \mathbb{E}[M(Z)]\Big).
\end{aligned}$$

Moreover, by the law of total expectation and using that $\mathbb{P}(Z\in\mathcal{C}) = F_{\mathcal{C},j}(\alpha)$ (as $Z$ is a discrete random variable),

$$\mathbb{E}[M(Z)] = F_{\mathcal{C},j}(\alpha)\mu_{\mathcal{C},j}(\alpha) + \big(1 - F_{\mathcal{C},j}(\alpha)\big)\mu_{\mathcal{C}^{\complement},j}(\alpha).$$

Therefore, $F_{\mathcal{C},j}$ satisfies the following ordinary differential equation (ODE), which is nearly the ODE satisfied by the sigmoid function up to the term $\left(\mu_{\mathcal{C},j}(\alpha) - \mu_{\mathcal{C}^{\complement},j}(\alpha)\right)$:

$$\frac{\mathrm{d}}{\mathrm{d}\alpha} F_{\mathcal{C},j}(\alpha) = F_{\mathcal{C},j}(\alpha)\left(1 - F_{\mathcal{C},j}(\alpha)\right)\left(\mu_{\mathcal{C},j}(\alpha) - \mu_{\mathcal{C}^{\complement},j}(\alpha)\right), \tag{11}$$

Since $F_{\mathcal{C},j}(\alpha) \in (0,1)$, we can divide both sides by $F_{\mathcal{C},j}(\alpha)\left(1 - F_{\mathcal{C},j}(\alpha)\right)$, and direct computations yield

$$\frac{\mathrm{d}}{\mathrm{d}\alpha} \log\left(\frac{F_{\mathcal{C},j}(\alpha)}{1 - F_{\mathcal{C},j}(\alpha)}\right) = \mu_{\mathcal{C},j}(\alpha) - \mu_{\mathcal{C}^{\complement},j}(\alpha).$$

Integrating both sides of the previous display from $0$ to $\alpha$ yields

$$\log\left(\frac{F_{\mathcal{C},j}(\alpha)}{1 - F_{\mathcal{C},j}(\alpha)}\right) = r_j + \nu_j(\alpha), \qquad r_j := \log\left(\frac{F_{\mathcal{C},j}(0)}{1 - F_{\mathcal{C},j}(0)}\right), \qquad \nu_j(\alpha) := \int_0^\alpha \left(\mu_{\mathcal{C},j}(t) - \mu_{\mathcal{C}^{\complement},j}(t)\right) \mathrm{d}t.$$

The previous display is the logit function, which is the inverse of the sigmoid function $\phi$. Hence $F_{\mathcal{C},j}(\alpha) = \phi(\nu_j(\alpha) + r_j)$ and

$$\Delta p(\mathcal{C} \mid \mathbf{c}_j, \alpha) = \frac{1}{|\mathcal{C}|}\left(F_{\mathcal{C},j}(\alpha) - F_{\mathcal{C},j}(0)\right) = \frac{1}{|\mathcal{C}|}\left(\phi(\nu_j(\alpha)+r_j) - \phi(r_j)\right) = \frac{1}{2|\mathcal{C}|}\left(\tanh\left(\frac{\nu_j(\alpha)+r_j}{2}\right) - \tanh\left(\frac{r_j}{2}\right)\right),$$

as $\tanh(x) = 2\phi(2x) - 1$. Setting $r_j' := \tanh\left(\frac{r_j}{2}\right)$ gives the claimed representation in Theorem 3.6.

**Proving monotonicity of $\Delta p(\mathcal{T}|\alpha)$, $\Delta p(\mathcal{C}|\alpha)$ and limits of $\Delta p(\mathcal{C}'|\alpha)$.** Let $\mathcal{T}$ be the target concept we steer towards, Lemma B.4 gives $M(z) > 0$ for $z \in \mathcal{T}$ and $M(z) \leq 0$ for $z \notin \mathcal{T}$, hence $\mu_{\mathcal{T},j}(\alpha) > 0$ and $\mu_{\mathcal{T}^{\complement},j}(\alpha) \leq 0$ for all $\alpha$. Thus $\mu_{\mathcal{T},j}(\alpha) - \mu_{\mathcal{T}^{\complement},j}(\alpha) > 0$, implying $\frac{\mathrm{d}}{\mathrm{d}\alpha} F_{\mathcal{T},j}(\alpha) > 0$ by Equation (11). Meaning, $\Delta p(\mathcal{T} \mid \mathbf{c}_j, \alpha)$ is strictly increasing in $\alpha$. Additionally, the growth of $\nu_j$ is at most linear because $\left|\mu_{\mathcal{T},j}(t) - \mu_{\mathcal{T}^{\complement},j}(t)\right| \leq \max_{z \in [V]} M(z) - \min_{z \in [V]} M(z)$ as the log-odds $M(z)$ are bounded w.r.t $\alpha$. Implying the following by linearity of the integral:

$$|\nu_j(\alpha)| \leq \left(\max_{z \in [V]} M(z) - \min_{z \in [V]} M(z)\right) |\alpha|.$$

Next, if $\mathcal{C}' \neq \mathcal{T}$ and $\mathcal{C}' \cap (\overline{M} \cup \underline{M}) = \varnothing$, Equation (7) in Proposition B.1 implies $F_{\mathcal{C}',j}(\alpha) \to 0$ as $\alpha \to \pm\infty$, hence

$$\lim_{\alpha \to \pm\infty} \Delta p(\mathcal{C}' \mid \mathbf{c}_j, \alpha) = \lim_{\alpha \to \pm\infty} \frac{F_{\mathcal{C}',j}(\alpha) - F_{\mathcal{C}',j}(0)}{|\mathcal{C}'|} = -\frac{F_{\mathcal{C}',j}(0)}{|\mathcal{C}'|} = -\frac{1}{|\mathcal{C}'|}\sum_{z \in \mathcal{C}'} p(z \mid \mathbf{c}_j).$$

Finally, if $\max_{z \in \mathcal{C}} M(z) \leq \min_{z \notin \mathcal{C}} M(z)$, then for all $\alpha$,

$$\mu_{\mathcal{C},j}(\alpha) \leq \max_{z \in \mathcal{C}} M(z) \leq \min_{z \notin \mathcal{C}} M(z) \leq \mu_{\mathcal{C}^{\complement},j}(\alpha),$$

then $\mu_{\mathcal{C},j}(\alpha) - \mu_{\mathcal{C}^{\complement},j}(\alpha) \leq 0$, implying $\frac{\mathrm{d}}{\mathrm{d}\alpha} F_{\mathcal{C},j}(\alpha) \leq 0$ by Equation (11). Meaning, $\Delta p(\mathcal{C} \mid \mathbf{c}_j, \alpha)$ is decreasing in $\alpha$. $\quad\square$

### B.5. Proof of Theorem 3.8

First, as in Thrampoulidis (2024), we can rewrite the cross-entropy as follows:

$$\mathrm{CE}(f) := -\sum_{j \in [m]} \pi_j \sum_{z \in [V]} p(z \mid \mathbf{c}_j) \log\left(\sigma_z(f(\mathbf{c}_j))\right),$$

where $\pi_j \in (0,1]$ is the probability of each **distinct** context $\mathbf{c}_j$ defined as $\pi_j := \frac{1}{n}\sum_{i \in [n]} \mathbb{1}_{\mathbf{c}_i = \mathbf{c}_j}$. Then, the Taylor expansion at order 2 of $\Delta\mathrm{CE}(\alpha)$ around $\alpha = 0$ gives us:

$$\Delta\mathrm{CE}(\alpha) = \Delta\mathrm{CE}(0) + \Delta\mathrm{CE}'(0)\alpha + \frac{1}{2}\Delta\mathrm{CE}''(0)\alpha^2 + o(\alpha^2). \tag{12}$$

Obviously, $\Delta\mathrm{CE}(0) = 0$. We start by computing the derivative $\Delta\mathrm{CE}'(\alpha)$ using Lemma B.5 and chain-rule:

$$\Delta\mathrm{CE}'(\alpha) = -\sum_{j\in[m]} \pi_j \sum_{z\in[V]} p(z\mid\mathbf{c}_j)\frac{\mathrm{d}}{\mathrm{d}\alpha}\log\left(\sigma_z(f_\alpha(\mathbf{c}_j))\right) + 0$$

Computing $\frac{\mathrm{d}}{\mathrm{d}\alpha}\log\left(\sigma_z(f_\alpha(\mathbf{c}_j))\right)$ by chain rule and replacing in the above display gives

$$\Delta\mathrm{CE}'(\alpha) = -\sum_{j\in[m]} \pi_j \sum_{z\in[V]} p(z\mid\mathbf{c}_j)\frac{\sigma_z\left(f_\alpha(\mathbf{c}_j)\right)\left(M(z) - \mathbb{E}_{Z\sim\sigma\left(f_\alpha(\mathbf{c}_j)\right)}[M(Z)]\right)}{\sigma_z\left(f_\alpha(\mathbf{c}_j)\right)}$$

$$= \sum_{j\in[m]} \pi_j \sum_{z\in[V]} p(z\mid\mathbf{c}_j)\left(\mathbb{E}_{Z\sim\sigma\left(f_\alpha(\mathbf{c}_j)\right)}[M(Z)] - M(z)\right)$$

$$= \sum_{j\in[m]} \pi_j \left(\mathbb{E}_{Z\sim\sigma\left(f_\alpha(\mathbf{c}_j)\right)}[M(Z)]\sum_{z\in[V]} p(z\mid\mathbf{c}_j) - \sum_{z\in[V]} p(z\mid\mathbf{c}_j)M(z)\right)$$

By definition of the expectation $\mathbb{E}_{Z\sim\sigma\left(f(\mathbf{c}_j)\right)}[M(Z)] = \sum_{z\in[V]} p(z\mid\mathbf{c}_j)M(z)$, which gives:

$$\Delta\mathrm{CE}'(\alpha) = \sum_{j\in[m]} \pi_j \left(\mathbb{E}_{Z\sim\sigma\left(f_\alpha(\mathbf{c}_j)\right)}[M(Z)] - \mathbb{E}_{Z\sim\sigma\left(f(\mathbf{c}_j)\right)}[M(Z)]\right) \qquad \left(\text{as } \sum_{z\in[V]} p(z\mid\mathbf{c}_j) = 1.\right)$$

Under Assumption 2, we have $\sigma_z\left(f(\mathbf{c}_j)\right) = p(z\mid\mathbf{c}_j)$ which implies that $\Delta\mathrm{CE}'(0) = 0$. Using Equation (9), we get

$$\frac{\mathrm{d}}{\mathrm{d}\alpha}\mathbb{E}_{Z\sim\sigma\left(f_\alpha(\mathbf{c}_j)\right)}[M(Z)] = \mathrm{Var}_{Z\sim\sigma\left(f_\alpha(\mathbf{c}_j)\right)}(M(Z)).$$

With this, we compute the second derivative $\Delta\mathrm{CE}''(\alpha)$:

$$\Delta\mathrm{CE}''(\alpha) = \sum_{j\in[m]} \pi_j\,\mathrm{Var}_{Z\sim\sigma\left(f_\alpha(\mathbf{c}_j)\right)}(M(Z)),$$

In the statement of Theorem 3.8, we define $\mathrm{Var}_j(M(Z)) := \mathrm{Var}_{Z\sim\sigma\left(f(\mathbf{c}_j)\right)}(M(Z))$. We finish the proof by injecting the computed derivative and second derivative into the Taylor expansion of Equation (12). □

### B.6. Proof of Proposition 4.1

**Proving the expression of the steered logits $\mathbf{y}(\alpha)$ from Section 4.** Using the notation of Section 4, we apply steering at layer $\ell$ by modifying the residual stream $\mathbf{h}^{(\ell)}$. We track the effect of this intervention across subsequent layers by defining the steered residual streams $\mathbf{h}^{(k,\alpha)}$ inductively as

$$\begin{cases} \mathbf{h}^{(\ell,\alpha)} := \mathbf{h}^{(\ell)} + \alpha\mathbf{v}, & \text{(initialization)} \\ \mathbf{h}^{(k+1,\alpha)} := \mathbf{h}^{(k,\alpha)} + F(\mathbf{h}^{(k,\alpha)}), & \text{for } k\in[\ell, L-1]. \end{cases}$$

Here, $F(\mathbf{h}) := \mathrm{ATTN}(\mathrm{LN}(\mathbf{h})) + \mathrm{FFN}[\mathrm{LN}\{\mathbf{h} + \mathrm{ATTN}(\mathrm{LN}(\mathbf{h}))\}]$ captures the update applied by a single transformer block. This definition is a direct reformulation of Equation (5) adapted to our steering setting.

Unrolling this recursion up to the final layer yields

$$\mathbf{h}^{(L,\alpha)} = \mathbf{h}^{(\ell)} + \alpha\mathbf{v} + R(\alpha),$$

where $R(\alpha) := \sum_{k\in[\ell,L-1]} F(\mathbf{h}^{(k,\alpha)})$ aggregates all downstream effects induced by the steering intervention.

Substituting $\mathbf{h}^{(L,\alpha)}$ for $\mathbf{h}^{(L)}$ in $\mathbf{y} := \mathrm{LN}\left(\mathbf{h}^{(L)}\right)\mathbf{W}^\top$ then gives the steered logits expression

$$\mathbf{y}(\alpha) := \mathrm{LN}\left(\mathbf{h}^{(\ell)} + \alpha\mathbf{v} + R(\alpha)\right)\mathbf{W}^\top.$$

**Proving Proposition 4.1.** The key observation is that the presence of layer normalization inside the definition of $F$ ensures that each component of $R(\alpha)$ remains bounded (for arbitrarily large $\alpha$), *i.e.* there exists a constant $c_R$ independent of $\alpha$ such that:

$$|(R(\alpha))_{i,j}| \le c_R \,.$$

To formalize the previous display, consider RMSNorm applied to a single token representation $\mathbf{h} \in \mathbb{R}^d$:

$$\mathrm{LN}(\mathbf{h}) := \sqrt{d}\,\frac{\mathbf{h}}{\|\mathbf{h}\|} \odot \gamma \,,$$

where $\gamma \in \mathbb{R}^d$ and $\odot$ denotes the Hadamard product. Then, as $\alpha \to +\infty$,

$$\mathrm{LN}(\mathbf{h} + \alpha\mathbf{v}) = \sqrt{d}\,\frac{\mathbf{h} + \alpha\mathbf{v}}{\|\mathbf{h} + \alpha\mathbf{v}\|} \odot \gamma \;\longrightarrow\; \sqrt{d}\,\frac{\mathbf{v}}{\|\mathbf{v}\|} \odot \gamma = \mathrm{LN}(\mathbf{v})\,,$$

and, similarly, as $\alpha \to -\infty$,

$$\mathrm{LN}(\mathbf{h} + \alpha\mathbf{v}) \;\longrightarrow\; \left(-\sqrt{d}\,\frac{\mathbf{v}}{\|\mathbf{v}\|}\right) \odot \gamma = \mathrm{LN}(-\mathbf{v})\,,$$

The same argument applies to LayerNorm, **with the caveat** that the condition $\mathbf{v}_{i,:} \ne \mathbf{0}$ becomes $P\mathbf{v}_{i,:} \ne \mathbf{0}$, where $P := I_d - \frac{1}{d}\mathbf{1}\mathbf{1}^\top$ ($I_d \in \mathbb{R}^{d\times d}$ is the identity matrix and $\mathbf{1} \in \mathbb{R}^d$ is the all-ones vector). As a result, the dominant term in $\mathbf{h}^{(\ell)} + \alpha\mathbf{v} + R(\alpha)$ as $\alpha \to \pm\infty$ is $\alpha\mathbf{v}$, meaning

$$\left(\mathbf{h}^{(\ell)} + \alpha\mathbf{v} + R(\alpha)\right)_{i,j} \sim_{\pm\infty} \alpha\mathbf{v}_{i,j}\,.$$

This directly yields

$$\lim_{\alpha\to\pm\infty} \mathrm{LN}\big(\mathbf{h}^{(\ell)} + \alpha\mathbf{v} + R(\alpha)\big)\,\mathbf{W}^\top = \mathrm{LN}(\pm\mathbf{v})\,\mathbf{W}^\top \,.$$

$\square$

