# OpenReview forum: "Towards Understanding Steering Strength"
_ICML.cc/2026/Conference — ICML 2026 regular_

### Official Review · Reviewer_xH7n · 2026-03-06

**Soundness:** 3
**Presentation:** 2
**Significance:** 2
**Originality:** 3
**Overall Recommendation:** 4
**Confidence:** 3

**Summary:**

This paper addresses the limited understanding of steering strength $\alpha$ by employing a simplified Unconstrained Features Model (UFM) formulation. The authors characterize the non-monotonic "bump" behavior in token-level probabilities, the sigmoidal trend of concept probability, and locally quadratic performance degradation. These theoretical discoveries are validated through extensive experiments across eleven language models. Ultimately, the work provides a formal foundation to ground what has previously been purely empirical steering practices.

**Compliance With Llm Reviewing Policy:**

Affirmed.

**Final Justification:**

The authors' rebuttal addressed my concerns to some degree by adding experiments on complex tasks like sycophancy and reasoning, and by explaining empirical discrepancies through sampling-level degeneration. This research serves as a theoretical first step in understanding how steering vector weight selection affects performance. However, I maintain a conservative view regarding its practical utility. The authors note that not every token exhibits the identified phenomena but fail to provide a method to identify which specific tokens are affected. This gap makes it difficult to reliably transfer the toy model findings to real LLMs. Additionally, the real-world applications proposed in the rebuttal still show limited practical feasibility. While there is clear room for improvement in making the method applicable to real-world scenarios, the theoretical foundation is meaningful. Therefore, I will maintain my score of Weak Accept.

**Key Questions For Authors:**

1) Please refer to **Weakness 1** and account for the observed inconsistencies between the experimental results and the main derivations. Do these results imply that the derivations only hold for specific models or settings rather than being general?

2) How can the derivations in this paper be used to practically guide strength selection and potentially optimize the selection process beyond trial-and-error or blind grid search?

3) How do these derivations apply to more difficult steering tasks (e.g., hallucination or reasoning), and do you anticipate patterns more complex than those identified in this paper?

**Limitations:**

yes

**Strengths And Weaknesses:**

**Strengths**
1) This work is quite meaningful, as it provides a more formal framework for steering strength selection instead of relying solely on empirical trial-and-error or blind grid search.
2) The authors use rigorous proofs to characterize the behavior of steering strengths through detailed phenomenon analysis.
3) Extensive experimental results cover a wide range of models, steering scenarios, and layer selections.

**Weaknesses**

1) While the paper provides extensive results, several observations appear inconsistent with the main derivations. For example, the number of tokens that increase monotonically seems to match or even surpass those exhibiting a "**bump**" in Figures A.7 and A.8, which weakens the "**sweet spot**" argument. Additionally, regarding the sigmoidal trend, quite a few concept probability curves encounter a peak and then degrade rather than continuing to increase.
2) The paper provides a theoretical justification for current empirical practices for steering strength selection, but its practical significance would be improved if it offered explicit guidelines (e.g., a method to simplify grid searches) for selecting the optimal strength.
3) The experimental tasks are limited to relatively simple topic-induced steering, which may result in relatively simple patterns. It remains to be explored how these patterns translate to more difficult steering tasks, such as mitigating hallucination, sycophancy, or improving reasoning.

---

> ### Author Rebuttal · Authors · 2026-03-30
>
> We thank the reviewer for the thorough feedback, and recognizing that our paper takes an important step toward a "rigorous" understanding of how steering strength affects key quantities (next-token probabilities, concept presence, and cross-entropy), which may help move strength selection beyond "empirical trial-and-error or blind grid search". We also thank the reviewer for highlighting "the breadth of our empirical results" (11 LLMs, all layers, several steering design choices, 8 concepts, etc.). We believe that most concerns are addressed by **new experiments** available at https://anonymous.4open.science/r/rebuttal-icml-2026-steering-9C4D/README.md
>
> - *Applicability to more difficult steering tasks (sycophancy, reasoning, ...).*
> We agree this is important, and therefore ran additional experiments on more difficult steering tasks. We provide anonymized results for **sycophancy, math, medical, and coding** (supplementary Figures 8-13), which will be added to the main paper.
> Importantly, we ran the same experiments as in the main paper for these four concepts: next-token probabilities, concept presence, and cross-entropy, and the trends remain **qualitatively similar** to the main paper.
>
> - *Apparent inconsistencies between the derivations and Figures A.7/A.8.*
> We agree with this assessment, which is perhaps not clearly apparent in the current writing. We will tone down the "sweet spot" discussion around line 233. That said, we do not believe these observations contradict the theory.
> First, our derivations do **not** predict that every token must exhibit a bump: some tokens may increase monotonically, especially those with maximal log-odds $M(z)$. Second, even when a bump is present, its location may vary across tokens, so over a finite plotted range some curves can look nearly monotonic.
> More importantly, our **sweet spot** claim does not rely solely on observing bumps for most tokens. The bump is only one indication, not a formal proof. The stronger basis is instead the combination of: (i) increasing concept presence (Theorem 3.6), and (ii) degrading output quality as steering strength grows (Theorem 3.8). Thus, the token-level bump predicted by Theorem 3.3 is only one mechanism supporting this picture.
> We will add around line 233 that this bump pattern suggests a possible steering "sweet spot:" a range of $\alpha$ where target-token probabilities increase while the next-token distribution has not yet collapsed onto a few tokens, helping preserve output quality.
>
> - *Regarding concept-presence curves that peak and then decrease.*
> This is briefly discussed in the main text (line 437), but we will make it clearer. For some steering vectors and sufficiently large $\alpha$, autoregressive sampling can produce **degenerated outputs**, which are not captured by our next-token theory. We observe two common failure modes: (i) repetitive concept-related outputs, e.g. *“life is love love love love joy joy …”*; and (ii) largely unreadable continuations, e.g. *“life is ,,,,,,,,,,,............,,,,,,,,,.”* In the second case especially, the output may contain no meaningful realization of the concept, so measured concept presence decreases. We therefore interpret peak-then-decrease behavior as a **sampling-level degeneration effect**, rather than a contradiction of the idealized next-token derivation.
> A second difficulty, especially for harder tasks such as sycophancy, is measurement: concept-presence curves can look less sigmoidal because it is harder for a judge model to detect such concepts than simpler ones like sentiment. For example, at large steering strengths, the model may repeat words such as "amazing." While this can be characteristic of sycophantic output, it is not sufficient for either a judge LLM or a human to classify the output as genuinely sycophantic. As a result, measured concept presence may decrease again at high steering strengths. More broadly, we do not view this as a limitation of the theory to simple concepts; the main challenge is constructing reliable positive/negative sets of prompts and robust judges for harder steering tasks.
>
> - *How the theory can guide practical strength selection.*
> This is an interesting methodological direction, but goes beyond the scope of the present paper. Nonetheless, the current theory already suggests a **two-step selection strategy**. First, it helps restrict the search space to $[0,a]$, where $a$ is the largest steering strength before the model exhibits clear degeneration, such as probability mass concentrating on only a few tokens or the output becoming effectively prompt-independent, as in Proposition 4.1. Second, within $[0,a]$, one can calibrate $\alpha$ by maximizing concept presence subject to a quality budget, for example requiring task accuracy or cross-entropy degradation to remain below a threshold $\eta$. Thus, while the theory does not eliminate search entirely, it already **narrows and structures it beyond blind grid search**.

---

> > ### Author Rebuttal · Reviewer_xH7n · 2026-04-02
> >
> > Thanks for the response. To further strengthen the paper, I still suggest filtering out the degenerated outputs from your statistical analyses, as this would provide a much clearer view of the underlying trends. Additionally, expanding the discussion section to elaborate on practical applications (and the current one included in rebuttal does not seem quite practical yet) would significantly enhance the work's impact and utility for the broader community.

---

> > > ### Author Response · Authors · 2026-04-05
> > >
> > > We thank the reviewer for acknowledging our rebuttal and are pleased that it addressed all of their concerns.
> > >
> > > We also thank the reviewer for their helpful suggestion. In response, we ran new experiments, following a very similar setup with the difference that we **clean degenerate steered completions before computing the concept-presence curves.** We then recompute the behavior plots before and after cleaning. As the reviewer anticipated, this yields substantially clearer concept-presence curves; the supplementary results for both "evil" and "joy" on gemma-3-1b are provided at https://anonymous.4open.science/r/rebuttal-icml-2026-steering-9C4D/fig_cleaning_prompt.pdf (the figure file name is fig_cleaning_prompt.pdf).
> > >
> > > To be more precise, we use the same judge model throughout this procedure, namely **Gemma 3 12B Instruct**, but with two different prompts in a two-stage pipeline.
> > >
> > > 1. **Cleaning stage.** We first prompt the judge model as a cleaner and ask it whether a completion should be kept for evaluation.
> > >    - For **"evil"**, the LLM is instructed to keep completions only when they still express full words, while rejecting filler-like outputs dominated by fragments such as: all, little, oh, want, be, ...
> > >    - For **"joy"**, the LLM is also instructed to keep completions only when they still express express full words, while rejecting filler-like outputs.
> > >    - **In both cases**, punctuation-only outputs (e.g., ''life is ,,,,,,,,,,,............,,,,,,,,,'') are removed as degenerate by the judge LLM.
> > >
> > > 2. **Scoring stage.** After cleaning, we use the same judge model in a separate inference step to score the remaining completions for concept presence, exactly as described in the paper.
> > >
> > > This two-stage setup lets us test whether irregularities in the steering plots are caused by degenerate generations rather than by the steering effect itself. For both steering vector ("evil" and "joy"), the cleaned plots are noticeably smoother and more sigmoid-shaped. We will run additional experiments and add these results to the revised version of the paper, since they further support the idea that ill-behaved concept-presence curves are due to degenerate outputs.
> > >
> > > We hope this fully answers the reviewer's curiosity, and that they will consider raising their score accordingly, as they had done previously.

---

### Official Review · Reviewer_dmkZ · 2026-03-12

**Soundness:** 3
**Presentation:** 2
**Significance:** 2
**Originality:** 3
**Overall Recommendation:** 4
**Confidence:** 4

**Summary:**

This paper performs a theoretical analysis of steering strength for the Unconstrained Features Model (UFM; Zhao et al., 2024), an idealized/simplified model of modern LLMs in which a context is first embedded into a vector space and then linearly mapped back to vocabulary logits. Under strong assumptions (most importantly, that the latent concepts to be steered are disjoint in token space and that the UFM is perfectly trained), the authors derive closed-form characterizations of how steering strength affects the model’s predictions. The main results are that (1) for individual tokens, steered probabilities typically show a non-monotonic “bump” as steering strength increases; (2) at the concept level, the probability of the target concept follows a tanh-like sigmoidal curve as a function of steering strength; and (3) steering always degrades global performance, with cross-entropy increasing locally in a quadratic/U-shaped way around zero. The paper also extends the analysis toward real transformers, arguing that at very large steering strengths the output becomes largely prompt-independent, and it validates these qualitative predictions empirically across multiple language models.

**Compliance With Llm Reviewing Policy:**

Affirmed.

**Key Questions For Authors:**

See weaknesses

**Limitations:**

Yes

**Strengths And Weaknesses:**

Strengths:
- The paper addresses an important and under-theorized question in activation steering: how steering strength controls the trade-off between steering efficacy and output distortion.
- The theory is useful despite the simplified setting.
- The paper is easy to follow. The setup, assumptions, and progression from token-level to concept-level to performance-level analysis are clearly organized.

Weaknesses:
- The strongest theoretical performance result is somewhat less surprising because of the assumptions. Since the UFM is assumed to be perfectly trained on the target distribution, it is natural that perturbing activations away from the optimum should locally worsen training cross-entropy. So while Theorem 3.8 is still useful in giving the exact local form and coefficient of degradation, the qualitative conclusion that steering hurts global performance is partly built into the idealized UFM setup.
- The concept model is very restrictive. The paper assumes that concepts are disjoint subsets of tokens, and even acknowledges that this is unrealistic because abstract concepts rarely map cleanly onto token subsets and contexts may mix multiple concepts. This makes the theory tractable, but it also limits how well the framework captures semantic concepts in real language. For example, concepts such as “depression” and “joy,” or negated expressions like “not depressed” and “not happy,” are not naturally representable as disjoint token sets. It is therefore unclear how the analysis extends to compositional, negated, or overlapping semantic concepts.
- The theory is fundamentally a next-token analysis. The UFM predicts next-token distributions for fixed contexts, and the main theorems characterize how steering changes those next-token probabilities. In actual generation, however, steered outputs are fed back into the context, so steering affects a full autoregressive trajectory rather than a single conditional distribution. That feedback loop is not captured by the theory, making it unclear how directly the results transfer to multi-step generation.

---

> ### Author Rebuttal · Authors · 2026-03-30
>
> We thank the reviewer for the thorough feedback, and for recognizing that our paper addresses an "important and under-theorized question:" how steering strength affects next-token probabilities, concept presence, and cross-entropy. We also thank the reviewer for noting that the paper is "clearly organized" and "easy to follow."
>
> - *The theoretical result on locally U-shaped cross-entropy (Th. 3.8) not surprising given perfect training assumption.*
> While we agree that Theorem 3.8 is intuitive under the perfect training assumption, we believe that is it still meaningful in two aspects. First, it provides a simple theoretical model whose behavior under steering closely matches what we observe in practical LLMs. In particular, **next-token probabilities, concept presence, and cross-entropy** exhibit qualitatively similar behavior. Second, the theorem gives the **exact local quadratic form** of the cross-entropy under steering, together with its **leading coefficient**.
>
>   We now discuss the role of the perfect training assumption, what can still be shown without it, and what breaks. For Theorem 3.8, perfect training is sufficient but not necessary for the local U-shape of cross-entropy under steering. The same conclusion holds as soon as $\Delta CE'(0)=0$. A weaker sufficient condition is $\mathbb{E}\_{Z \sim p}[M(Z)] = \mathbb{E}\_{Z \sim q}[M(Z)]$, where $M(Z)$ is the log-odds from Definition 3.2, $p$ denotes the data next-token probabilities (Definition 2.1), and $q$ the learned next-token distribution. Empirically, the predicted local U-shape is also **consistently observed** in our experiments on real LLMs.
>
>   Another fundamental difficulty in the imperfectly-trained setting lies in Lemma B.3, the first point where Assumption 2 is used in an essential way where it allows us to represent each unsteered logit vector as $\log p(\cdot \mid \mathbf{c}_j) + \beta_j$ (line 1357). Without this step, we lose the clean statement that steering can be expressed directly in terms of the data next-token probabilities $p$. This direct dependence makes the results easy to interpret, since $p$ is defined at the dataset level and carries the concept partition structure.
>
> - *The assumptions on concepts are restrictive.*
> The disjoint-token concept model is a tractability assumption, and similar assumptions appear in prior works using concepts in their data model, such as [1,2]. Allowing concepts to overlap would break Lemma B.4, which establishes the sign of the log-odds $M(z)$. This sign result is important to show that steering works in the intended way, namely that tokens from the steered concept become more likely or exhibit a bump behavior.
> The argument breaks because the proof of Lemma B.4 relies directly on concept disjointness: target tokens receive probability $a_z$ on contexts containing the concept and $b_z$ on negative contexts. If concepts overlap, a token $z$ from the concept we want to steer may also count as in-concept for some negative contexts, and conversely an off-target token may overlap with the target concept. In that case, the sign of the log-odds $M(z)$ no longer indicates concept membership, i.e. $M(z)>0$ *iff* $z$ is a target token no longer holds. Extending the analysis beyond the disjoint-token setting is therefore an important direction for future work.
>
> - *The theory studies next-token probabilities rather than full autoregressive trajectories.*
> We agree that extending the analysis to full autoregressive generation is an important next step. But the next-token setting is the **natural starting point**, and the same quantities studied here, such as the log-odds in Definition 3.2, are precisely the ones that play a role in sampling the autoregressive trajectory. We therefore see the current theory as a necessary first step toward a future analysis of steering in full-text generation.
> Moreover, the current UFM framework is **fundamentally designed for next-token analysis** (see [3,4,5]). Once a next token is sampled and appended to the current context, the resulting input context may no longer belong to the embedding matrix $\mathbf{H} \in \mathbb{R}^{d \times m}$, where $m$ is the number of distinct training contexts. For this reason, extending the theory from next-token prediction to full autoregressive generation is not immediate, and we view it as a promising direction for future work.
>
> **References**
>
> [1] Ravfogel et al., *Emergence of Linear Truth Encodings in Language Models*, NeurIPS 2025.
> [2] Li et al., *How Do Transformers Learn Topic Structure: Towards a Mechanistic Understanding*, ICML 2023.
> [3] Thrampoulidis, *Implicit Optimization Bias of Next-Token Prediction in Linear Models*, NeurIPS 2024.
> [4] Zhao et al., *Implicit Geometry of Next-Token Prediction: From Language Sparsity Patterns to Model Representations*, COLM 2024.
> [5] Mixon et al., *Neural Collapse with Unconstrained Features*, 2020.

---

> > ### Author Rebuttal · Reviewer_dmkZ · 2026-04-02
> >
> > Thanks for the responses.

---

> > > ### Author Response · Authors · 2026-04-05
> > >
> > > We thank the reviewer for acknowledging our rebuttal and are pleased that it addressed all of the concerns raised. We hope the reviewer will consider raising their score accordingly.

---

### Official Review · Reviewer_4cef · 2026-03-13

**Soundness:** 3
**Presentation:** 2
**Significance:** 2
**Originality:** 2
**Overall Recommendation:** 4
**Confidence:** 4

**Summary:**

The paper experiment analyzes the effect of steering factor when applying steering vectors. The paper first establishes a theoretical framework from the Unconstrained Feature Model. With respect to the next token prediction, the paper first shows that in the range of the steering factor, the given next tokens exhibit a “bump” behavior where the target tokens get prompted when steering factor increases, reach peak then decrease. With respect to the concept probability, this shows the S curve with the strength of the steering vector. With respect to the cross entropy the paper shows that steering will necessarily degrade model performance. On top of the theoretical framework, the paper empirically shows the results on language models for next token probability, concept probability and cross entropy.

**Compliance With Llm Reviewing Policy:**

Affirmed.

**Final Justification:**

The rebuttal address my main concerns

**Key Questions For Authors:**

1. How much does the quality of the steering vector affect the conclusion? If the steering vector is generated with very few pairs of contrastive data, does the conclusion still hold?

2. The concept probability metric used in the experiments relies on a judge LLM？How much does this choice of judge and criterion affect the sigmoidal shape observed in Figure 7? Would a stricter or more semantically grounded evaluation change the conclusions? Did the paper cross check with a different judge?

**Limitations:**

yes

**Strengths And Weaknesses:**

## Soundness
1. The conclusion itself is clean and interesting. The analysis is build up on a nice theoretical framework

2. The steering vector used throughout is extracted with a simple difference-of-means approach (CAA), but there are many other ways to compute or parameterize steering directions — for example, representation fine-tuning , sparse autoencoder (SAE)-based directions, or learned task vectors. The theoretical results are derived specifically for the difference-of-means construction, and it is unclear how much the observations transfer to vectors trained in other ways.

3. The experiment is very limited with little concepts. And doesn’t explore more across different layers.

## Originality
1. The paper explores an underexplored direction of steering strength for linear vector conditioned on the linear hypothesis. The idea of examining the steering strength is new and novel.

2. The steering strength is also explored in other papers like Axbench (Wu et al) but from the perspective of how this influences instruction following and fluency. Related to the cross entropy and logit but in a more realistic setting.

## Presentation
1. The writing can use a lot of improvement. The paper give a lot of theoretical framework without a good connection to experimental wise what does it mean. For example, how are these concepts defined. And how does the whole vocabulary map to different concepts? Are these concepts mutually exclusive.

2. In many of the figures, the lines are not labelled. For example, in figure 6, the lines are not labeled. In figure 7 the y axis is not labelled

Significance

1. It only examined on 1d linear vector and it’s hard to understand what is the real world application beyond that .The paper can either make a more significant connection with linear hypothesis and explain what does this mean for the hypothesis and advance the understanding on it or evaluate the steering on more than just linear vectors.

---

> ### Author Rebuttal · Authors · 2026-03-30
>
> We thank the reviewer for the thoughtful feedback and for recognizing that our work provides a "novel and useful" framework for analyzing steering strength. We addressed the main concerns of the reviewer by running **extensive new experiments**, available at the following **supplementary link**: https://anonymous.4open.science/r/rebuttal-icml-2026-steering-9C4D/README.md.
>
> **Key questions**
>
> - *Do the results still hold with very few contrastive pairs?*
> Prior work [1] showed that steering can already work with a vector built from a **single contrastive pair**. We ran experiments using only **10 contrastive pairs** (supplementary Figures 3-7). The main qualitative conclusions remain unchanged.
>
> - *How much does the choice of judge affect Figure 7?*
> Prior work evaluates judge quality through instruction-following [2] and challenging judging tasks [3], showing that models without modern post-training can fail as judges. This motivated our use of Gemma 3 12B. We added experiments with **Qwen3-0.6B-Base, Qwen2.5-7B-Instruct, and Mistral-7B-Instruct** (supplementary Figures 1-2). As expected, the non-instruction-tuned model (Qwen3-0.6B-Base) yields flat concept-presence curves, while the stronger instruction-tuned judges recover similar sigmoidal trend as Gemma 3 12B.
>
> **Weaknesses**
>
> - *Why not other steering vectors (classifier, SAE)?*
> We focus on **difference-of-means** vectors because they are simple and widely-used in practice [4-6]. By contrast, learned-basis methods such as SAEs are substantially more expensive and also have known limitations [7-10]. They also do not fit naturally in our theoretical framework: in the UFM model, the embeddings $\mathbf{H}$ already have interpretable geometry [11]. Additionally, the structure of the difference-of-means vector is instrumental in our proofs.
>
> - *Experiments are limited in concepts and do not explore enough layers.*
> We believe this partly reflects that we understated our experimental coverage. As noted at line 383, Appendix A already reports **additional layers, concepts,...**. For the main analyses, **we ran experiments on all layers** (supplementary Figures 15-16), restricting only the most expensive metrics, such as cross-entropy, and reported only representative early, middle, and late layers.
> We also report **8 concepts** (Appendix Table 1), which is comparable to many papers in the field [1,4-6]. To further address the concern, we **added four concepts**: **sycophancy, math, coding, and medical** (supplementary Figures 8–13). These additional plots are qualitatively consistent with the main-text results.
>
> - *How are these concepts defined?*
> This is described in the paper (line 353) and Appendix A. Our concepts are inspired by the "personas" concepts of [12], including "evil," and extended to concepts such as "joy" and "humorous." Concretely, concept prompts are produced by a fixed LLM (Gemma 3 12B) using a prompt that asks for both a concept definition and example sentences exhibiting the concept.
>
> - *How does the vocabulary map to concepts? Are concepts mutually exclusive?*
> These points are addressed in the paper: concepts are **not mutually exclusive** and do not map cleanly to tokens (line 129).
>
> - *The work only studies 1D linear steering vectors.*
> We study 1D because a single steering direction is the most common setting [4-6]. Extending beyond 1D, following for instance [13] which explores 2D rotational steering,  is an interesting research direction. However, it would require substantial changes in our work, since the current theoretical machinery rely on the difference-of-means structure.
>
> **References**
>
> [1] Turner et al., *Steering language models with activation engineering,* 2023
>
> [2] Zeng et al., *Evaluating Large Language Models at Evaluating Instruction Following*, ICLR 2024
>
> [3] Tan et al., *JudgeBench: A Benchmark for Evaluating LLM-Based Judges,* ICLR 2025
>
> [4] Cyberey et al., *Unsupervised Concept Vector Extraction for Bias Control in LLMs,* EMNLP 2025
>
> [5] Potertì et al., *Can Role Vectors Affect LLM Behaviour?* EMNLP 2025
>
> [6] Wehner et al., *Taxonomy, Opportunities, and Challenges of Representation Engineering for Large Language Models* TMLR 2025
>
> [7] Leask et al., *Sparse Autoencoders Do Not Find Canonical Units of Analysis* ICLR 2025
>
> [8] Kantamneni et al., *Are Sparse Autoencoders Useful? A Case Study in Sparse Probing* ICML 2025
>
> [9] Wu et al., *AxBench: Steering LLMs? Even Simple Baselines Outperform Sparse Autoencoders* ICML 2025
>
> [10] Cui al., *On the Limits of Sparse Autoencoders: A Theoretical Framework and Reweighted Remedy* ICLR 2026
>
> [11] Zhao et al., *Implicit geometry of next-token prediction: From language sparsity patterns to model representations.* COLM 2024
>
> [12] Chen al., *Persona Vectors: Monitoring and Controlling Character Traits in Language Models* preprint, 2025
>
> [13] Vu et al., *Angular Steering: Behavior Control via Rotation in Activation Space*  NeurIPS 2025

---

> > ### Author Rebuttal · Reviewer_4cef · 2026-04-03
> >
> > The new experiments address my concern but I would still encourage the author to supervised finetuning for their experiment (different from SAE). It will be nice to see for a vector that's sharper and more precise on the concept would the steering strength tolerate a larger variance.

---

> > > ### Author Response · Authors · 2026-04-05
> > >
> > > We thank the reviewer for acknowledging our rebuttal and are pleased that it addressed all of their concerns. Following the reviewer’s suggestion, we conducted a preliminary experiment using steering vectors derived from the Rank-1 Representation Finetuning method (ReFT-r1) from [9]. We ran experiments for the models {gpt2-large, gemma-3-1b-it}, and the concepts {evil, joy}. The results can be found at https://anonymous.4open.science/r/rebuttal-icml-2026-steering-9C4D/figs_reft-r1.pdf (figure file name is figs_reft-r1.pdf)
> > >
> > > In these experiments, the resulting figures exhibit behavior similar to that observed with the difference-in-means steering vector across all three quantities: **next-token probabilities**, which show bumps and tokens with monotonic probability changes under steering; **concept presence**, which follows a sigmoidal-like shape; and **cross-entropy**, which is locally U-shaped. This is an interesting avenue: while the theory relies on the difference of means computation of the steering vectors, the observed behavior may hold for a larger class of steering directions. We will conduct further experiments and include the results in the revised version.
> > >
> > > We thank the reviewer for this suggestion. We hope that these results satisfy the reviewer’s curiosity and that they will consider raising their score accordingly.
> > >
> > > **References:**
> > >
> > > [9] Wu et al., AxBench: Steering LLMs? Even Simple Baselines Outperform Sparse Autoencoders ICML 2025

---

### Official Review · Reviewer_JWMz · 2026-03-13

**Soundness:** 4
**Presentation:** 3
**Significance:** 3
**Originality:** 3
**Overall Recommendation:** 5
**Confidence:** 4

**Summary:**

This paper presents a theoretical analysis on a simplified model, of how difference-of-means steering strength $\alpha$ impacts next token probability, presence of a concept, and cross-entropy. The authors then validate the influence of $\alpha$ on the mentioned quantities in practice.

**Compliance With Llm Reviewing Policy:**

Affirmed.

**Final Justification:**

My concerns have been addressed and I am keeping my (high) score.

**Key Questions For Authors:**

N/A

**Limitations:**

yes

**Strengths And Weaknesses:**

## Strengths
This paper is exactly what we need in the activation steering literature: it's rigorous (though on a simplified model, it states the assumptions clearly) and the theoretical results are qualitatively validated on real LLMs. The writing is very clear and well-structured.

## Weaknesses

Overall, this is a great paper, but the novelty is somewhat overstated. Cheng and Amo Alonso, 2024, as well as a followup Hedstrom et al., 2025 conduct a similar theoretical analysis of steering strength, where, e.g., Cheng and Amo Alonso derive the optimal steering vector in closed form and a one-to-one correspondence is established between steering strength and presence of a concept. Please adjust the claims in the abstract/intro and also compare your method to the Cheng and Hedstrom papers in the related work. As a starting point, some novel aspects wrt Cheng and Hedstrom is that you consider (1) difference-of-means steering; (2) the impact on next-token probabilities and cross-entropy.

Nit: In Fig 6, please make the colors for the top/bottom rows consistent

---

> ### Author Rebuttal · Authors · 2026-03-30
>
> We thank the reviewer for the positive feedback, and in particular for recognizing that this is "exactly what we need in this literature" by being a "rigorous" step toward assessing the impact of steering strength on several key quantities (cross-entropy, next-token probabilities, and concept presence) with "extensive empirical validation" on real-world LLMs.
>
> We also thank the reviewer for pointing out additional references, especially [1]. While there are similarities with our work, however, we do not believe that these papers address the same theoretical questions as ours:
>
>  - [1] introduces and study **adaptive, context-dependent steering** and derive guarantees for these interventions. More specifically, [1] formulates steering as a control problem: intervening at a given layer to guide generations toward a desirable region when necessary.
> In this respect, the obtained results fundamentally differ from ours. In particular, Lemma C.1 in [1] shows that activation-space guarantees can transfer to guarantees on the concept presence, provided a calibration condition holds. More precisely, under strong assumptions on the probe function, by constraining activations to a region, one can constrain the concept presence to an interval $\mathcal{A}^\star$. By contrast, our contribution is to derive the **explicit functional form** of concept presence in the output as steering strength varies which is stronger than Lemma C.1. Thus, [1] provides a conditional transfer guarantee, whereas our result is about the expression of the concept presence curve itself.
> Finally, [1] empirically observe a monotonic relationship between steering strength and concept presence in the output (as [3]). We did not find a theoretical justification for this relationship in [1] and [3], in contrast to our Theorem 3.6.
>
> - [2] (which we already cite below Th. 3.3) studies adaptive steering for error mitigation. For instance, their Eq. (6) describes the best choice of $\alpha$ for optimization problem (5), which is dependent on the linear probe and the input prompt. In contrast, we study the effect of varying $\alpha$ in all generality, without relying on linear probes. Therefore, we do not see a strong overlap with our work.
>
> - We therefore view the contributions as **complementary**: the cited works study **how to calibrate adaptive steering interventions, and under what conditions guarantees transfer to the output text**, whereas our paper studies **the effect of varying the steering strength over a wide range (e.g., $\alpha \in [-200,200]$) for difference-of-means steering**.
>
>
> To complement our related work section, we will add the following paragraph:
> > Additionally, several prior works have proposed steering methods with adaptive steering strength. Cheng and Amo Alonso (2024) formulate steering as a context-dependent activation control problem and derive guarantees for driving activations into a prescribed region, while Hedström et al. (2025) develop an adaptive steering intervention for error mitigation.
>
> **References:**
>
> [1] Cheng et al., *Linearly Controlled Language Generation with Performative Guarantees*, preprint 2025.
>
> [2] Hedström et al., *To Steer or Not to Steer? Mechanistic Error Reduction with Abstention for Language Models*,  ICML 2025.
>
> [3] von Rütte et al., *A Language Model's Guide Through Latent Space*, ICML 2024.

---

> > ### Author Rebuttal · Reviewer_JWMz · 2026-04-01
> >
> > Great, my concerns have been resolved. Apologies for missing the Hedstrom citation---the added paragraph is clear and better situates the present work in the LM control landscape.

---

> > > ### Author Response · Authors · 2026-04-05
> > >
> > > We thank the reviewer for acknowledging our rebuttal and are pleased that it addressed all of their concerns.

---

### Decision · Program_Chairs · 2026-04-30

**Decision:**

Accept (regular)

**Comment:**

This paper studies a crucial but underexplored question in activation steering: how steering strength affects behavior and performance. The authors provide a theoretical analysis in the Unconstrained Features Model and characterize several qualitative phenomena, including non-monotonic effects on token probabilities, sigmoidal concept-presence curves, and local degradation of cross-entropy. These predictions are then validated empirically across a broad set of language models.

Reviewers generally agreed that the paper makes a meaningful contribution by bringing theoretical grounding to a topic that is often treated empirically. The strengths most frequently highlighted were the clarity of the theoretical setup, the relevance of the question, and the breadth of the empirical validation. The paper was viewed as a useful first step toward understanding how to choose steering strength more systematically.

The main concerns were about the scope of the theory and its practical implications: the analysis is tied to a simplified setting and primarily to difference-of-means steering; some assumptions are restrictive, especially the concept model; and the paper stops short of providing a practically actionable method for strength selection. Reviewers also raised issues about positioning with respect to prior work, clarity of presentation, and the extent to which the results transfer beyond next-token analysis to more realistic steering settings.

In my assessment, the rebuttal addressed these concerns reasonably well. The authors clarified the relation to prior work, added experiments on harder concepts and additional judges, and provided further evidence that the qualitative trends are robust across models, layers, and even alternative steering vectors. At the same time, some limitations remain: this is still primarily a theoretical first step rather than a practically complete recipe, and the gap between the toy-model analysis and real autoregressive steering is not fully closed.

Overall, I find the paper technically solid, novel in its focus, and likely to be useful to researchers working on activation steering and LLM control. The contribution is somewhat bounded by its assumptions and practical scope, which is why I view this as a weak accept. Given the reviewer consensus and the successful rebuttal, I lean positive, though I think the final decision could reasonably benefit from SAC’s comments relative to the bar for acceptance.